# Incident allergic diseases in post-COVID-19 condition: multinational cohort studies from South Korea, Japan and the UK

Jiyeon Oh[1,12], Myeongcheol Lee[2,3,12], Minji Kim [2,3,12], Hyeon Jin Kim[2,3,12], Seung Won Lee [4], Sang Youl Rhee [2,5], Ai Koyanagi[6], Lee Smith[7], Min Seo Kim [8], Hayeon Lee [2,9,13] ✉, Jinseok Lee [9,10,13] ✉ & Dong Keon Yon [1,2,3,11,13] ✉

As mounting evidence suggests a higher incidence of adverse consequences, such as disruption of the immune system, among patients with a history of COVID-19, we aimed to investigate post-COVID-19 conditions on a comprehensive set of allergic diseases including asthma, allergic rhinitis, atopic dermatitis, and food allergy. We used nationwide claims-based cohorts in South Korea (K-CoV-N; $n = 836,164$; main cohort) and Japan (JMDC; $n = 2,541,021$; replication cohort A) and the UK Biobank cohort (UKB; $n = 325,843$; replication cohort B) after 1:5 propensity score matching. Among the 836,164 individuals in the main cohort (mean age, 50.25 years [SD, 13.86]; 372,914 [44.6%] women), 147,824 were infected with SARS-CoV-2 during the follow-up period (2020−2021). The risk of developing allergic diseases, beyond the first 30 days of diagnosis of COVID-19, significantly increased (HR, 1.20; 95% CI, 1.13−1.27), notably in asthma (HR, 2.25; 95% CI, 1.80−2.83) and allergic rhinitis (HR, 1.23; 95% CI, 1.15−1.32). This risk gradually decreased over time, but it persisted throughout the follow-up period ($\geq 6$ months). In addition, the risk increased with increasing severity of COVID-19. Notably, COVID-19 vaccination of at least two doses had a protective effect against subsequent allergic diseases (HR, 0.81; 95% CI, 0.68−0.96). Similar findings were reported in the replication cohorts A and B. Although the potential for misclassification of pre-existing allergic conditions as incident diseases remains a limitation, ethnic diversity for evidence of incident allergic diseases in post-COVID-19 condition has been validated by utilizing multinational and independent population-based cohorts.

Severe acute respiratory syndrome coronavirus 2 (SARS-CoV-2) infection, which is responsible for the coronavirus disease 2019 (COVID-19) pandemic, has led to mortality and morbidity globally[1,2]. As we enter a post-COVID-19 era, mounting evidence suggests persistent or new-onset health consequences, lasting more than a month, following the acute phase of SARS-CoV-2 infection[3]. This condition is now

described as 'post-COVID-19 conditions', or post-acute sequelae of COVID-19[3]. Several studies have demonstrated the disruptions of the immune system due to post-COVID-19 conditions[4–6], which may contribute to allergic outcomes.

Little is known about the long-term allergic outcomes of COVID-19. However, the association between allergic diseases and COVID-19

outcomes[7], has been well-described in previous studies. In addition, clinical outcomes, including diabetes[8], cardiovascular diseases[9], neurologic diseases[4], and dyslipidemia[10], associated with post-COVID-19 conditions have been demonstrated in nationwide cohort studies. Indeed, ~45% of the global infected population has experienced post-COVID-19 conditions[11], there is a strong need for a precise investigation of allergic disease burden in the post-acute phase of infection, which may provide guidelines for constructing the care strategies for patients with post-COVID-19 conditions.

In this study, we examined the long-term allergic sequelae after SARS-CoV-2 infection compared to contemporary controls comprising non-infected individuals, using a multinational, large-scale, and population-based cohort (Table S1). As ethnicity is suggested to be novel risk factors for developing post-COVID-19 conditions (Table S2)[12], we constructed the cohort consisting of over 22 million participants using multinational cohort studies of South Korea, Japan, and the UK. We further analyzed whether the risk of allergic diseases after COVID-19 diagnosis attenuated over time and whether COVID-19 vaccination has a protective effect against the onset of allergic diseases.

## Results
In the main cohort of South Korea, there are a total of 10,027,506 participants (mean age 48.4 [standard deviation, 13.4] years; 5,000,621 [49.9%] women). After 1:5 propensity score matching, 836,164 individuals were eventually included in this study ($n = 147,824$ for SARS-CoV-2 infected). For the replication cohorts of Japan and the UK, the final sample size after matching was 2,541,021 ($n = 542,497$ for COVID-19 infected) and 325,843 ($n = 76,894$ for SARS-CoV-2 infected), respectively (Fig. 1 and Table 1). The demographic characteristics of the participants in the nationwide unmatched and matched cohorts of Japan and the UK are presented in Tables S3–S5. Standardized mean difference (SMD) values, presented in Table 1, suggested that there were no major imbalances of the covariates after propensity score matching.

Our study includes a detailed justification for diverse sensitivity analyses, as provided in Table S6. According to the maximally adjusted model (model 2) in Table 2, the increased risks of incident overall allergic diseases (hazard ratio [HR], 1.20; 95% confidence interval [CI],

1.13 to 1.27), asthma (HR, 2.25; 95% CI, 1.80 to 2.83) and allergic rhinitis (AR; HR, 1.23; 95% CI, 1.15 to 1.32) were associated with SARS-CoV-2 infection; however, no significant risk was observed in atopic dermatitis (AD; HR, 1.15; 95% CI, 0.96 to 1.37) and food allergy (FA; HR, 0.85; 95% CI, 0.71 to 1.00). The risk of overall allergic diseases attenuated but remained persisted over time while the strength of time attenuation effect varies among two cohorts (Table 3).

The main cohort which contains information on vaccines enables us to perform additional analyses assessing the influence of COVID-19 severity, vaccination and SARS-CoV-2 strains on allergic outcomes. Moderate-to-severe COVID-19 resulted in a higher risk of overall allergic sequelae (HR, 1.48; 95% CI, 1.31 to 1.66) compared to the mild (HR, 1.14; 95% CI, 1.07 to 1.21). Relative to the contemporary controls of those who have not been infected with SARS-CoV-2, both infections with original and delta strains were associated with an increased risk of incident allergic diseases.

COVID-19 vaccination was associated with the attenuated risk (vaccinated once [HR, 1.44; 95% CI, 1.22 to 1.69] and at least twice [HR, 0.81; 95% CI, 0.68 to 0.96]). Interestingly, the risks of overall allergic disease and its subtypes, including asthma, AD, AR, and FA were no longer significantly higher than that of the non-infected controls when vaccinated at least twice (Table 4 and S7–S10).

We performed stratification analyses by sex, age, income level, Charlson comorbidity index (CCI), body mass index, alcohol intake, aerobic physical activity and SARS-CoV-2 strains to handle unintended mediated effects. The consistent results for the long-term allergic consequences of infection with SARS-CoV-2 were shown in the main cohort and replication cohort A (Tables S11–S20). Additionally, we conducted sensitivity analyses to examine the impact of COVID-19 severity on allergic diseases. According to these analyses, the results primarily indicated higher incidences in patients with moderate to severe COVID-19 (Tables S21 and S22).

## Discussion
### Key findings of this study
This is the first study, to date, that provides comprehensive evidence for the association between SARS-CoV-2 infection and subsequent

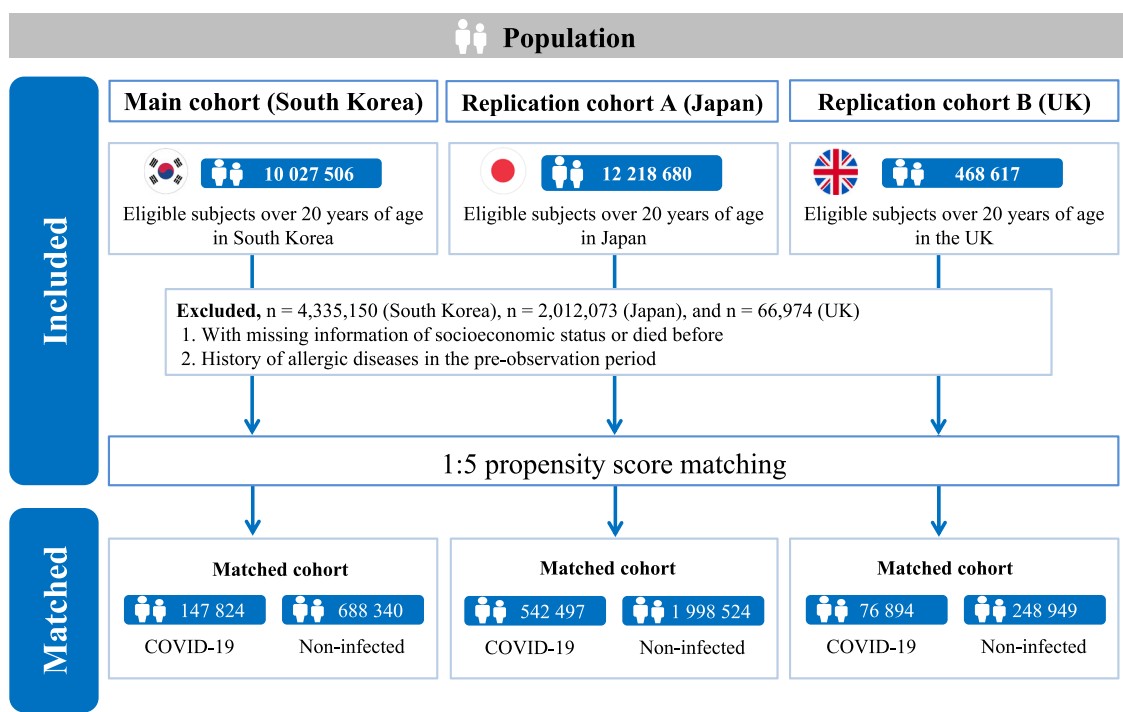

**Fig. 1 |** Study population in South Korea, Japan and UK cohorts.

**Table 1 | Baseline characteristics for 1:5 propensity score-matched in main cohort (South Korea)**

| Covariates | 1:5 matched cohort (n = 836,164) | | | |
| --- | --- | --- | --- | --- |
| | Total | COVID-19 (n = 147,824) | Non-COVID-19 (n = 688,340) | SMD |
| Age, years, mean (SD) | 50.25 (13.86) | 50.98 (14.93) | 50.09 (13.61) | 0.062 |
| Age, years, n (%) | | | | 0.015 |
| 20–39 | 210,320 (25.15) | 37,030 (25.05) | 173,290 (25.18) | |
| 40–59 | 362,624 (43.37) | 63,421 (42.90) | 299,203 (43.47) | |
| ≥60 | 263,220 (31.48) | 47,373 (32.05) | 215,847 (31.36) | |
| Sex, n (%) | | | | 0.014 |
| Male | 463,250 (55.40) | 81,075 (54.85) | 382,175 (55.52) | |
| Female | 372,914 (44.60) | 66,749 (45.15) | 306,165 (44.48) | |
| Region of residence, n (%) | | | | 0.002 |
| Urban | 448,021 (53.58) | 79,097 (53.51) | 368,924 (53.60) | |
| Rural | 388,143 (46.42) | 68,727 (46.49) | 319,416 (46.40) | |
| Household income, n (%) | | | | 0.011 |
| Low (0–39 percentile) | 362,567 (43.36) | 63,769 (43.14) | 298,798 (43.41) | |
| Middle (40–79 percentile) | 318,188 (38.05) | 56,080 (37.94) | 262,108 (38.08) | |
| High (80–100 percentile) | 155,409 (18.59) | 27,975 (18.92) | 127,434 (18.51) | |
| Charlson comorbidity index, n (%) | | | | 0.037 |
| 0 | 724,067 (86.59) | 126,952 (85.88) | 597,115 (86.75) | |
| 1 | 76,829 (9.19) | 13,712 (9.28) | 63,117 (9.17) | |
| ≥2 | 35,268 (4.22) | 7160 (4.84) | 28,108 (4.08) | |
| History of cardiovascular disease, n (%) | 40,586 (4.85) | 7545 (5.10) | 33,041 (4.80) | 0.014 |
| History of chronic kidney disease, n (%) | 17,826 (2.13) | 3399 (2.30) | 14,427 (2.10) | 0.014 |
| History of chronic obstructive pulmonary disease, n (%) | 2316 (0.28) | 476 (0.32) | 1840 (0.27) | 0.010 |
| History of medication use for diabetes, n (%) | 178,955 (21.40) | 32,463 (21.96) | 146,492 (21.28) | 0.017 |
| History of medication use for hyperlipidemia, n (%) | 152,766 (18.27) | 27,918 (18.89) | 124,848 (18.14) | 0.019 |
| History of medication use for hypertension, n (%) | 75,926 (9.08) | 14,563 (9.85) | 61,363 (8.91) | 0.032 |
| Body mass index, kg/m$^2$, n (%) | | | | 0.013 |
| Underweight (<18.5) | 22,105 (2.64) | 4108 (2.78) | 17,997 (2.61) | |
| Normal (18.5–23.0) | 268,626 (32.13) | 47,707 (32.27) | 220,919 (32.09) | |
| Overweight (23.0–25.0) | 194,116 (23.22) | 34,255 (23.17) | 159,861 (23.22) | |
| Obese (≥25.0) | 351,170 (42.00) | 61,718 (41.75) | 289,452 (42.05) | |
| Unknown | 147 (0.02) | 36 (0.02) | 111 (0.02) | |
| Blood pressure, n (%) | | | | 0.012 |
| SBP < 140 mmHg and DBP < 90 mmHg | 705,753 (84.40) | 124,254 (84.06) | 581,499 (84.48) | |
| SBP ≥ 140 mmHg or DBP ≥ 90 mmHg | 125,873 (15.05) | 22,790 (15.42) | 103,083 (14.98) | |
| Unknown | 4538 (0.54) | 780 (0.53) | 3758 (0.55) | |
| Fasting blood glucose, mg/dL, n (%) | | | | 0.007 |
| <100 | 488,560 (58.43) | 85,994 (58.17) | 402,566 (58.48) | |
| ≥100 | 342,918 (41.01) | 61,031 (41.29) | 281,887 (40.95) | |
| Unknown | 4686 (0.56) | 799 (0.54) | 3887 (0.56) | |
| Serum total cholesterol, mg/dL, n (%) | | | | 0.007 |
| <200 | 244,704 (29.27) | 43,310 (29.30) | 201,394 (29.26) | |
| 200–240 | 141,449 (16.92) | 25,061 (16.95) | 116,388 (16.91) | |
| ≥240 | 60,733 (7.26) | 10,940 (7.40) | 49,793 (7.23) | |
| Unknown | 389,278 (46.56) | 68,513 (46.35) | 320,765 (46.60) | |
| Glomerular filtration rate, mL/min/1.73 m$^2$, n (%) | | | | 0.021 |
| <60 | 29,094 (3.48) | 5619 (3.80) | 23,475 (3.41) | |
| 60–90 | 372,253 (44.52) | 65,663 (44.42) | 306,590 (44.54) | |
| ≥90 | 429,662 (51.38) | 75,622 (51.16) | 354,040 (51.43) | |
| Unknown | 5155 (0.62) | 920 (0.62) | 4235 (0.62) | |
| Smoking status, n (%) | | | | 0.021 |
| Non-smoker | 530,925 (63.50) | 94,341 (63.82) | 436,584 (63.43) | |
| Ex-smoker | 154,348 (18.46) | 27,243 (18.43) | 127,105 (18.47) | |
| Current smoker | 150,393 (17.99) | 26,202 (17.73) | 124,191 (18.04) | |
| Unknown | 498 (0.06) | 38 (0.03) | 460 (0.07) | |

**Table 1 (continued) | Baseline characteristics for 1:5 propensity score-matched in main cohort (South Korea)**

| Covariates | 1:5 matched cohort (*n* = 836,164) | | | |
| --- | --- | --- | --- | --- |
| | Total | COVID-19 (*n* = 147,824) | Non-COVID-19 (*n* = 688,340) | SMD |
| Alcoholic drinks, days per week, *n* (%) | | | | 0.026 |
| <1 | 479,152 (57.30) | 85,120 (57.58) | 394,032 (57.24) | |
| 1–2 | 240,584 (28.77) | 42,049 (28.45) | 198,535 (28.84) | |
| 3–4 | 86,049 (10.29) | 15,085 (10.20) | 70,964 (10.31) | |
| ≥5 | 29,829 (3.57) | 5531 (3.74) | 24,298 (3.53) | |
| Unknown | 550 (0.07) | 39 (0.03) | 511 (0.07) | |
| Aerobic physical activity, *n* (%) | | | | 0.013 |
| Insufficient | 422,182 (50.49) | 74,978 (50.72) | 347,204 (50.44) | |
| Sufficient | 413,267 (49.42) | 72,760 (49.22) | 340,507 (49.47) | |
| Unknown | 715 (0.09) | 86 (0.06) | 629 (0.09) | |

*DBP* diastolic blood pressure, *SBP* systolic blood pressure, *SD* standard deviation, *SMD* standardized mean difference.

incident allergic outcomes using multinational databases in South Korea, Japan and the UK consisting of over 22 million participants in total. First, long-term risk of incident allergic diseases, especially asthma and AR, were significantly associated with SARS-CoV-2 infection, considering the follow up period. Second, the time attenuation effect on the risk of incident allergic diseases after SARS-CoV-2 infection was observed while its strength varies by country. These findings were consistent for all three different national cohorts, indicating the post-COVID-19 effect on allergic diseases regardless of ethnicity. Third, the greater severity of COVID-19 leads to a higher likelihood of developing allergic diseases including asthma and AR. Fourth, vaccination at least twice attenuated the risk at a level of the non-infected control, suggesting that COVID-19 vaccination may be an effective way to prevent post-COVID-19 outcomes.

## Comparisons with previous studies

Some studies highlighted worse COVID-19 outcomes in patients with allergy[7,13]; on the other hand, no studies, to date, have identified post-COVID-19 conditions on a comprehensive set of all allergic diseases including asthma, AR, AD and FA. Numerous case series and follow-up studies reported allergic/asthma-related symptoms, in a broad spectrum, such as shortness of breath, coughing and itchy skin[14,15]. However, these investigations were limited to non-peer-reviewed evidence, hospitalized patients, lack robustness due to uncontrolled potential confounding variables, and a small sample size. One recent study identified an elevated risk of new-onset AD after COVID-19[16]. Meanwhile, incident outcomes of diabetes[8], cardiovascular diseases[9], neurologic diseases[4], and dyslipidemia[10] following COVID-19 have been under investigation using nationwide cohorts. Thus, there is a need to investigate allergic sequelae of COVID-19 with multinational scale and population-based cohorts.

## Plausible mechanisms

The impacts of post-COVID-19 conditions on immune regulation have been under investigation[17], which may assist in understanding an increased risk of developing allergic diseases including asthma and AR after the acute phase of SARS-CoV-2 infection. First, disruptions in T cell homeostasis can result from post-COVID-19 conditions[18]. It is well-established that viral infections, in general, stimulate morphological alternations including tissue remodeling, and trigger immune responses, which contributes to the initiation of allergic diseases[19]. Moreover, regulatory T cells perturbation driven by post-COVID-19 conditions induces uninhibited action of effector cells and enables latent SARS-CoV-2[20], which may lead to post-acute sequelae of allergy. Also, a 'cytokine storm,' which is linked to the severe form of COVID-19, contributes to hyperinflammation and allergic sensitization that may be implicated in critical sequelae in respiratory tracts[21,22].

For the ameliorated effect of COVID-19 vaccination, previous studies are consistent with the observation in the current study[23,24]. It is suggested that the clearance of a viral reservoir, may be improved due to the adaptive immunity formed by additional dosage of vaccines. The notable number of the SARS-CoV-2 protein spike in patients with post-COVID-19 condition and the positive relationship between the number of protein spikes and post-COVID-19 symptoms supports this hypothesis[25]. Furthermore, we drew conclusions that double-vaccinated significantly ameliorates long-term sequelae of allergic diseases to the level of the non-COVID-19-infected controls. Previous studies showed similar results with two-dose vaccines and overall incidence of post-COVID-19 conditions compared to unvaccinated controls[26].

## Limitations and strengths

The current study has several strengths. First, we built the main cohort in South Korea comprising over 10 million participants and validated the findings with two different nationwide cohorts from replication cohort A (Japan) and B (UK) consisting of over 12 million individuals in total. Second, we increased assurance of the findings by performing exposure-driven propensity score matching in each cohort. Third, we controlled the confounding effects of numerous variables including health conditions, economic status, behaviors (smoking, alcohol drinks, and physical activity) and ethnicity. Fourth, the main cohort, which includes COVID-19 vaccination data, enables us to explore the association between the COVID-19 vaccine and allergic outcomes following SARS-CoV-2 infection. Fifth, ethnic diversity in the replication cohort B (UK) enhanced the robustness of the study, given that the symptoms and manifestations of post-COVID-19 conditions vary by ethnicity[6,12].

On the other hand, this study has some limitations as follows. First, allergy is a disease that the recognition and diagnosis distinctly reflect cultural and ethnic contexts. In addition, the 'hygiene hypothesis' suggests the incident allergic disorder is linked to exposure to microbes, size of family, and hygiene standards[27,28]. Although all diagnoses of allergic outcomes were based on the same International Classification of Diseases 10th (ICD-10) codes, we observed consistently and remarkably higher incidence rates of allergic diseases in Japan than those of the others. Second, the present study defined disease according to ICD-10 codes; thus, the findings should be interpreted with caution[29,30]. The potential misclassification of dyspnea as asthma, particularly due to the diagnostic complexities introduced by COVID-19, represents a limitation. Also, although we excluded individuals who had been diagnosed with asthma to focus on the development of asthma, there may be some people with pre-existing asthma but undiagnosed in the baseline period. Therefore, the potential for disease misclassification necessitates a cautious

**Table 2 | The HR with 95% CI for the long-term sequelae risk of incident allergic diseases following COVID-19 diagnosis of patients in the propensity score-matched cohorts in main cohort (South Korea), replication cohort A (Japan), and replication cohort B (UK)**

| Cohort | Exposure | Incidence rate[a] | HR (95% CI) | |
|---|---|---|---|---|
| | | | Model 1[b] | Model 2[c] |
| **Allergic diseases** | | | | |
| Main cohort | None | 24.0 | 1.00 (reference) | 1.00 (reference) |
| Main cohort | Patients with COVID-19 | 28.7 | **1.20 (1.13–1.27)** | **1.20 (1.13–1.27)** |
| Replication cohort A | None | 27.4 | 1.00 (reference) | 1.00 (reference) |
| Replication cohort A | Patients with COVID-19 | 75.1 | **2.65 (2.61–2.69)** | **2.56 (2.52–2.59)** |
| Replication cohort B | None | 7.2 | 1.00 (reference) | 1.00 (reference) |
| Replication cohort B | Patients with COVID-19 | 8.3 | **1.14 (1.06–1.22)** | **1.12 (1.04–1.20)** |
| **Asthma** | | | | |
| Main cohort | None | 1.0 | 1.00 (reference) | 1.00 (reference) |
| Main cohort | Patients with COVID-19 | 2.2 | **2.27 (1.81–2.85)** | **2.25 (1.80–2.83)** |
| Replication cohort A | None | 2.7 | 1.00 (reference) | 1.00 (reference) |
| Replication cohort A | Patients with COVID-19 | 7.6 | **2.77 (2.65–2.90)** | **2.63 (2.51–2.75)** |
| Replication cohort B | None | 3.2 | 1.00 (reference) | 1.00 (reference) |
| Replication cohort B | Patients with COVID-19 | 3.8 | **1.16 (1.05–1.29)** | **1.14 (1.03–1.27)** |
| **Allergic rhinitis** | | | | |
| Main cohort | None | 17.3 | 1.00 (reference) | 1.00 (reference) |
| Main cohort | Patients with COVID-19 | 21.3 | **1.23 (1.15–1.32)** | **1.23 (1.15–1.32)** |
| Replication cohort A | None | 20.4 | 1.00 (reference) | 1.00 (reference) |
| Replication cohort A | Patients with COVID-19 | 62.6 | **2.98 (2.93–3.03)** | **2.88 (2.83–2.93)** |
| Replication cohort B | None | 1.1 | 1.00 (reference) | 1.00 (reference) |
| Replication cohort B | Patients with COVID-19 | 1.4 | **1.21 (1.02–1.44)** | **1.20 (1.01–1.43)** |
| **Atopic dermatitis** | | | | |
| Main cohort | None | 2.6 | 1.00 (reference) | 1.00 (reference) |
| Main cohort | Patients with COVID-19 | 3.0 | 1.15 (0.96–1.37) | 1.15 (0.96–1.37) |
| Replication cohort A | None | 5.2 | 1.00 (reference) | 1.00 (reference) |
| Replication cohort A | Patients with COVID-19 | 6.8 | **1.25 (1.20–1.30)** | **1.21 (1.16–1.26)** |
| Replication cohort B | None | 0.04 | 1.00 (reference) | 1.00 (reference) |
| Replication cohort B | Patients with COVID-19 | 0.05 | 1.18 (0.46–2.99) | 1.19 (0.47–3.01) |
| **Food allergy** | | | | |
| Main cohort | None | 3.6 | 1.00 (reference) | 1.00 (reference) |
| Main cohort | Patients with COVID-19 | 3.0 | 0.85 (0.71–1.01) | 0.85 (0.71–1.00) |
| Replication cohort A | None | 1.3 | 1.00 (reference) | 1.00 (reference) |
| Replication cohort A | Patients with COVID-19 | 2.7 | **2.10 (1.95–2.25)** | **1.84 (1.71–1.98)** |
| Replication cohort B | None | 3.1 | 1.00 (reference) | 1.00 (reference) |
| Replication cohort B | Patients with COVID-19 | 3.4 | 1.09 (0.97–1.21) | 1.07 (0.96–1.20) |

The data in bold indicate significant differences ($P < 0.05$).

*BMI* body mass index, *CI* confidence interval, *HR* hazard ratio.

[a]Incidence rate is expressed as per 1000 person-years.

[b]*Model 1*: Adjusted for age (20–39, 40–59, and ≥60 years) and sex.

[c]*Model 2 (main cohort)*: Adjusted for age (20–39, 40–59, and ≥60 years); sex; household income (low income, middle income, and high income); region of residence (urban and rural); Charlson comorbidity index (0, 1, and ≥2); BMI (underweight [<18.5 kg/m$^2$], normal [18.5–23.0 kg/m$^2$], overweight [23.0–25.0 kg/m$^2$], obese [≥25.0 kg/m$^2$], and unknown); blood pressure (systolic blood pressure <140 mmHg and diastolic blood pressure <90 mmHg, systolic blood pressure ≥140 mmHg or diastolic blood pressure ≥90 mmHg, and unknown); fasting blood glucose (<100, ≥100 mg/dL, and unknown); serum total cholesterol (<200, 200–240, ≥240 mg/dL, and unknown); glomerular filtration rate (<60, 60–90, ≥90 mL/min/1.73 m$^2$, and unknown); smoking status (non-, ex-, current smoker, and unknown); alcoholic drinks (<1, 1–2, 3–4, ≥5 days per week, and unknown); aerobic physical activity (sufficient, insufficient, and unknown); previous history of cardiovascular disease, chronic kidney disease, and chronic obstructive pulmonary disease; history of medication use for diabetes mellitus, dyslipidemia, and hypertension; and missing indicators (BMI missing indicator [yes or no], blood pressure missing indicator [yes or no], fasting blood glucose missing indicator [yes or no], serum total cholesterol missing indicator [yes or no], glomerular filtration rate missing indicator [yes or no], smoking status missing indicator [yes or no], alcoholic drinks missing indicator [yes or no], and aerobic physical activity missing indicator [yes or no]).

[c]*Model 2 (replication cohort A)*: Adjusted for age (20–39, 40–59, and ≥60 years); sex; Charlson comorbidity index (0, 1, and ≥2); BMI (underweight [<18.5 kg/m$^2$], normal [18.5–23.0 kg/m$^2$], overweight [23.0–25.0 kg/m$^2$], obese [≥25.0 kg/m$^2$], and unknown); blood pressure (systolic blood pressure <140 mmHg and diastolic blood pressure <90 mmHg, systolic blood pressure ≥140 mmHg or diastolic blood pressure ≥90 mmHg, and unknown); fasting blood glucose (<100, ≥100 mg/dL, and unknown); serum total cholesterol (<200, 200–240, ≥240 mg/dL, and unknown); glomerular filtration rate (<60, 60–90, ≥90 mL/min/1.73 m$^2$, and unknown); smoking status (non- and current smoker, and unknown); alcoholic drinks (<1, 1–2, 3–4, ≥5 days per week, and unknown); aerobic physical activity (sufficient, insufficient, and unknown); previous history of cardiovascular disease, chronic kidney disease, and chronic obstructive pulmonary disease; history of medication use for diabetes mellitus, dyslipidemia, and hypertension; and missing indicators (BMI missing indicator [yes or no], blood pressure missing indicator [yes or no], fasting blood glucose missing indicator [yes or no], serum total cholesterol missing indicator [yes or no], glomerular filtration rate missing indicator [yes or no], smoking status missing indicator [yes or no], alcoholic drinks missing indicator [yes or no], and aerobic physical activity missing indicator [yes or no]).

[c]*Model 2 (replication cohort B)*: Adjusted for age (20–39, 40–59, and ≥ 60 years); sex; household income (<£18,000, £18,000–£30,999, £31,000–£51,999, £52,000–£100,000, >£100,000), and unknown; region of residence (urban and rural); townsend deprivation index (T1 [last deprived], T2, T3 [most deprived], and unknown); ethnicity (white, mixed, Asian, black, others, and unknown); Charlson comorbidity index (0, 1, and ≥ 2); BMI (normal [<25.0 kg/m$^2$], overweight [25.0–30.0 kg/m$^2$], obese [≥ 30.0 kg/m$^2$], and unknown); education levels (≤10, 11–12, >12, and unknown); blood pressure (systolic blood pressure <140 mmHg and diastolic blood pressure <90 mmHg, systolic blood pressure ≥140 mmHg or diastolic blood pressure ≥ 90 mmHg, and unknown); fasting blood glucose (<100, ≥100 mg/dL, and unknown); smoking status (non- and current smoker, and unknown); alcohol consumption (every day, sometimes, rarely days per week, and unknown); aerobic physical activity (low, moderate, high, and unknown); previous history of cardiovascular disease, chronic kidney disease, and chronic obstructive pulmonary disease; history of medication use for diabetes mellitus, dyslipidemia, and hypertension; and missing indicators (household income missing indicator [yes or no], townsend deprivation index missing indicator [yes or no], ethnicity missing indicator [yes or no], education levels missing indicator [yes or no], obesity missing indicator [yes or no], blood pressure missing indicator [yes or no], fasting blood glucose missing indicator [yes or no], serum total cholesterol missing indicator [yes or no], glomerular filtration rate missing indicator [yes or no], smoking status missing indicator [yes or no], alcoholic drinks missing indicator [yes or no], and aerobic physical activity missing indicator [yes or no]).

**Table 3 | Time attenuation effect on the development of allergic diseases after SARS-CoV-2 infection (model 2; adjusted HR with 95% CI)**

|  | HR (95% CI) | |
|---|---|---|
|  | Main cohort (South Korea) | Replication cohort A (Japan) |
| Allergic diseases | | |
| <3 months | **1.42 (1.29–1.56)** | **3.30 (3.24–3.36)** |
| 3–6 months | **1.14 (1.01–1.29)** | **1.77 (1.70–1.84)** |
| ≥6 months | 1.00 (0.91–1.11) | **1.61 (1.56–1.67)** |

The data in bold indicate significant differences ($P < 0.05$).

*CI* confidence interval, *HR* hazard ratio, *SARS-CoV-2* severe acute respiratory syndrome coronavirus.

*Model 2 (main cohort)*: Adjusted for age (20–39, 40–59, and ≥60 years); sex; household income (low income, middle income, and high income); region of residence (urban and rural); Charlson comorbidity index (0, 1, and ≥2); BMI (underweight [<18.5 kg/m$^2$], normal [18.5–23.0 kg/m$^2$], overweight [23.0–25.0 kg/m$^2$], obese [≥25.0 kg/m$^2$], and unknown); blood pressure (systolic blood pressure <140 mmHg and diastolic blood pressure <90 mmHg, systolic blood pressure ≥140 mmHg or diastolic blood pressure ≥90 mmHg, and unknown); fasting blood glucose (<100, ≥100 mg/dL, and unknown); serum total cholesterol (<200, 200–240, ≥240 mg/dL, and unknown); glomerular filtration rate (<60, 60–90, ≥90 mL/min/1.73 m$^2$, and unknown); smoking status (non-, ex-, current smoker, and unknown); alcoholic drinks (<1, 1–2, 3–4, ≥5 days per week, and unknown); aerobic physical activity (sufficient, insufficient, and unknown); previous history of cardiovascular disease, chronic kidney disease, and chronic obstructive pulmonary disease; history of medication use for diabetes mellitus, dyslipidemia, and hypertension; and missing indicators (BMI missing indicator [yes or no], blood pressure missing indicator [yes or no], fasting blood glucose missing indicator [yes or no], serum total cholesterol missing indicator [yes or no], glomerular filtration rate missing indicator [yes or no], smoking status missing indicator [yes or no], alcoholic drinks missing indicator [yes or no], and aerobic physical activity missing indicator [yes or no]).

*Model 2 (replication cohort A)*: Adjusted for age (20–39, 40–59, and ≥60 years); sex; Charlson comorbidity index (0, 1, and ≥ 2); BMI (underweight [<18.5 kg/m$^2$], normal [18.5–23.0 kg/m$^2$], overweight [23.0–25.0 kg/m$^2$], obese [≥25.0 kg/m$^2$], and unknown); blood pressure (systolic blood pressure <140 mmHg and diastolic blood pressure <90 mmHg, systolic blood pressure ≥140 mmHg or diastolic blood pressure ≥90 mmHg, and unknown); fasting blood glucose (<100, ≥100 mg/dL, and unknown); serum total cholesterol (<200, 200–240, ≥240 mg/dL, and unknown); glomerular filtration rate (<60, 60–90, ≥90 mL/min/1.73 m$^2$, and unknown); smoking status (non- and current smoker, and unknown); alcoholic drinks (<1, 1–2, 3–4, ≥5 days per week, and unknown); aerobic physical activity (sufficient, insufficient, and unknown); previous history of cardiovascular disease, chronic kidney disease, and chronic obstructive pulmonary disease; history of medication use for diabetes mellitus, dyslipidemia, and hypertension; and missing indicators (BMI missing indicator [yes or no], blood pressure missing indicator [yes or no], fasting blood glucose missing indicator [yes or no], serum total cholesterol missing indicator [yes or no], glomerular filtration rate missing indicator [yes or no], smoking status missing indicator [yes or no], alcoholic drinks missing indicator [yes or no], and aerobic physical activity missing indicator [yes or no]).

interpretation of the data. Third, information on COVID-19 vaccination status was not included in the replication cohorts A and B, we could not perform an analysis of validation for the influences of COVID-19 vaccine on the allergic outcomes. Fourth, although we adjusted for a large number of covariates, there are residual potential confounders such as asymptomatic SARS-CoV-2 infections[31]. Fifth, the current study is limited to the adult population; therefore, there is a need for a future study on the children population. Sixth, our current data set limits our ability to consider genetic factors related to parents' allergic diseases. However, in adults, the indications of a clear genetic predisposition to allergic conditions may not be as evident. Seventh, we did not take the previous history of severe acute respiratory syndrome and Middle East respiratory syndrome epidemics in South Korea, Japan, and the UK into consideration, which may serve as a potential confounder. Eighth, we used three cohorts with different reporting formats (self-report for the UKB cohort and insurance claims for the K-COV-N and JMDC cohorts) and construction of dataset. However, the results were aligned with one another, which rather strengthens the robustness of the study. Ninth, we conducted additional sensitivity analyses to capture mild cases of COVID-19 as comprehensively as possible (Tables S21 and S22). However, the potential exclusion of milder cases still exists. Additionally, our data may be biased due to different treatment methods for patients with COVID-19 based on the severity of

their illness. Tenth, all asymptomatic cases may not be identified in the cohorts in spite of the dedication of governments to reducing misdiagnosis. Finally, it is not assured that allergic outcomes followed COVID-19 exclusively. Though we executed a comparison analysis with contemporary controls of those who have not been infected with SARS-CoV-2 as previous studies did[4,8,32], further research is required using other infections as a comparator to strengthen the findings in this study.

## Clinical and policy implications

The current study shows the risk of incident allergic diseases increased in the post-acute phase of COVID-19. This finding addressed a need for persistent health policies to manage the severity of SARS-CoV-2 infection, which is an efficient way to occlude post-COVID-19 conditions. As struggling with the ongoing pandemic, governments should be prepared to deal with long-lasting allergic consequences following COVID-19 in the post-COVID era. Allergic diseases are common chronic diseases[33], Early detection is required unless they may turn to aggravated, life-risking forms. We further found that vaccination reduced the risk of post-COVID-19 effects of allergic diseases, advocating for a vaccine uptake as a mechanism to prevent post-COVID-19 conditions.

In conclusion, this study addresses a consistent and significant increased risk of new-onset of allergic diseases in people with previous COVID-19 diagnosis using multinational scale cohorts in South Korea, Japan and the UK. The time attenuation effect on the risk of incident allergic diseases after SARS-CoV-2 infection was observed while its strength varies by country. The greater severity of COVID-19 leads to a higher likelihood of developing allergic diseases. The risk gradually reduced over time while COVID-19 vaccination showed a protective effect against incident allergic diseases following SARS-CoV-2 infection. It is encouraged for survivors of COVID-19 to be aware of the manifestations of allergic diseases.

## Methods
### Data source
The Kyung Hee University (KHUH 2022-06-042), the Korea Disease Control and Prevention Agency (KDCA), the National Health Insurance Service (NHIS; KDCA-NHIS-2022-1-632) of South Korea, JMDC (PHP-00002201-04), and UKB (94075) approved the study protocol.

Written informed consent was obtained from all participants at enrollment. We used three large-scale, nationwide and population-based cohort designs in this study: a South Korean nationwide cohort (K-COV-N cohort [main cohort]; total $n = 10,027,506$), a Japanese claims-based cohort (JMDC cohort [replication cohort A]; total $n = 12,218,680$) and a UK prospective cohort from the UK Biobank (UKB cohort [replication cohort B]; total $n = 468,617$). Both the K-COV-N and JMDC cohorts employ a universal health insurance system. The UKB, meanwhile, is a dataset comprised of voluntary participation, including biomedical samples and health information. Detailed explanations of the JMDC and UKB cohorts can be found in supplemental material section.

### K-COV-N cohort (main)
The K-COV-N cohort is a large-scale, nationwide, general population-based cohort in South Korea, covering 98% of the South Korean population[34]. The cohort was developed and provided by the NHIS of South Korea and KDCA focused on individuals aged ≥20 years between January 1, 2018, and December 31, 2021. It contained information on COVID-19 vaccination, SARS-CoV-2 test results, COVID-19-related outcomes, results of national health examination, death records, and health insurance data including outpatient and inpatient information. The following characteristics of the Korean database enable us to construct a well-designed cohort: (1) A comprehensive healthcare system, implemented by the Korean government, covers people who have been infected with SARS-CoV-2; (2) all information was

**Table 4 | Propensity-score-matched subgroup analysis of HR (95% CI) of allergic diseases following COVID-19 diagnosis stratified by COVID-19 severity, SARS-CoV-2 strain type, and number of vaccinations in main cohort (South Korea)**

| Factors | Group | Exposure | Events/total number (%) | HR (95% CI) | |
|---|---|---|---|---|---|
| | | | | Model 1[b] | Model 2[c] |
| COVID-19 severity | Total | Non-infected control | 5633/688,340 (0.82) | 1.00 (reference) | 1.00 (reference) |
| | | Mild COVID-19 | 1125/127,641 (0.88) | **1.13 (1.06–1.20)** | **1.14 (1.07–1.21)** |
| | | Moderate to severe COVID-19 | 293/20,183 (1.45) | **1.55 (1.38–1.74)** | **1.48 (1.31–1.66)** |
| Strain type (original) | Total | Non-infected control at the same index date[a] | 4174/220,904 (1.89) | 1.00 (reference) | 1.00 (reference) |
| | | Original SARS-CoV-2 infection | 1050/46,900 (2.24) | **1.20 (1.12–1.29)** | **1.20 (1.12–1.29)** |
| Strain type (delta) | Total | Non-infected control at the same index date[a] | 1459/467,436 (0.31) | 1.00 (reference) | 1.00 (reference) |
| | | Delta SARS-CoV-2 infection | 368/100,924 (0.36) | **1.18 (1.05–1.32)** | **1.18 (1.05–1.32)** |
| Number of SARS-CoV-2 vaccinations | Patients with COVID-19 | Non-infected control | 5633/688,340 (0.82) | 1.00 (reference) | 1.00 (reference) |
| | | Without vaccination | 1133/68,456 (1.66) | **1.24 (1.16–1.32)** | **1.24 (1.16–1.32)** |
| | | Vaccination 1 time | 148/14,125 (1.05) | **1.43 (1.22–1.69)** | **1.44 (1.22–1.69)** |
| | | Vaccination ≥2 times | 137/65,243 (0.21) | **0.81 (0.68–0.96)** | **0.81 (0.68–0.96)** |

The data in bold indicate significant differences ($P < 0.05$).

*HR* hazard ratio, *CI* confidence interval, *SARS-CoV-2* severe acute respiratory syndrome coronavirus.

[a]Comparators defined only 1:5 matched comparators in each patient group at the same index date to reduce immortal bias.

[b]*Model 1*: Adjusted for age (20–39, 40–59, and ≥60 years) and sex.

[c]*Model 2 (main cohort)*: Adjusted for age (20–39, 40–59, and ≥60 years); sex; household income (low income, middle income, and high income); region of residence (urban and rural); Charlson comorbidity index (0, 1, and ≥2); BMI (underweight [<18.5 kg/m²], normal [18.5–23.0 kg/m²], overweight [23.0–25.0 kg/m²], obese [≥25.0 kg/m²], and unknown); blood pressure (systolic blood pressure <140 mmHg and diastolic blood pressure <90 mmHg, systolic blood pressure ≥140 mmHg or diastolic blood pressure ≥90 mmHg, and unknown); fasting blood glucose (<100, ≥100 mg/dL, and unknown); serum total cholesterol (<200, 200–240, ≥240 mg/dL, and unknown); glomerular filtration rate (<60, 60–90, ≥90 mL/min/1.73 m², and unknown); smoking status (non-, ex-, current smoker, and unknown); alcoholic drinks (<1, 1–2, 3–4, ≥5 days per week, and unknown); aerobic physical activity (sufficient, insufficient, and unknown); previous history of cardiovascular disease, chronic kidney disease, and chronic obstructive pulmonary disease; history of medication use for diabetes mellitus, dyslipidemia, and hypertension; and missing indicators (BMI missing indicator [yes or no], blood pressure missing indicator [yes or no], fasting blood glucose missing indicator [yes or no], serum total cholesterol missing indicator [yes or no], glomerular filtration rate missing indicator [yes or no], smoking status missing indicator [yes or no], alcoholic drinks missing indicator [yes or no], and aerobic physical activity missing indicator [yes or no]).

anonymized by the Korean government[34]; (3) It includes SARS-CoV-2 test results, vaccination status, and COVID-19-related hospital records; and (4) the overall predictive value for diagnostic records of the NHIS was 82% according to a previous study[6,35,36].

We included all individuals aged ≥20 years with COVID-19 and non-infected participants from 2020 to 2021 (total $n = 10,027,506$). We precluded those who meet the following criteria: (1) insufficient socioeconomic information or died before; and (2) history of allergic diseases in the pre-observation period, defined as two years ($n = 4,335,150$). Eventually, 5,692,356 individuals were included from South Korea in this study.

### Exposures and outcomes

The exposure was SARS-CoV-2 infection, which was defined if the participants tested positive for COVID-19 either by real-time reverse transcriptase polymerase chain reaction or rapid antigen testing of nasopharyngeal swabs. We considered the original SARS-CoV-2 if the initial infection was before July 31, 2021, and the delta variant was from August 1, 2021[37]. Patients who were admitted to an intensive care unit and those who required oxygen therapy, extracorporeal membrane oxygenation, renal replacement, or cardio resuscitation were perceived as having moderate to severe COVID-19[38]. The others were considered having mild COVID-19. The COVID-19 vaccination status was categorized according to dosage (unvaccinated, 1, and ≥2 times). Individuals who were vaccinated with the Johnson & Johnson/Janssen vaccine were considered twice vaccinated after the single dose.

The primary outcome was the onset of allergic diseases, including: asthma, AR, AD, and FA[7]. Also, the term 'allergic diseases' refers to a diagnosis of any of the following condition: asthma, AR, AD, or FA[39,40]. Allergic asthma was identified as asthma combined with an additional allergic disorder (AR, AD, or FA), while non-allergic asthma was classified as asthma occurring in the absence of any allergic diseases[7]. We defined patients with allergic diseases as those having at least two claims during the observation period and were taking relevant

medications. We provided a list of the ICD-10 codes and medications used to define each disease in this study (Table S1).

### Covariates

The demographic characteristics of the participants were obtained from the health insurance database as followings: sex, age (20–39, 40–59, and ≥60 years), household income (low [0–39 percentile], middle [40–79 percentile], and high [80–100 percentile]), and region of residence (urban and rural)[34]. The information on body mass index (underweight [<18.5 kg/m²], normal [18.5–23.0 kg/m²], overweight [23.0–25.0 kg/m²], obese [≥25.0 kg/m²], and unknown), blood pressure (systolic blood pressure <140 mmHg and diastolic blood pressure <90 mmHg, systolic blood pressure ≥140 mmHg or diastolic blood pressure ≥90 mmHg, and unknown), fasting blood glucose (<100, ≥100 mg/dL, and unknown), serum total cholesterol (<200, 200–240, ≥240 mg/dL, and unknown) and glomerular filtration rate (<60, 60–90, ≥90 mL/min/1.73 m², and unknown) were included from the fasting serum samples of national health examination[41]. The CCI, history of cardiovascular disease, chronic kidney disease, and chronic obstructive pulmonary disease, history of medication use for diabetes, hyperlipidemia, and hypertension, smoking status (non-, ex-, and current smoker), alcoholic drinks (<1, 1–2, 3–4, ≥5 days per week, and unknown), and aerobic physical activity (sufficient [≥600 Metabolic Equivalent Task scores], insufficient, and unknown) were collected based on ICD-10 code and/or results of national health examination[12,42]. Additionally, to minimize bias related to missing data, we focused on the missing indicator method, generating missing indicator variables and incorporating them into the adjustment variables[43].

### Propensity score matching

We executed 1:5 exposure-driven propensity score matching to balance the distribution of covariates in the two groups. We used a 'greedy nearest-neighbor' algorithm with random selection without replacement within caliper widths of 0.001 standard deviations[44,45]. We

assessed the adequacy of matching by comparing SMDs. A SMD < 0.1 indicated no major imbalance in the two groups[44,45]. We constructed the following covariates as matching variables for South Korea: age, sex, household income, region of residence, CCI, body mass index, blood pressure, fasting blood glucose, serum total cholesterol, glomerular filtration rate, smoking status, alcoholic drinks, aerobic physical activity, and history of medication use for diabetes mellitus, dyslipidemia, and hypertension. For the replication cohorts of Japan and the UK, we also used similar covariates as matching variables (Supplement Material). All covariates were regarded as adjustment variables in further statistical models. After propensity score matching, a total of 836,164 individuals were included in the study (Figure S1 and Table 1).

### JMDC cohort (replication A) and UKB cohort (replication B)

The same ICD-10 codes, definition of exposures and outcomes, observation period, and propensity score matching were utilized for the JMDC and the UKB cohorts as well (Supplement Material). Due to the absence of SARS-CoV-2 vaccination data[41], the JMDC and the UKB cohort were used only to validate the main findings of the K-COV-N cohort. After propensity score matching, the JMDC and the UKB cohorts consisted of 2,541,021 and 325,843 individuals, respectively (Figs. S2 and S3).

### Statistical analysis

As aforementioned, SARS-CoV-2 infection was defined as primary exposure and the incident allergic diseases after at least 30 days of infection was defined as the primary outcome in the general population-based cohorts of South Korea, Japan and the UK (Tables S2–S3). To overcome immortal time bias, the date of the first diagnosis of SARS-CoV-2 was perceived as the 'individual index date'. We considered 2018–2019 the pre-observation period to observe the history of medical diagnosis. The observation period of the Korean cohort was between January 1, 2020, and December 31, 2021. The follow-up ended on December 31, 2021, or upon the death of the subject (Fig. S4).

We performed 1:5 exposure-driven propensity matching in the nationwide cohorts of South Korea, Japan, and the UK (Table 1 and S4, S5). A Cox proportional hazard regression model with estimates of HRs and 95% CIs was used to explore incident overall and four subtypes (asthma, AR, AD, and FA) of allergic diseases associated with post-COVID-19 conditions[45]. We further assessed the time attenuation effect of allergic diseases following SARS-CoV-2 infection (<3, 3–6, and ≥6 months) to reduce reverse causation. This refers to the duration it took for patients infected with COVID-19 to be diagnosed with allergic diseases, and includes individuals who had not been diagnosed during the pre-observation period. We performed several subgroup analyses to the following parameters: severity of COVID-19 (mild and moderate to severe), strain type (original and delta), and dosage of SARS-CoV-2 vaccination (0, 1, and ≥2 times). In addition, we executed stratification analyses according to sex, age, household income, CCI, body mass index, alcohol drinking status, aerobic physical activity and strain type of SARS-CoV-2 (Tables S11–S20). We used SAS (version 9.4; SAS Institute Inc., Cary, NC, USA) to perform all statistical analyses in this study. A two-sided p-value less than 0.05 was considered statistically significant (Tables S23–S25).

### Sensitivity analysis

We conducted sensitivity analyses to assess the reliability of the findings from our primary analyses. First, to validate the study results and identify detection bias, we included tympanic membrane perforation disease as a negative control in our analyses for both the main and replication cohorts (Table S26)[46]. Second, to reduce misclassification bias due to dyspnea, we performed an analysis excluding symptoms of dyspnea in asthma cases. (Table S27). Third, we established a strict diagnostic criterion for asthma in the main cohort (Table S28). We conducted analyses on cases diagnosed with asthma, considering those with a history of emergency department visits or hospitalization[47]. Fourth, allergic asthma and non-allergic asthma were compared as distinct groups due to differences in the asthma phenotype (Table S29). Fifth, in order to examine the impact of COVID-19 severity on allergic diseases, the mild group and the moderate to severe group were analyzed as two separate cohorts (Tables S21 and S22). Sixth, we analyzed the onset of allergic diseases in relation to SARS-CoV-2 infection status among individuals with the same number of vaccine doses, for understanding the long-term immune protection provided by the COVID-19 vaccine and its effectiveness extent (Table S30). In the same context, we conducted a time attenuation analysis to identify potential impacts, including the decrease in immunity over time (Table S31).

### Patient and public involvement

In the case of the main cohort and replication cohort A, the outcome measures were determined independently, without any involvement from the participants. In contrast, for replication cohort B, the participants were directly involved in determining the outcome measures through a process of voluntary reporting. The study design and implementation were conducted without consultation. However, we plan to disseminate the results of this study to all study participants and wider relevant communities upon request.

### Reporting summary

Further information on research design is available in the Nature Portfolio Reporting Summary linked to this article.

## Data availability

The datasets analyzed during the current study are available in the National Health Insurance Service in South Korea (https://nhiss.nhis.or.kr/bd/ab/bdaba000eng.do), the JMDC in Japan (https://www.jmdc.co.jp/en/jmdc-claims-database/), and the UKB in the UK. This protects the confidentiality of the data and ensures that Information Governance is robust. Applications to access health data in South Korea are submitted to the National Health Insurance Service in South Korea. Information can be found at https://nhiss.nhis.or.kr/bd/ab/bdaba000eng.do. Applications to access health data in Japan are submitted to the JMDC, Japan. Information can be found at https://www.jmdc.co.jp/en/jmdc-claims-database/. Applications to access health data in the UK are submitted to the UKB. Information can be found at https://www.ukbiobank.ac.uk/.

## Code availability

This study did not generate new or customized code/algorithm. Statistical analyses were performed using SAS (version 9.4; SAS Institute Inc., Cary, NC, USA) for analysis of big data. The codes utilized in the analysis are available from the corresponding author.

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

## Acknowledgements

This study used the database of the Korea Disease Control and Prevention Agency (KDCA) and the National Health Insurance Service (NHIS) for policy and academic research. The research number of this study is KDCA-NHIS-2022-1-632. This research was supported by the National Research Foundation of Korea (NRF) grant funded by the Korea government (MSIT; RS-2023-00248157; D.K.Y.) and the grant of the Korea Health Technology R&D Project through the Korea Health Industry Development Institute (KHIDI), funded by the Ministry of Health & Welfare, Republic of Korea (HV22C0233; D.K.Y.). Hayeon Lee was supported by grant from the health fellowship foundation (2023; H.L.). The funders had no role in study design, data collection, data analysis, data interpretation, or writing of the report.

## Author contributions

D.K.Y. had full access to all of the data in the study and took responsibility for the integrity of the data and the accuracy of the data analysis. J.O., M.L., M.K., H.J.K., S.W.L., S.Y.R., A.K., L.S., M.S.K., H.L., J.L. and D.K.Y. approved the final version before submission. *Study concept and design*: J.O., M.L., M.K., H.J.K., H.L., J.L. and D.K.Y.; *Acquisition, analysis, or interpretation of data*: J.O., M.L., M.K., H.J.K., H.L., J.L. and D.K.Y.; *Drafting of the manuscript*: J.O., M.L., M.K., H.J.K., H.L., J.L. and D.K.Y.; *Critical revision of the manuscript for important intellectual content*: J.O., M.L., M.K., H.J.K., S.W.L., S.Y.R., A.K., L.S., M.S.K., H.L., J.L. and D.K.Y.; *Statistical analysis*: J.O., M.L., M.K., H.J.K., H.L., J.L. and D.K.Y.; *Study supervision*: D.K.Y. D.K.Y. is guarantor for this study. J.O., M.L., M.K. and H.J.K. were contributed equally as co-first authors. H.L., J.L. and D.K.Y. were contributed equally as co-corresponding authors. D.K.Y. is the senior author. The corresponding author attests that all listed authors meet authorship criteria and that no others meeting the criteria have been omitted.

## Competing interests

The authors declare no competing interests.

## Additional information

[1]Department of Medicine, Kyung Hee University College of Medicine, Seoul, South Korea. [2]Center for Digital Health, Medical Science Research Institute, Kyung Hee University College of Medicine, Seoul, South Korea. [3]Department of Regulatory Science, Kyung Hee University, Seoul, South Korea. [4]Department of Precision Medicine, Sungkyunkwan University School of Medicine, Suwon, South Korea. [5]Department of Endocrinology and Metabolism, Kyung Hee University School of Medicine, Seoul, South Korea. [6]Research and Development Unit, Parc Sanitari Sant Joan de Deu, Barcelona, Spain. [7]Centre for Health, Performance and Wellbeing, Anglia Ruskin University, Cambridge, UK. [8]Medical and Population Genetics and Cardiovascular Disease Initiative, Broad Institute of MIT and Harvard, Cambridge, MA, USA. [9]Department of Biomedical Engineering, Kyung Hee University, Yongin, South Korea. [10]Department of Electronics and Information Convergence Engineering, Kyung Hee University, Yongin, South Korea. [11]Department of Pediatrics, Kyung Hee University College of Medicine, Seoul, South Korea. [12]These authors contributed equally: Jiyeon Oh, Myeongcheol Lee, Minji Kim, Hyeon Jin Kim. [13]These authors jointly supervised this work: Hayeon Lee, Jinseok Lee, Dong Keon Yon. ✉e-mail: wwhy28@khu.ac.kr; gonasago@khu.ac.kr; yonkkang@gmail.com

