## [Peer Review File · Nature Communications]

Incident allergic diseases in post-COVID-19 condition: multinational cohort studies from South Korea, Japan and the UKEditorial Note: Parts of this Peer Review File have been redacted as indicated to remove third-party material where no permission to publish could be obtained.

REVIEWER COMMENTS

Reviewer #1 (Remarks to the Author):

The manuscript entitled "Risks of incident allergic diseases in long COVID: multinational population-based cohort studies from South Korea, Japan, and the UK" presents interesting and novel findings regarding the role of SARS-CoV-2 infection and long-COVID on the subsequent development of allergic diseases such as asthma, allergic rhinitis, atopic dermatitis, and food allergy. Based on analysis of three impressive cohorts from Korea, Japan, and the United Kingdom SARS-CoV-2 infection resulted in an elevated risk of allergic disease development. This effect was particularly strong for asthma and allergic rhinitis and was independent of the racial and ethnic diversity of the investigated cohorts. The highest risk of allergic disease development was observed beyond the 30 days after COVID-19 diagnosis, decreased over time, but remained persistent throughout follow-up. The risk of development of asthma and allergic rhinitis post-COVID was related to COVID-19 disease severity and vaccination status. Importantly, two or more vaccinations against COVID-19 demonstrated protective effects against the incidence of allergic diseases. To ensure the highest quality of the manuscript several comments should be taken into consideration:

1. What is the effect of two or more SARS-CoV-2 vaccinations in patients who were not infected with the SARS-CoV-2 virus? Is it possible to curate such a population from available cohorts? If understood correctly vaccinated and infected group of patients was compared to unvaccinated and not infected control group.
2. It would be very interesting to perform similar comparisons in children.
3. Did the asthma group consist of confirmed allergic asthma patients, or consist also of T2-low asthma patients? This should be investigated and compared in the manuscript.
4. The data clearly demonstrate that allergic burden in long-COVID applies only to respiratory system, but not the skin or GI tract. It would be very relevant to investigate parental history of allergic diseases as one of the criteria, which can influence allergic diseases development in presented cohorts.
5. In the context of newly developed allergic diseases. What are patients allergic to? Are those airborne, food, or other allergens?
6. What is the history of SARS-CoV and MERS epidemics in the geographic regions of the included cohort? Can the results of long-COVID be influenced by previous contact with other coronaviruses? Is it possible to get such information for included cohorts?
7. Can presented results be biased by different treatment regimens applied for moderate or severe COVID-19 patients in different countries?
8. Line 197. There is a typo in the COVID-19 word.

Reviewer #2 (Remarks to the Author):

The study uses large cohorts with clinical marker to investigate the onset of allergic diseases as a result of a SARS-CoV-2 Infection. One of the cohorts even has information on vaccination status. However, the absolute onset of allergic disease after the infection is low and the highest effect sizes are reported for asthma. I believe that the asthma diagnosis is likely caused by misclassification of dyspnea.

Approach related comments:

The study did not use negative control endpoints to rule out detection bias. Those with a SARS-CoV-2 infection might be under better surveillance, which results in an earlier detection of disease than in the control group due to higher healthcare utilization. Most effect estimates are below 30% risk increase/decrease. The exception is the JMDC cohort and asthma. Asthma could be misclassified by physicians as dyspnea. Dyspnea is one of the few symptoms, which made it into the WHO clinical case definition of post-COVID (for adults) in October 2021. This bias could be higher in Japan than South Korea rather than showing a much higher risk for developing asthma after the infection in Japan. The only way to rule it out would be to test how many onset asthma

patients have no incident diagnosis of dyspnea. Also, the analysis would benefit from a more specific definition (along the sensitive definition used) of the allergic disease with ICD-10 code and suitable medication to reduce misclassification bias for rare endpoints. Table 3 shows the time attenuation of the effect. In the K-CoV-N cohort the effect nearly disappeared over time, which raised the question if an onset of allergic diseases were observed or only flares of already existing disease (despite a 2-year baseline period).

The incidence in the non-COVID groups varies between the three cohorts. Especially UK Biobank reports a much higher onset of diseases. Due to an older population it should be less. This fact is likely due to different data collection (patient reported diseases vs. physician coded diseases). The Clinical Practice Research Datalink (CPRD) or THIN would be the comparably cohort for the United Kingdom.

The definition of "obese" is odd, as obese is defined by the WHO as an BMI above 30 not above 25. Also, overweight would be an BMI above 25 not above 23. This classification could likely result in residual confounding.

For the matching process the authors used age, sex, and region type. It is unclear why matching was not exact on these variable and if not why the introduction of other variable in the matching process would induce immortal time bias. "To reduce the immortal bias, variables requiring index date, such as previous history of diseases, are not included as matching covariates" If immortal time bias is an issue, this should also apply to adjustment of the variables. In so far it is not understandable why certain variables were selected in the matching process.

For unknown reason in K-CoV-N nearly all cases with unknown body mass index, blood pressure, fasting glucose, glomerular filtration rate, smoking status, alcoholic drinks, days per week and aerobic physical activity were found the COVID-19 group. In JMDC on the other site nearly all unknown information are in the control group.

Formal comments:

The authors uses „race and ethnicity" throughout the text. This is uncommon the should use one of these term.

The Figure S1 and Figure S2 are identical and UK Biobank is missing. If the selection was done the same way for all cohorts one figure would be enough.

At the first mention of the "pre-observation period" it should be written that the duration is two years.

On page 11 the reference to table S2 is wrong as the table has only the ICD-10 Code in it.

On page 12 the effect of AD was reported as insignificant, but the corresponding table 2 reported otherwise.

Unclear what "(n=3477)." should mean for reference 23

As there are not prognostic measures used, the term's discovery/validation are misplaced. It is advised to use replication cohort for the other cohorts or refer to each cohort as separate entities as K-CoV-N, JMDC and UK Biobank.

Reviewer #3 (Remarks to the Author):

This study presents interesting findings, with one of its primary strengths being the utilization of multiple extensive datasets. However, there are several critical aspects of the study that remain unaddressed, and addressing these would significantly enhance its depth and clarity.

First and foremost, it is important to consider whether the diseases under investigation (the primary outcomes) are accurately captured across all datasets. Given that the use of ICD-10 codes typically signifies diagnoses in hospital settings, it raises questions about the potential exclusion of milder cases that may have been diagnosed in primary care settings. This might introduce a bias, as those who are generally less unwell (and therefore less likely to experience severe COVID-19 outcomes) may also be less prone to severe allergic issues. Can the authors undertake any sensitivity analysis to rule this out or elaborate on this in the discussion?

Second, it's important to acknowledge that the datasets used in this study may exhibit substantial variation. For instance, the UK BioBank relies on voluntary participation, introducing potential selection bias that could impact the results. For readers who may not be familiar with these datasets, it would be highly beneficial if the authors could provide a clear exposition of their key characteristics and any fundamental distinctions between them. A pertinent question to consider is whether the other datasets also rely on voluntary participation. Providing such context would be invaluable.

Additionally, the paper requires further elaboration on testing strategies. The suggestion that asymptomatic cases might not be identified hints at the possibility that testing was predominantly limited to individuals exhibiting symptoms. This aspect merits more detailed exploration to better comprehend its implications.

Furthermore, the paper should address the extent of its population coverage and whether it encompasses a representative cross-section of individuals.

With regard to vaccination, it would be pertinent to examine the potential effects over time, including waning immunity.

In conclusion, while this paper holds promise, there are key dimensions that necessitate further clarification and consideration. Addressing these aspects would enhance the study's comprehensiveness and its ability to offer meaningful insights.

Point-to-point Response

Major changes and additions to the revised manuscript:

We conducted extensive revisions and provided a number of additional re-analyses.

1. To validate the study results and identify detection bias, we conducted an analysis including tympanic membrane perforation disease as a negative control group for each cohort.

2. In response to the feedback about misclassification bias in asthma, we performed two sensitivity analysis: asthma excluding dyspnea symptoms and asthma with using strict diagnostic criteria (history of emergency department visits or hospitalization). In addition, we redefined allergic diseases more accurately using ICD-10 codes and appropriate medications.

3. Taking into account the comments from the reviewer, we reconstructed the matching covariates to align the purpose of matching.

4. To address the concern of missing data between COVID-19 cases/controls in the different cohorts, we generated a missing indicator and included it as adjustment variable in each model.

5. Considering potential impact of diagnosed in primary care settings, we conducted additional analysis by dividing the group of mild and moderate to severe categories for COVID-19 severity.

6. Finally, we revised all manuscripts and tables to refer to the groups as the main cohort (South Korea), the replication cohort A (Japan), and the replication cohort B (UK) instead of the term

"discovery/validation" to reflect your comments.

The following table summarized the information about the revised manuscript versus the previous version.

	Original version	Revised version
Supplement Page	61p	97p
Supplement Figure	2	4
Supplement Table	15	28
Supplement material	0	1

Table of Contents

1. Reviewer #1	4
2. Reviewer #2.....	43
3. Reviewer #3.....	185

Reviewer #1**Comment 0.**

The manuscript entitled “Risks of incident allergic diseases in long COVID: multinational population-based cohort studies from South Korea, Japan, and the UK” presents interesting and novel findings regarding the role of SARS-COV-2 infection and long-COVID on the subsequent development of allergic diseases such as asthma, allergic rhinitis, atopic dermatitis, and food allergy. Based on analysis of three impressive cohorts from Korea, Japan, and the United Kingdom SARS-CoV-2 infection resulted in an elevated risk of allergic disease development. This effect was particularly strong for asthma and allergic rhinitis and was independent of the racial and ethnic diversity of the investigated cohorts. The highest risk of allergic disease development was observed beyond the 30 days after COVID-19 diagnosis, decreased over time, but remained persistent throughout follow-up. The risk of development of asthma and allergic rhinitis post-COVID was related to COVID-19 disease severity and vaccination status. Importantly, two or more vaccinations against COVID-19 demonstrated protective effects against the incidence of allergic diseases. To ensure the highest quality of the manuscript several comments should be taken into consideration:

Response:

Thank you for taking the time to review the manuscript entitled "Risks of incident allergic diseases in long COVID: multinational population-based cohort studies from South Korea, Japan, and the UK." Your insightful comments and suggestions are highly appreciated. We are pleased to hear that you found the findings presented in the manuscript to be interesting and novel, particularly about the impact of SARS-CoV-2 infection on the subsequent development of allergic diseases. We completely agree with your valuable suggestions for enhancing the quality of the manuscript. Strengthening the ‘Results’ section through additional statistical

analyses and clarification of certain ambiguous aspects will undoubtedly improve the reliability and validity of our findings. Furthermore, we appreciate your emphasis on the importance of clearly outlining the limitations of our study and discussing the implications of the results for future research directions. In addition, providing more context to ensure a clear and cohesive flow throughout the manuscript is a point well taken. We will aim to provide a more comprehensive explanation of the approaches and analyses employed in the methods section to further solidify the validity of our research. Lastly, we will ensure that the citations are thoroughly reviewed and updated to reflect the most recent and pertinent research, thereby reinforcing the overall credibility of the manuscript. We assure you that we will carefully consider each of your suggestions to improve the quality of the manuscript. Thank you once again for your time and valuable feedback.

Comment 1.

What is the effect of two or more SARS-CoV-2 vaccinations in patients who were not infected with the SARS-CoV-2 virus? Is it possible to curate such a population from available cohorts? If understood correctly vaccinated and infected group of patients was compared to unvaccinated and not infected control group.

Response:

Thank you for your insightful comment. Our primary analysis compared a group of individuals not infected with SARS-CoV-2 to those who were infected, regardless of their vaccination status. The infected group was further divided into subgroups: without vaccination, vaccination 1 time, and vaccination ≥ 2 times. Through this classification, we aimed to assess the risk and compare the allergic outcomes between those who were infected with COVID-19 and the others. Therefore, we did not analyze vaccinated individuals within the non-infected control group.

Factors	Group	Incidence rate*	HR (95% CI)		
			Model 1 ^a	Model 2 ^b	
Allergic diseases					
Number of SARS-CoV-2 vaccinations	Non-infected control without vaccination	15.5	1.00 (reference)	1.00 (reference)	
	Non-infected control with vaccination 1 time	29.8	1.92 (1.45 to 2.55)	1.91 (1.44 to 2.54)	
	Non-infected control with vaccination ≥ 2 times	24.5	1.55 (1.35 to 1.79)	1.57 (1.36 to 1.81)	

However, as you suggested, we conducted an additional analysis to examine the effect of the number of vaccine doses on allergic diseases within the non-infected group. In South Korea, priority for COVID-19 vaccination was given to individuals with lower immune function or at

high risk of severe illness and complications, which includes those who are more prone to visit healthcare settings such as allergy sufferers. This could imply a bias in the vaccinated group, potentially leading to a perceived higher association of allergic diseases in vaccinated individuals compared to the unvaccinated. This suggests a potential bias in vaccination, leading to the observation that vaccinated individuals may have a higher association with allergic diseases compared to those who are unvaccinated.

Therefore, we also aimed to investigate the onset of allergic diseases in relation to SARS-CoV-2 infection status among individuals with the same number of vaccine doses (Table S27). This approach is significant because it allows for a clearer understanding of the long-term immune protection provided by the COVID-19 vaccine and the extent of its effectiveness. This result indicates that receiving two doses of the vaccine strengthens the immune response and prolongs the duration of antibodies and memory immune cells, thereby reducing the risk of COVID-19 and long-COVID.¹ We hope our alternations are in line with your expectations.

<Reference>

1. Arunachalam PS, Scott MKD, Hagan T, Li C, Feng Y, Wimmers F, Grigoryan L, Trisal M, Edara VV, Lai L, Chang SE, Feng A, Dhingra S, Shah M, Lee AS, Chinthrajah S, Sindher SB, Mallajosyula V, Gao F, Sigal N, Kowli S, Gupta S, Pellegrini K, Tharp G, Maysel-Auslender S, Hamilton S, Aoued H, Hrusovsky K, Roskey M, Bosinger SE, Maecker HT, Boyd SD, Davis MM, Utz PJ, Suthar MS, Khatri P, Nadeau KC, Pulendran B. Systems vaccinology of the BNT162b2 mRNA vaccine in humans. *Nature*. 2021 Aug;596(7872):410-416. doi: 10.1038/s41586-021-03791-x. Epub 2021 Jul 12. PMID: 34252919; PMCID: PMC8761119.

Changes in text:

Methods, *Sensitivity analysis*

Sixth, we analyzed the onset of allergic diseases in relation to SARS-CoV-2 infection status among individuals with the same number of vaccine doses, for understanding the long-term immune protection provided by the COVID-19 vaccine and its effectiveness extent (**Table S27**).

Table S27. The HR with 95% CI for the long-term sequelae risk of incident allergic disease by vaccine dose following COVID-19 diagnosis in the propensity score-matched main cohort (South Korea).

Factors	Group	Incidence rate*	HR (95% CI)	
			Model 1 ^a	Model 2 ^b
Allergic disease				
Number of SARS-CoV-2 vaccinations	Non-infected control without vaccination	15.5	1.00 (reference)	1.00 (reference)
	Patients infected COVID-19 without vaccination	29.7	1.92 (1.66 to 2.22)	1.94 (1.67 to 2.24)
	Non-infected control with vaccination 1 time	29.8	1.00 (reference)	1.00 (reference)
	Patients infected COVID-19 with vaccination 1 time	35.6	1.14 (0.82 to 1.59)	1.16 (0.83 to 1.62)
	Non-infected control with vaccination ≥ 2 times	24.5	1.00 (reference)	1.00 (reference)
	Patients infected COVID-19 with vaccination ≥ 2 times	21.4	0.80 (0.64 to 1.00)	0.80 (0.64 to 1.00)
Asthma				
Number of SARS-CoV-2 vaccinations	Non-infected control without vaccination	0.6	1.00 (reference)	1.00 (reference)
	Patients infected COVID-19 without vaccination	2.1	3.18 (1.57 to 6.45)	3.40 (1.68 to 6.91)
	Non-infected control with vaccination 1 time	4.1	1.00 (reference)	1.00 (reference)
	Patients infected COVID-19 with vaccination 1 time	3.3	0.69 (0.20 to 2.32)	0.77 (0.23 to 2.60)
	Non-infected control with vaccination ≥ 2 times	1.0	1.00 (reference)	1.00 (reference)
	Patients infected COVID-19 with vaccination ≥ 2 times	1.8	1.63 (0.57 to 4.64)	1.62 (0.56 to 4.62)
Allergic rhinitis				

Number of SARS-CoV-2 vaccinations	Non-infected control without vaccination	10.5	1.00 (reference)	1.00 (reference)
	Patients infected COVID-19 without vaccination	22.0	2.14 (1.79 to 2.55)	2.15 (1.80 to 2.56)
	Non-infected control with vaccination 1 time	20.0	1.00 (reference)	1.00 (reference)
	Patients infected COVID-19 with vaccination 1 time	25.2	1.22 (0.82 to 1.82)	1.24 (0.83 to 1.85)
	Non-infected control with vaccination ≥ 2 times	17.7	1.00 (reference)	1.00 (reference)
	Patients infected COVID-19 with vaccination ≥ 2 times	16.4	0.86 (0.66 to 1.12)	0.85 (0.65 to 1.12)
Atopic dermatitis				
Number of SARS-CoV-2 vaccinations	Non-infected control without vaccination	2.3	1.00 (reference)	1.00 (reference)
	Patients infected COVID-19 without vaccination	3.2	1.47 (1.00 to 2.16)	1.49 (1.01 to 2.20)
	Non-infected control with vaccination 1 time	1.8	1.00 (reference)	1.00 (reference)
	Patients infected COVID-19 with vaccination 1 time	3.3	1.94 (0.61 to 6.17)	2.04 (0.64 to 6.45)
	Non-infected control with vaccination ≥ 2 times	2.6	1.00 (reference)	1.00 (reference)
	Patients infected COVID-19 with vaccination ≥ 2 times	1.7	0.68 (0.36 to 1.28)	0.67 (0.36 to 1.27)
Food allergy				
Number of SARS-CoV-2 vaccinations	Non-infected control without vaccination	2.4	1.00 (reference)	1.00 (reference)
	Patients infected COVID-19 without vaccination	3.3	1.25 (0.86 to 1.82)	1.26 (0.86 to 1.83)
	Non-infected control with vaccination 1 time	5.1	1.00 (reference)	1.00 (reference)

Patients infected COVID-19 with vaccination 1 time	4.1	0.69 (0.30 to 1.59)	0.69 (0.30 to 1.60)
Non-infected control with vaccination ≥ 2 times	3.7	1.00 (reference)	1.00 (reference)
Patients infected COVID-19 with vaccination ≥ 2 times	1.6	0.38 (0.20 to 0.71)	0.38 (0.20 to 0.71)

CI, confidence interval; HR, hazard ratio.

* Incidence rate is expressed as per 1,000 person-years.

^a **Model 1:** Adjusted for age (20–39, 40–59, and ≥ 60 years) and sex.

^b **Model 2 (South Korea):** Adjusted for age (20–39, 40–59, and ≥ 60 years); sex; household income (low income, middle income, and high income); region of residence (urban and rural); Charlson comorbidity index (0, 1, and ≥ 2); BMI (underweight [$< 18.5 \text{ kg/m}^2$], normal [$18.5\text{--}23.0 \text{ kg/m}^2$], overweight [$23.0\text{--}25.0 \text{ kg/m}^2$], obese [$\geq 25.0 \text{ kg/m}^2$], and unknown); blood pressure (systolic blood pressure $< 140 \text{ mmHg}$ and diastolic blood pressure $< 90 \text{ mmHg}$, systolic blood pressure $\geq 140 \text{ mmHg}$ or diastolic blood pressure $\geq 90 \text{ mmHg}$, and unknown); fasting blood glucose (< 100 , $\geq 100 \text{ mg/dL}$, and unknown); serum total cholesterol (< 200 , $200\text{--}240$, $\geq 240 \text{ mg/dL}$, and unknown); glomerular filtration rate (< 60 , $60\text{--}90$, $\geq 90 \text{ mL/min/1.73 m}^2$, and unknown); smoking status (non-, ex-, current smoker, and unknown); alcoholic drinks (< 1 , $1\text{--}2$, $3\text{--}4$, ≥ 5 days per week, and unknown); aerobic physical activity (sufficient, insufficient, and unknown); previous history of cardiovascular disease, chronic kidney disease, and chronic obstructive pulmonary disease; history of medication use for diabetes mellitus, dyslipidemia, and hypertension; and missing indicators (BMI missing indicator [yes or no], blood pressure missing indicator [yes or no], fasting blood glucose missing indicator [yes or no], serum total cholesterol missing indicator [yes or no], glomerular filtration rate missing indicator [yes or no], smoking status missing indicator [yes or no], alcoholic drinks missing indicator [yes or no], and aerobic physical activity missing indicator [yes or no]).

Comment 2.

It would be very interesting to perform similar comparisons in children.

Response:

We appreciate your suggestion regarding the comparison of children. The distinction in physiological and immunological responses between children and adults indeed makes the idea of conducting similar analyses in a pediatric population quite interesting. However, our current dataset is limited to adult populations. The dataset used in the analysis is a claims-based cohort focused on adults in South Korea and Japan, and the data from the UK also focus on adults aged 40 and above. Consequently, due to these dataset constraints, it is presently challenging to execute comparable studies in children. However, we have studied the association between COVID-19 and pediatric allergies in other study, specifically investigating the impact of maternal SARS-CoV-2 infection on childhood allergies.¹ This study used a birth cohort from the National Health Insurance Service. In the birth cohort, children were matched with their mothers using a unique registration number assigned to each individual.

- **Kim M**, Choi Y, Lee M, Kang J, Kang SM, Lee DG, **Yon DK***. Maternal SARS-CoV-2 infection during pregnancy and subsequent risk of atopic dermatitis in offspring: a nationwide birth cohort study in South Korea. **Br J Dermatol. 2023** Nov 30:ljad478. doi: 10.1093/bjd/ljad478. Epub ahead of print. PMID: 38035775.

Table 1. Adjusted hazard ratio and 95% CI for the association between prenatal SARS-CoV-2 infection during pregnancy and atopic dermatitis in childhood in the propensity score matching cohort analysis.

Parameter	n (%)	AD event (%)	Person-years	AD incidence rate*	Adjusted hazard ratio (95% CI)†
Overall					
Children without prenatal SARS-CoV-2 infection	12,695 (90.8)	2834 (22.3)	9662	29.3	1.0 (reference)
Children with prenatal SARS-CoV-2 infection	1286 (9.2)	225 (17.5)	621	36.2	1.19 (1.04 to 1.36)

AD, atopic dermatitis; CI, confidence interval; SARS-CoV-2, severe acute respiratory syndrome coronavirus 2.

* Atopic dermatitis incidence rate is expressed as per 100 person-years.

The findings indicated that children born to mothers with SARS-CoV-2 infection had a 19% increased risk of developing atopic dermatitis compared to children of mothers without SARS-CoV-2 infection. These results suggest that SARS-CoV-2 infection can also impact allergic outcomes in the pediatric population.

In another similar prior study, a meta-analysis was conducted to examine the prevalence of Long-COVID in children and adolescents.² Additionally, a different study systematically reviewed and evaluated the epidemiological evidence regarding allergic diseases as risk factors for Long-COVID.³ However, no study has extensively investigated the long-term effects of COVID-19 on all allergic conditions, including asthma, allergic rhinitis (AR), atopic dermatitis (AD), and food allergies (FA), across both adult and child populations. Owing to these factors, while your proposal is indeed of significant value, it regrettably falls outside the scope of the current research endeavors. Nevertheless, the subject you have suggested undoubtedly represents an engaging area for future study. We extend our gratitude for your suggestion.

<Reference>

1. Kim M, Choi Y, Lee M, Kang J, Kang SM, Lee DG, Yon DK. Maternal SARS-CoV-2 infection during pregnancy and subsequent risk of atopic dermatitis in offspring: a nationwide birth cohort study in South Korea. *Br J Dermatol*. 2023 Nov 30;ljad478. doi: 10.1093/bjd/ljad478. Epub ahead of print. PMID: 38035775.
2. Lopez-Leon S, Wegman-Ostrosky T, Ayuzo Del Valle NC, Perelman C, Sepulveda R, Rebolledo PA, Cuapio A, Villapol S. Long-COVID in children and adolescents: a systematic review and meta-analyses. *Sci Rep*. 2022 Jun 23;12(1):9950. doi: 10.1038/s41598-022-13495-5. PMID: 35739136; PMCID: PMC9226045.
3. Wolff D, Drewitz KP, Ulrich A, Siegels D, Deckert S, Sprenger AA, Kuper PR, Schmitt J, Munblit D, Apfelbacher C. Allergic diseases as risk factors for Long-COVID symptoms: Systematic review of prospective cohort studies. *Clin Exp Allergy*. 2023 Nov;53(11):1162-1176. doi: 10.1111/cea.14391. Epub 2023 Nov 8. PMID: 37936547.

Changes in text:

Discussion, *Limitations and strengths*

Fourth, the current study is limited to the adult population; therefore, there is a need for a future study on the children population.

Comment 3.

Did the asthma group consist of confirmed allergic asthma patients, or consist also of T2-low asthma patients? This should be investigated and compared in the manuscript.

Response:

Thank you for your critical comment. First time, defining asthma using only ICD-10 codes presented challenges in distinguishing between patients with allergic asthma and non-allergic asthma due to the lack of precise classification within the patients with asthma. However, if you are referring to a comparison between patients with allergic asthma and those with non-allergic asthma, it was feasible when we defined these groups as follows (Table S1). We classified allergic asthma as cases where a patient has both asthma and at least one of these conditions: atopic dermatitis, allergic rhinitis, or food allergy. Non-allergic asthma was identified as asthma occurring without any of these allergic conditions. We have conducted a comparison analysis between allergic and non-allergic asthma in our study (Table S24).

The risk of allergic asthma increased in South Korea (adjusted hazard ratio [aHR], 1.93; 95% CI, 1.23 to 3.01), Japan (aHR, 2.66; 95% CI, 2.50 to 2.84), and the UK (aHR, 0.97; 95% CI, 0.62 to 1.50). Similarly, for non-allergic asthma, an elevated risk has also been noted in South Korea (aHR, 2.38; 95% CI, 1.83 to 3.11), Japan (aHR, 2.60; 95% CI, 2.44 to 2.78), and the UK (aHR, 1.15; 95% CI, 1.04 to 1.28). The risk of non-allergic asthma is more likely to increase compared to that of allergic asthma. This may be attributed to fundamental pathophysiological differences between non-allergic and allergic asthma.

First, non-allergic asthma typically involves different inflammatory pathways compared to allergic asthma. Allergic asthma is primarily driven by Th2 (Type 2 helper T-cells) immune responses, which are often characterized by eosinophilic inflammation and high levels of IgE antibodies, typically triggered by allergens.¹⁻³ In contrast, non-allergic asthma is not

primarily driven by Th2 responses and may involve neutrophilic inflammation.¹⁻³ This difference in underlying inflammation could impact how the body responds to and recovers from viral infections like COVID-19. The inflammation associated with non-allergic asthma might not provide the same protective effects against severe COVID-19 as seen in allergic asthma.

Second, the expression of the ACE2 receptor, which the SARS-CoV-2 uses to enter cells, may vary between individuals with non-allergic and allergic asthma. Some studies suggest that allergic asthma, particularly in response to cytokines like IL-13, may lead to reduced expression of ACE2 receptors in the airways, potentially providing some protection against severe COVID-19.^{4,5} In contrast, non-allergic asthma might not exhibit this reduced ACE2 expression, making these individuals more susceptible to severe infection.

Third, non-allergic asthma often involves chronic inflammation and may be associated with immune system dysregulation.^{6,7} This ongoing inflammation and dysregulation could exacerbate the inflammatory response to COVID-19, leading to a higher risk of severe outcomes and prolonged recovery periods, as seen in long-COVID.

To summarize, the potential for prolonged COVID-related illness in individuals with non-allergic asthma may stem from differences in underlying inflammatory processes, the expression of viral receptors, and the overall function of the immune system, compared to those with allergic asthma. Incorporating this additional analysis deepened our understanding and enhanced the comprehensiveness of our manuscript. Once again, we express our gratitude for your feedback.

<Reference>

1. Eggert LE, He Z, Collins W, Lee AS, Dhondalay G, Jiang SY, Fitzpatrick J, Snow TT,

- Pinsky BA, Artandi M, Barman L, Puri R, Wittman R, Ahuja N, Blomkalns A, O'Hara R, Cao S, Desai M, Sindher SB, Nadeau K, Chinthrajah RS. Asthma phenotypes, associated comorbidities, and long-term symptoms in COVID-19. *Allergy*. 2022 Jan;77(1):173-185. doi: 10.1111/all.14972. Epub 2021 Jun 19. PMID: 34080210; PMCID: PMC8222896.
2. Han X, Krempski JW, Nadeau K. Advances and novel developments in mechanisms of allergic inflammation. *Allergy*. 2020 Dec;75(12):3100-3111. doi: 10.1111/all.14632. Epub 2020 Nov 4. PMID: 33068299.
 3. Baos S, Calzada D, Cremades-Jimeno L, Sastre J, Picado C, Quiralte J, Florido F, Lahoz C, Cárdbaba B. Nonallergic Asthma and Its Severity: Biomarkers for Its Discrimination in Peripheral Samples. *Front Immunol*. 2018 Jun 21;9:1416. doi: 10.3389/fimmu.2018.01416. PMID: 29977241; PMCID: PMC6021512.
 4. Jackson DJ, Busse WW, Bacharier LB, Kattan M, O'Connor GT, Wood RA, Visness CM, Durham SR, Larson D, Esnault S, Ober C, Gergen PJ, Becker P, Togias A, Gern JE, Altman MC. Association of respiratory allergy, asthma, and expression of the SARS-CoV-2 receptor ACE2. *J Allergy Clin Immunol*. 2020 Jul;146(1):203-206.e3. doi: 10.1016/j.jaci.2020.04.009. Epub 2020 Apr 22. PMID: 32333915; PMCID: PMC7175851.
 5. Chhapola Shukla S. ACE2 expression in allergic airway disease may decrease the risk and severity of COVID-19. *Eur Arch Otorhinolaryngol*. 2021 Jul;278(7):2637-2640. doi: 10.1007/s00405-020-06408-7. Epub 2020 Oct 6. PMID: 33025046; PMCID: PMC7538174.
 6. Zhu Z, Hasegawa K, Ma B, Fujiogi M, Camargo CA Jr, Liang L. Association of asthma and its genetic predisposition with the risk of severe COVID-19. *J Allergy Clin Immunol*. 2020 Aug;146(2):327-329.e4. doi: 10.1016/j.jaci.2020.06.001. Epub 2020 Jun 6. PMID: 32522462; PMCID: PMC7423602.

7. Murphy TR, Busse W, Holweg CTJ, Rajput Y, Raimundo K, Meyer CS, Seetasith A, Gupta S, Iqbal A, Kaner RJ. Patients with allergic asthma have lower risk of severe COVID-19 outcomes than patients with nonallergic asthma. *BMC Pulm Med.* 2022 Nov 14;22(1):418. doi: 10.1186/s12890-022-02230-5. PMID: 36376851; PMCID: PMC9660106.

Changes in text:

Methods, *Exposures and outcomes*

Allergic asthma was identified as asthma combined with an additional allergic disorder (AR, AD, or FA), while non-allergic asthma was classified as asthma occurring in the absence of any allergic diseases.

Methods, *Sensitivity analysis*

Fourth, allergic asthma and non-allergic asthma were compared as distinct groups due to differences in the asthma phenotype (**Table S24**).

Reference

6. Yang JM, *et al.* Allergic disorders and susceptibility to and severity of COVID-19: A nationwide cohort study. *The Journal of allergy and clinical immunology* **146**, 790-798 (2020).

Table S1. Definitions of diseases.

Diseases	ICD-10 codes
Asthma	J45 or J46 with ≥ 2 claims and the use of asthma related medications (inhaled corticosteroid and systemic corticosteroids, long- and short-acting $\beta 2$ -agonists, and/or leukotriene antagonists)
Allergic asthma	Asthma with at least one of the following conditions: allergic rhinitis, atopic dermatitis, or food allergy.
Non-allergic asthma	Asthma occurring without any of the following conditions: allergic rhinitis, atopic dermatitis, and food allergy.
Allergic rhinitis	J30.1, J30.2, J30.3, or J30.4 with ≥ 2 claims and the use of allergic rhinitis related medications (antihistamines, corticosteroid nasal sprays, leukotriene antagonists, and/or mast cell stabilizers).
Atopic dermatitis	L20 with ≥ 2 claims and the use of atopic dermatitis related medications (topical and systemic corticosteroids, calcineurin inhibitors, and systemic immunosuppressants [azathioprine, cyclosporine, mycophenolate mofetil, and/or methotrexate]).
Food allergy	K52.2, K52.3, K52.8, K52.9, L27.2, L23.6, T78.0, T78.1, or Z91.0 with ≥ 2 claims
History of cardiovascular disease	I00-I02, I05-I09, I50, I51, I52, or I20-I45
History of chronic kidney disease	E10.2, E11.2, E13.2, E14.2, I12.0, M20.0, M31.3, M31.9, M32.1B, N02-N08, N11, N12, N14, N15.8, N15.9, N16.0, N16.2, N16.3, N16.4, N16.8, N18, N19, N26, Q61.2, Q61.3, or Q61.5 with ≥ 2 claims
History of chronic obstructive pulmonary disease	J44.0, J44.1, or J44.9 with ≥ 2 claims
Dyspnea	R06.0 with ≥ 1 claims
Tympanic membrane perforation	H72 with ≥ 2 claims

ICD-10, International Classification of Diseases, 10th edition.

Table S24. The HR with 95% CI for the long-term sequelae risk of incident asthma phenotype (allergic asthma and non-allergic asthma) following COVID-19 diagnosis in the propensity score-matched main cohort (South Korea), replication cohort A (Japan), and replication cohort B (UK).

Cohort	Exposure	Incidence rate*	HR (95% CI)	
			Model 1 ^a	Model 2 ^b
Allergic asthma				
Main cohort	Non-infected	0.3	1.00 (reference)	1.00 (reference)
Main cohort	Patients with COVID-19	0.5	1.95 (1.24 to 3.04)	1.93 (1.23 to 3.01)
Replication cohort A	Non-infected	1.3	1.00 (reference)	1.00 (reference)
Replication cohort A	Patients with COVID-19	3.8	2.78 (2.61 to 2.97)	2.66 (2.50 to 2.84)
Replication cohort B	Non-infected	0.2	1.00 (reference)	1.00 (reference)
Replication cohort B	Patients with COVID-19	0.2	1.00 (0.64 to 1.55)	0.97 (0.62 to 1.50)
Non-allergic asthma				
Main cohort	Non-infected	0.7	1.00 (reference)	1.00 (reference)
Main cohort	Patients with COVID-19	1.7	2.40 (1.84 to 3.13)	2.38 (1.83 to 3.11)
Replication cohort A	Non-infected	1.4	1.00 (reference)	1.00 (reference)
Replication cohort A	Patients with COVID-19	3.8	2.76 (2.59 to 2.94)	2.60 (2.44 to 2.78)
Replication cohort B	Non-infected	3.0	1.00 (reference)	1.00 (reference)
Replication cohort B	Patients with COVID-19	3.6	1.17 (1.05 to 1.30)	1.15 (1.04 to 1.28)

CI, confidence interval; HR, hazard ratio.

* Incidence rate is expressed as per 1,000 person-years.

The data in bold indicate significant differences ($P < 0.05$).

^a **Model 1:** Adjusted for age (20–39, 40–59, and ≥60 years) and sex.

^b **Model 2 (main cohort):** Adjusted for age (20–39, 40–59, and ≥60 years); sex; household income (low income, middle income, and high income); region of residence (urban and rural); Charlson comorbidity index (0, 1, and ≥2); BMI (underweight [$<18.5 \text{ kg/m}^2$], normal [$18.5\text{--}23.0 \text{ kg/m}^2$], overweight [$23.0\text{--}25.0 \text{ kg/m}^2$], obese [$\geq 25.0 \text{ kg/m}^2$], and unknown); blood pressure (systolic blood pressure $<140 \text{ mmHg}$ and diastolic blood pressure $<90 \text{ mmHg}$, systolic blood pressure $\geq 140 \text{ mmHg}$ or diastolic blood pressure $\geq 90 \text{ mmHg}$, and unknown); fasting blood glucose (<100 , $\geq 100 \text{ mg/dL}$, and unknown); serum total cholesterol (<200 , $200\text{--}240$, $\geq 240 \text{ mg/dL}$, and unknown); glomerular filtration rate (<60 , $60\text{--}90$, $\geq 90 \text{ mL/min/1.73 m}^2$, and unknown); smoking status (non-, ex-, current smoker, and unknown); alcoholic drinks (<1 , $1\text{--}2$, $3\text{--}4$, ≥ 5 days per week, and unknown); aerobic physical activity (sufficient, insufficient, and unknown); previous history of cardiovascular disease, chronic kidney disease, and chronic obstructive pulmonary disease; history of medication use for diabetes mellitus, dyslipidemia, and hypertension; and missing indicators (BMI missing indicator [yes or no], blood pressure missing indicator [yes or no], fasting blood glucose missing indicator [yes or no], serum total cholesterol missing indicator [yes or no], glomerular filtration rate missing indicator [yes or no], smoking status missing indicator [yes or no], alcoholic drinks missing indicator [yes or no], and aerobic physical activity missing indicator [yes or no]).

^b **Model 2 (replication cohort A):** Adjusted for age (20–39, 40–59, and ≥60 years); sex; Charlson comorbidity index (0, 1, and ≥ 2); BMI (underweight [$<18.5 \text{ kg/m}^2$], normal [$18.5\text{--}23.0 \text{ kg/m}^2$], overweight [$23.0\text{--}25.0 \text{ kg/m}^2$], obese [$\geq 25.0 \text{ kg/m}^2$], and unknown); blood pressure (systolic blood pressure $<140 \text{ mmHg}$ and diastolic blood pressure $<90 \text{ mmHg}$, systolic blood pressure $\geq 140 \text{ mmHg}$ or diastolic blood pressure $\geq 90 \text{ mmHg}$, and unknown); fasting blood glucose (<100 , $\geq 100 \text{ mg/dL}$, and unknown); serum total cholesterol (<200 , $200\text{--}240$, $\geq 240 \text{ mg/dL}$, and unknown); glomerular filtration rate (<60 , $60\text{--}90$, $\geq 90 \text{ mL/min/1.73 m}^2$, and unknown); smoking status (non- and current smoker, and unknown); alcoholic drinks (drinks; <1 , $1\text{--}2$, $3\text{--}4$, ≥ 5 days per week, and unknown); aerobic physical activity (sufficient, insufficient, and unknown); previous history of cardiovascular disease, chronic kidney disease, and chronic obstructive pulmonary disease; history of medication use for diabetes mellitus, dyslipidemia, and hypertension; and missing indicators (BMI missing indicator [yes or no], blood pressure missing indicator [yes or no], fasting blood glucose missing indicator [yes or no], serum total cholesterol missing indicator [yes or no], glomerular filtration rate missing indicator [yes or no], smoking status missing indicator [yes or no], alcoholic drinks missing indicator [yes or no], and aerobic physical activity missing indicator [yes or no]).

^b Model 2 (replication cohort B): Adjusted for age (20–39, 40–59, and ≥ 60 years); sex; household income (<£18,000, £18,000–£30,999, £31,000–£51,999, £52,000–£100,000, >£100,000), and unknown); region of residence (urban and rural); townsend deprivation index (T1[least deprived], T2, T3 [most deprived], and unknown); race (white, mixed, Asian, black, others, and unknown); Charlson comorbidity index (0, 1, and ≥ 2); BMI (normal [$<25.0 \text{ kg/m}^2$], overweight [$25.0\text{--}30.0 \text{ kg/m}^2$], obese [$\geq 30.0 \text{ kg/m}^2$], and unknown); education levels (≤ 10 , 11-12, >12 , and unknown); blood pressure (systolic blood pressure $< 140 \text{ mmHg}$ and diastolic blood pressure $< 90 \text{ mmHg}$, systolic blood pressure $\geq 140 \text{ mmHg}$ or diastolic blood pressure $\geq 90 \text{ mmHg}$, and unknown); fasting blood glucose (<100 , $\geq 100 \text{ mg/dL}$, and unknown); smoking status (non- and current smoker, and unknown); alcohol consumption (every day, sometimes, rarely days per week, and unknown); aerobic physical activity (low, moderate, high, and unknown); previous history of cardiovascular disease, chronic kidney disease, and chronic obstructive pulmonary disease; history of medication use for diabetes mellitus, dyslipidemia, and hypertension; and missing indicators (household income missing indicator [yes or no], townsend deprivation index missing indicator [yes or no], race missing indicator [yes or no], education levels [yes or no], obesity missing indicator [yes or no], blood pressure missing indicator [yes or no], fasting blood glucose missing indicator [yes or no], serum total cholesterol missing indicator [yes or no], glomerular filtration rate missing indicator [yes or no], smoking status missing indicator [yes or no], alcoholic drinks missing indicator [yes or no], and aerobic physical activity missing indicator [yes or no]).

Comment 4.

The data clearly demonstrate that allergic burden in long-COVID applies only to respiratory system, but not the skin or GI tract. It would be very relevant to investigate parental history of allergic diseases as one of the criteria, which can influence allergic diseases development in presented cohorts.

Response:

Thank you for your comment. As you mentioned, our results showed a clear risk of long-COVID in asthma and allergic rhinitis, but the findings were not significant for atopic dermatitis and food allergies. For this reason, we are also interested in investigating the parental history of allergic diseases to understand its impact on the susceptibility of offspring to such conditions. However, our data lacks information on the parents' history of allergies. Additionally, while there is evidence of such connections in children, it remains uncertain whether this association persists into adulthood.

The relevance of parents' history of allergic diseases in children and teenagers is due to the complex interaction of genetic and environmental factors that affect the development of allergic conditions.¹ In childhood, genetic influences might be more significant, highlighting the importance of a parent's allergic history in showing the impact of genetics. Previous research on the connection between a parent's allergic history and allergy occurrence in children has shown that genetics play a crucial role.^{2,3}

On the other hand, in adulthood, the accumulated exposure to environmental factors might play a more significant role. As people grow into adults, they experience various environments, and aspects like habits, occupation, and dietary choices often change significantly.^{4,5} As a result, the development of allergies in adults is probably more influenced by current environmental factors and personal exposures than by their parents' history of allergic diseases. Moreover, while genetic factors continue to play a role in adulthood, the

impact of a family history of allergies becomes less significant compared to other factors. For example, specific genetic markers linked to a predisposition for allergies, like variations in genes related to immunoglobulins or cytokines, might still be influential in adults.⁶ However, the complex interplay with numerous other factors reduces the relative impact of parental allergic history on the development of allergic conditions. Therefore, in adults, the manifestation of a clear genetic predisposition to allergic diseases may not be as apparent.

Although our current data does not allow us to consider the genetic factors related to parents' allergic diseases, which is a limitation, we do acknowledge the importance of this aspect. We believe that it could be a significant area for further exploration by the broader research community. This suggestion will certainly be considered as a potential area for future research in this field.

<Reference>

1. Gowett MQ, Perry SS, Aggarwal R, Zhou LT, Pavone ME, Duncan FE, Cheng WS. Associations of childhood allergies with parental reproductive and allergy history. *J Assist Reprod Genet.* 2023 Jun;40(6):1349-1359. doi: 10.1007/s10815-023-02801-3. Epub 2023 May 3. PMID: 37133690; PMCID: PMC10310638.
2. Thorsteinsdottir S, Stokholm J, Thyssen JP, Nørgaard S, Thorsen J, Chawes BL, Bønnelykke K, Waage J, Bisgaard H. Genetic, Clinical, and Environmental Factors Associated With Persistent Atopic Dermatitis in Childhood. *JAMA Dermatol.* 2019 Jan 1;155(1):50-57. doi: 10.1001/jamadermatol.2018.4061. PMID: 30427975; PMCID: PMC6439574.
3. Johnson CC, Chandran A, Havstad S, et al. US Childhood Asthma Incidence Rate Patterns From the ECHO Consortium to Identify High-risk Groups for Primary Prevention. *JAMA Pediatr.* 2021;175(9):919–927. doi:10.1001/jamapediatrics.2021.0667

4. Morales E, Duffy D. Genetics and Gene-Environment Interactions in Childhood and Adult Onset Asthma. *Front Pediatr.* 2019 Dec 11;7:499. doi: 10.3389/fped.2019.00499. PMID: 31921716; PMCID: PMC6918916.
5. Cookson W. The alliance of genes and environment in asthma and allergy. *Nature.* 1999 Nov 25;402(6760 Suppl):B5-11. doi: 10.1038/35037002. PMID: 10586889.
6. Chiarella SE, Fernandez R, Avila PC. The genes and the environment in nasal allergy. *Curr Opin Allergy Clin Immunol.* 2015 Oct;15(5):440-5. doi: 10.1097/ACI.0000000000000207. PMID: 26308330.

Changes in text:

Discussion, *Limitations and strengths*

Fifth, our current data set limits our ability to consider genetic factors related to parents' allergic diseases. However, in adults, the indications of a clear genetic predisposition to allergic conditions may not be as evident.

Comment 5.

In the context of newly developed allergic diseases. What are patients allergic to? Are those airborne, food, or other allergens?

Response:

Thank you for your detailed comments. First, we apologize for not adequately explaining ‘allergic diseases’ in our methods. In our manuscript, we have used the term ‘allergic diseases’ to collectively describe individuals with any of the following conditions: allergic rhinitis, atopic dermatitis, food allergy, or asthma. This definition has also been used in the following paper.

- Noh Y, Jeong HE, Choi A, Choi EY, Pasternak B, Nordeng H, Bliddal M, Man KKC, Wong ICK, Yon DK, Shin JY. Prenatal and Infant Exposure to Acid-Suppressive Medications and Risk of Allergic Diseases in Children. *JAMA Pediatr.* 2023 Mar 1;177(3):267-277. doi: 10.1001/jamapediatrics.2022.5193. PMID: 36622684; PMCID: PMC9857801.

[REDACTED]

→ **Methods “Outcomes”**: The primary outcome, allergic diseases, was a composite endpoint of asthma, allergic rhinitis, atopic dermatitis, and food allergy.

- Koo MJ, Kwon R, Lee SW, Choi YS, Shin YH, Rhee SY, Min C, Cho SH, Turner S, Kim SY, Lee J, Yeo SG, Abuabara K, Lee YJ, Shin JI, Kim JH, Shin JU, Yon DK, Papadopoulos NG. National trends in the prevalence of allergic diseases among Korean adolescents before and during COVID-19, 2009-2021: A serial analysis of the national representative study. *Allergy*. 2023 Jun;78(6):1665-1670. doi: 10.1111/all.15600. Epub 2022 Dec 8. PMID: 36440490.

[REDACTED]

→ **Methods:** Allergic morbidity was defined as having at least one of the three conditions mentioned above.

From a research perspective, we believe that categorizing these conditions together contributes to a deeper understanding of the common genetic, environmental, and immunological factors that contribute to allergic diseases. We have added the definition of the term to the “Methods” section and hope this addresses your concerns.

Changes in text:

Methods, *Exposures and outcomes*

Also, the term 'allergic diseases' refers to a diagnosis of any of the following condition: asthma, AR, AD, or FA.^{16,17}

Reference

16. Noh Y, et al. Prenatal and Infant Exposure to Acid-Suppressive Medications and Risk of Allergic Diseases in Children. *JAMA Pediatr* 177, 267-277 (2023).

17. Koo MJ, et al. National trends in the prevalence of allergic diseases among Korean adolescents before and during COVID-19, 2009-2021: A serial analysis of the national representative study. *Allergy* 78, 1665-1670 (2023).

Comment 6.

What is the history of SARS-CoV and MERS epidemics in the geographic regions of the included cohort? Can the results of long-COVID be influenced by previous contact with other coronaviruses? Is it possible to get such information for included cohorts?

Response:

Thank you for your interesting question. We agree that it is important to analyze how SARS-CoV and MERS-CoV may influence the results of long-COVID since previous exposure to other coronaviruses can affect the response of the immune system to SARS-CoV-2 infection. Indeed, there is a prior finding indicating that partial cross-immunity, which arises from previous MERS-CoV infections, could lead to a reduced risk of SARS-CoV-2 infections.¹ However, it does not cover the long-term effects of SARS-CoV-2 infection.

Unfortunately, the Korean, Japanese, and British cohorts that we used in the current study do not include information on these viruses. Hence, we were unable to investigate the history of SARS-CoV and MERS-CoV epidemics in these geographic regions and their potential influence on long-COVID results. We recognize the importance of additional analysis and suggest that future studies should be designed to include previous data on other coronaviruses. It would be beneficial to investigate whether individuals with a history of other coronavirus infections, such as SARS-CoV and MERS, exhibit different long-COVID patterns compared to those without such a history.

Meanwhile, according to the World Health Organization in 2003, there were 3, 0, and 4 confirmed cases of SARS in Korea, Japan, and the UK, respectively (<https://www.who.int/publications/m/item/summary-of-probable-sars-cases-with-onset-of-illness-from-1-november-2002-to-31-july-2003>). There have been 186 confirmed cases of MERS in Korea since 2015. However, there have been less than 10 accumulative cases of

MERS in Japan and the UK.² As Japan and the UK were spared to SARS and MERS epidemics, there may be no significant difference between before and after considering the history of other coronaviruses. The relatively small number of confirmed cases of SARS and MERS in Japan and the UK suggests that there might be no significant difference in association even after considering the history of other coronaviruses. A noticeable number of patients with MERS in Korea may overlook confounders.

Although, regrettably, we are unable to conduct this intriguing research directly, we recognize its importance. Therefore, we have included the potential impact of the history of other coronaviruses in the manuscript. Thank you for your constructive feedback, which will undoubtedly enrich this study.

<Reference>

1. El-Saed A, Othman F, Baffoe-Bonnie H, Almulhem R, Matalqah M, Alshammari L, Alshamrani MM. Symptomatic MERS-CoV infection reduces the risk of future COVID-19 disease; a retrospective cohort study. *BMC Infect Dis.* 2023 Nov 3;23(1):757. doi: 10.1186/s12879-023-08763-2. PMID: 37924004; PMCID: PMC10623690.
2. Zhao H, ParryFord F, Dabrera G, Sinnathamby M, Ellis J, Dunning J, Osman H, Machin N, Pebody R. Six-year experience of detection and investigation of possible Middle East Respiratory Syndrome coronavirus cases, England, 2012-2018. *Public Health.* 2020 Dec;189:141-143. doi: 10.1016/j.puhe.2020.10.007. Epub 2020 Oct 20. PMID: 33227597; PMCID: PMC7574929.

Changes in text:

Discussion, *Limitations and strengths*

Sixth, we did not take the previous history of severe acute respiratory syndrome and Middle East respiratory syndrome epidemics in South Korea, Japan, and the UK into consideration, which may serve as a potential confounder.

Comment 7.

Can presented results be biased by different treatment regimens applied for moderate or severe COVID-19 patients in different countries?

Response:

Thank you for your valuable question. We concur with your concern that different treatment regimens applied to COVID-19 patients based on the severity of their condition could bias the results. However, our data presents challenges in accounting for variations in treatment methods according to the severity, which is a limitation of our study. Nonetheless, to minimize the impact of this potential bias, we categorized patients into two groups: mild COVID-19 patients and moderate to severe COVID-19 patients (Table S25-S26). Additionally, we conducted further analyses in Japan and the UK.

Despite analyzing the groups separately based on severity, we found significant associations between SARS-CoV-2 infection and allergic outcomes in both groups. Also, the results primarily show higher incidences in patients with moderate to severe COVID-19. This observation can be attributed to the fact that moderate to severe COVID-19 is often associated with an elevated immune response and the release of various cytokines and inflammatory markers.¹ The excessive immune response can potentially induce or exacerbate allergic reactions.¹ We hope our alternations are in line with your expectations.

<Reference>

1. Que Y, Hu C, Wan K, Hu P, Wang R, Luo J, Li T, Ping R, Hu Q, Sun Y, Wu X, Tu L, Du Y, Chang C, Xu G. Cytokine release syndrome in COVID-19: a major mechanism of morbidity and mortality. *Int Rev Immunol.* 2022;41(2):217-230. doi:

10.1080/08830185.2021.1884248. Epub 2021 Feb 22. PMID: 33616462; PMCID: PMC7919105.

Changes in text:

Methods, *Sensitivity analysis*

Fifth, in order to examine the impact of COVID-19 severity on allergic diseases, the mild group and the moderate to severe group were analyzed as two separate cohorts (**Tables S25 and S26**).

Discussion, *Limitations and strengths*

Eighth, we conducted additional sensitivity analyses to capture mild cases of COVID-19 as comprehensively as possible (**Tables S25 and S26**). However, the potential exclusion of milder cases still exists. Additionally, our data may be biased due to different treatment methods for patients with COVID-19 based on the severity of their illness.

Table S25. The HR with 95% CI for the long-term sequelae risk of incident allergic diseases following mild COVID-19 diagnosis in the propensity score-matched main cohort (South Korea), replication cohort A (Japan), and replication cohort B (UK).

Cohort	Exposure	Incidence rate*	HR (95% CI)	
			Model 1 ^a	Model 2 ^b
Allergic diseases				
Main cohort	Non-infected	23.3	1.00 (reference)	1.00 (reference)
Main cohort	Patients with COVID-19	26.8	1.15 (1.08 to 1.23)	1.15 (1.08 to 1.23)
Replication cohort A	Non-infected	27.4	1.00 (reference)	1.00 (reference)
Replication cohort A	Patients with COVID-19	75.0	2.64 (2.61 to 2.68)	2.56 (2.52 to 2.59)
Replication cohort B	Non-infected	7.2	1.00 (reference)	1.00 (reference)
Replication cohort B	Patients with COVID-19	8.2	1.13 (1.05 to 1.21)	1.11 (1.04 to 1.19)
Asthma				
Main cohort	Non-infected	0.9	1.00 (reference)	1.00 (reference)
Main cohort	Patients with COVID-19	1.9	2.17 (1.67 to 2.83)	2.18 (1.67 to 2.84)
Replication cohort A	Non-infected	2.7	1.00 (reference)	1.00 (reference)
Replication cohort A	Patients with COVID-19	7.7	2.81 (2.68 to 2.94)	2.67 (2.55 to 2.80)
Replication cohort B	Non-infected	3.3	1.00 (reference)	1.00 (reference)
Replication cohort B	Patients with COVID-19	3.8	1.15 (1.04 to 1.28)	1.13 (1.01 to 1.25)
Allergic rhinitis				
Main cohort	Non-infected	16.9	1.00 (reference)	1.00 (reference)
Main cohort	Patients with COVID-19	20.1	1.19 (1.10 to 1.28)	1.19 (1.11 to 1.29)
Replication cohort A	Non-infected	20.4	1.00 (reference)	1.00 (reference)
Replication cohort A	Patients with COVID-19	62.4	2.97 (2.92 to 3.02)	2.88 (2.83 to 2.93)
Replication cohort B	Non-infected	1.1	1.00 (reference)	1.00 (reference)
Replication cohort B	Patients with COVID-19	1.4	1.22 (1.03 to 1.46)	1.21 (1.02 to 1.44)
Atopic dermatitis				

Main cohort	Non-infected	2.5	1.00 (reference)	1.00 (reference)
Main cohort	Patients with COVID-19	2.9	1.16 (0.95 to 1.41)	1.16 (0.95 to 1.41)
Replication cohort A	Non-infected	5.2	1.00 (reference)	1.00 (reference)
Replication cohort A	Patients with COVID-19	6.8	1.24 (1.19 to 1.29)	1.21 (1.16 to 1.26)
Replication cohort B	Non-infected	0.04	1.00 (reference)	1.00 (reference)
Replication cohort B	Patients with COVID-19	0.05	1.18 (0.47 to 2.99)	1.17 (0.46 to 2.98)
Food allergy				
Main cohort	Non-infected	3.4	1.00 (reference)	1.00 (reference)
Main cohort	Patients with COVID-19	2.6	0.76 (0.63 to 0.93)	0.76 (0.63 to 0.93)
Replication cohort A	Non-infected	1.3	1.00 (reference)	1.00 (reference)
Replication cohort A	Patients with COVID-19	2.6	2.05 (1.90 to 2.20)	1.82 (1.69 to 1.96)
Replication cohort B	Non-infected	3.1	1.00 (reference)	1.00 (reference)
Replication cohort B	Patients with COVID-19	3.4	1.07 (0.96 to 1.20)	1.06 (0.95 to 1.18)

CI, confidence interval; HR, hazard ratio.

* Incidence rate is expressed as per 1,000 person-years.

The data in bold indicate significant differences ($P < 0.05$).

^a **Model 1:** Adjusted for age (20–39, 40–59, and ≥ 60 years) and sex.

^b **Model 2 (main cohort):** Adjusted for age (20–39, 40–59, and ≥ 60 years); sex; household income (low income, middle income, and high income); region of residence (urban and rural); Charlson comorbidity index (0, 1, and ≥ 2); BMI (underweight [$< 18.5 \text{ kg/m}^2$], normal [$18.5\text{--}23.0 \text{ kg/m}^2$], overweight [$23.0\text{--}25.0 \text{ kg/m}^2$], obese [$\geq 25.0 \text{ kg/m}^2$], and unknown); blood pressure (systolic blood pressure $< 140 \text{ mmHg}$ and diastolic blood pressure $< 90 \text{ mmHg}$, systolic blood pressure $\geq 140 \text{ mmHg}$ or diastolic blood pressure $\geq 90 \text{ mmHg}$, and unknown); fasting blood glucose (< 100 , $\geq 100 \text{ mg/dL}$, and unknown); serum total cholesterol (< 200 , $200\text{--}240$, $\geq 240 \text{ mg/dL}$, and unknown); glomerular filtration rate

[revised manuscript text omitted]

Table S26. The HR with 95% CI for the long-term sequelae risk of incident allergic diseases following moderate to severe COVID-19 diagnosis in the propensity score-matched main cohort (South Korea), replication cohort A (Japan), and replication cohort B (UK).

Cohort	Exposure	Incidence rate*	HR (95% CI)	
			Model 1 ^a	Model 2 ^b
Allergic diseases				
Main cohort	Non-infected	27.8	1.00 (reference)	1.00 (reference)
Main cohort	Patients with COVID-19	39.3	1.41 (1.24 to 1.61)	1.42 (1.24 to 1.61)
Replication cohort A	Non-infected	27.6	1.00 (reference)	1.00 (reference)
Replication cohort A	Patients with COVID-19	81.0	2.93 (2.68 to 3.20)	2.75 (2.49 to 3.03)
Replication cohort B	Non-infected	5.8	1.00 (reference)	1.00 (reference)
Replication cohort B	Patients with COVID-19	12.2	2.09 (1.04 to 4.20)	1.97 (0.95 to 4.06)
Asthma				
Main cohort	Non-infected	1.5	1.00 (reference)	1.00 (reference)
Main cohort	Patients with COVID-19	4.0	2.59 (1.66 to 4.03)	2.47 (1.58 to 3.85)
Replication cohort A	Non-infected	3.1	1.00 (reference)	1.00 (reference)
Replication cohort A	Patients with COVID-19	4.4	1.43 (1.03 to 1.99)	1.19 (0.83 to 1.71)
Replication cohort B	Non-infected	1.7	1.00 (reference)	1.00 (reference)
Replication cohort B	Patients with COVID-19	5.6	3.3 (1.06 to 10.25)	3.33 (1.05 to 10.60)
Allergic rhinitis				
Main cohort	Non-infected	19.4	1.00 (reference)	1.00 (reference)
Main cohort	Patients with COVID-19	27.7	1.42 (1.22 to 1.66)	1.43 (1.22 to 1.67)
Replication cohort A	Non-infected	21.0	1.00 (reference)	1.00 (reference)
Replication cohort A	Patients with COVID-19	67.2	3.19 (2.89 to 3.53)	3.09 (2.77 to 3.45)
Replication cohort B	Non-infected	1.2	1.00 (reference)	1.00 (reference)
Replication cohort B	Patients with COVID-19	0	NA	NA
Atopic dermatitis				

Main cohort	Non-infected	2.8	1.00 (reference)	1.00 (reference)
Main cohort	Patients with COVID-19	3.0	1.10 (0.70 to 1.73)	1.11 (0.70 to 1.74)
Replication cohort A	Non-infected	4.2	1.00 (reference)	1.00 (reference)
Replication cohort A	Patients with COVID-19	6.8	1.64 (1.25 to 2.15)	1.57 (1.16 to 2.11)
Replication cohort B	Non-infected	0	1.00 (reference)	1.00 (reference)
Replication cohort B	Patients with COVID-19	0	NA	NA
Food allergy				
Main cohort	Non-infected	4.6	1.00 (reference)	1.00 (reference)
Main cohort	Patients with COVID-19	5.4	1.18 (0.84 to 1.67)	1.18 (0.84 to 1.66)
Replication cohort A	Non-infected	1.5	1.00 (reference)	1.00 (reference)
Replication cohort A	Patients with COVID-19	5.4	3.77 (2.64 to 5.39)	2.53 (1.70 to 3.77)
Replication cohort B	Non-infected	2.9	1.00 (reference)	1.00 (reference)
Replication cohort B	Patients with COVID-19	7.5	2.47 (0.97 to 6.28)	1.99 (0.74 to 5.39)

CI, confidence interval; HR, hazard ratio.

* Incidence rate is expressed as per 1,000 person-years.

The data in bold indicate significant differences ($P < 0.05$).

^a **Model 1:** Adjusted for age (20–39, 40–59, and ≥ 60 years) and sex.

^b **Model 2 (main cohort):** Adjusted for age (20–39, 40–59, and ≥ 60 years); sex; household income (low income, middle income, and high income); region of residence (urban and rural); Charlson comorbidity index (0, 1, and ≥ 2); BMI (underweight [$< 18.5 \text{ kg/m}^2$], normal [$18.5\text{--}23.0 \text{ kg/m}^2$], overweight [$23.0\text{--}25.0 \text{ kg/m}^2$], obese [$\geq 25.0 \text{ kg/m}^2$], and unknown); blood pressure (systolic blood pressure $< 140 \text{ mmHg}$ and diastolic blood pressure $< 90 \text{ mmHg}$, systolic blood pressure $\geq 140 \text{ mmHg}$ or diastolic blood pressure $\geq 90 \text{ mmHg}$, and unknown); fasting blood glucose (< 100 , $\geq 100 \text{ mg/dL}$, and unknown); serum total cholesterol (< 200 , $200\text{--}240$, $\geq 240 \text{ mg/dL}$, and unknown); glomerular filtration rate

[revised manuscript text omitted]

Comment 8.

Line 197. There is a typo in the COVID-19 word.

Response:

We appreciate your attention to detail and the identification of the typographical error in line 197. As you suggested, we attempted to correct the 'COIVD-19' typo during the revision process. However, in the course of revising, the section divisions were lost, leading to the deletion of that specific word. Instead, we have thoroughly reviewed the entire manuscript to ensure that similar typographical errors are absent throughout. Thank you for bringing this to our attention, and we will make the necessary corrections in the revised version of the manuscript.

Changes in text:

Results, ~~Incident allergic outcomes following COIVD-19~~

Reviewer #2**Comment 0.**

The study uses large cohorts with clinical marker to investigate the onset of allergic diseases as a result of a SARS-CoV-2 Infection. One of the cohorts even has information on vaccination status. However, the absolute onset of allergic disease after the infection is low and the highest effect sizes are reported for asthma. I believe that the asthma diagnosis is likely caused by misclassification of dyspnea.

Response:

Thank you for your insightful feedback regarding our study, where we investigated the onset of allergic diseases following a SARS-CoV-2 infection using large cohorts, including information on vaccination status. We are also aware of your observation that the absolute onset of allergic diseases after the infection is low, and that a high effect size was observed in asthma. To address these concerns, we conducted three additional sensitivity analyses to further enhance our findings: negative control analysis (Table S21), analysis of asthma excluding dyspnea symptoms (Table S22), and analysis of asthma with using strict diagnostic criteria (Table S23). We will elaborate on the details of these analyses in the following comment. These additional analyses are intended to strengthen the validity of our findings and address the potential issue of misclassification, particularly regarding asthma diagnoses after SARS-CoV-2 infection. We believe these measures will provide a more accurate understanding of the relationship between infection of SARS-CoV-2 and the onset of allergic diseases.

Approach related comments:**Comment 1.**

The study did not use negative control endpoints to rule out detection bias. Those with a SARS-CoV-2 infection might be under better surveillance, which results in an earlier detection of disease than in the control group due to higher healthcare utilization. Most effect estimates are below 30% risk increase/decrease. The exception is the JMDC cohort and asthma. Asthma could be misclassified by physicians as dyspnea. Dyspnea is one of the few symptoms, which made it into the WHO clinical case definition of post-COVID (for adults) in October 2021. This bias could be higher in Japan than South Korea rather than showing a much higher risk for developing asthma after the infection in Japan. The only way to rule it out would be to test how many onset asthma patients have no incident diagnosis of dyspnea. Also, the analysis would benefit from a more specific definition (along the sensitive definition used) of the allergic disease with ICD-10 code and suitable medication to reduce misclassification bias for rare endpoints. Table 3 shows the time attenuation of the effect. In the K-CoV-N cohort the effect nearly disappeared over time, which raised the question if an onset of allergic diseases were observed or only flares of already existing disease (despite a 2-year baseline period).

Response:

Thank you for your insightful comments on the key aspects of the current study. We have organized your feedback into three main categories. In this response, we intend to provide a detailed explanation of how each comment was addressed and incorporated into the revision of our study.

1. We acknowledge that our study, an observational study targeting specific participants, may be subject to detection bias. In particular, higher healthcare utilization among individuals infected with SARS-CoV-2, the possibility of including dyspnea in the

diagnosis of asthma, and the overall higher results for the Japanese cohort compared to the other two cohorts were enough to make us think that this bias may have been particularly relevant. Understanding these concerns, we have conducted three additional sensitivity analyses to address them. First, we conducted a negative control analysis for each of the three cohorts to verify the validity of our findings and to identify potential errors or biases. For this analysis, we estimated adjusted hazard ratios (aHR) for tympanic membrane perforation disease based on exposure to COVID-19 infection and found that the results were not statistically significant in all cohorts (Table S21).¹ Second, we performed an analysis for allergic diseases and asthma, excluding cases where codes for both asthma and dyspnea were recorded simultaneously (Table S22). For the replication cohorts (Japan and the UK), there was no significant difference before and after excluding dyspnea, with aHR values decreased by within 2%. In contrast, for the main cohort of South Korea, the aHR values showed a decrease of 4% for allergic disease and 13% for asthma, raising concerns about the potential impact of dyspnea in the diagnosis of asthma. Therefore, we established a strict diagnostic criterion for asthma in the main cohort and proceeded with a third additional sensitivity analysis. We conducted analyses on cases diagnosed with asthma, considering those with a history of emergency department visits or hospitalization (Table S23).² The analysis revealed that the aHR was 18% higher compared to the results using the previous definition of asthma, more clearly suggesting that exposure to COVID-19 significantly contributes to an increase in the onset of asthma.

2. Following your suggestion, we concur that in our study, more precisely defining allergic diseases using ICD-10 codes and appropriate medications will likely be

beneficial in reducing misclassification bias, particularly for rare endpoints. Therefore, we redefined medication related to each allergic disease in the main cohort and replication cohort A as follows: asthma-related medication (inhaled corticosteroids, systemic corticosteroids, long- and short-acting β 2-agonists, and/or leukotriene antagonists); allergic rhinitis (antihistamines, corticosteroid nasal sprays, leukotriene antagonists, and/or mast cell stabilizers); atopic dermatitis (topical and systemic corticosteroids, calcineurin inhibitors, and systemic immunosuppressants [azathioprine, cyclosporine, mycophenolate mofetil, and/or methotrexate]); and food allergy. We believe that these measures contribute to mitigating concerns about misclassification bias and enhance the reliability of the current study.

3. We admit that the explanation regarding the analysis based on the timeline in Table 3 may cause confusion and misunderstanding of the results. In each period we defined, individuals with an outcome were considered based on their initial diagnosis, including the pre-period. For example, individuals who were diagnosed after 6 months were those who did not receive a diagnosis during the pre-observation period as well. Therefore, we have revised the explanation to provide a clearer representation in the manuscript.

Through your comments, we identified the weaknesses in our study. We have incorporated supplementary details into the methods section, specifically addressing sensitivity analysis and aspects where methodological explanations were previously insufficient. We appreciate your insightful comments once again.

< Reference >

1. Lim SH, Ju HJ, Han JH, Lee JH, Lee WS, Bae JM, Lee S. Autoimmune and Autoinflammatory Connective Tissue Disorders Following COVID-19. *JAMA Netw Open*. 2023 Oct 2;6(10):e2336120. doi: 10.1001/jamanetworkopen.2023.36120. PMID: 37801317; PMCID: PMC10559181.
2. Christiansen SC, Schatz M, Yang SJ, Ngor E, Chen W, Zuraw BL. Hypertension and Asthma: A Comorbid Relationship. *J Allergy Clin Immunol Pract*. 2016 Jan-Feb;4(1):76-81. doi: 10.1016/j.jaip.2015.07.009. Epub 2015 Sep 3. PMID: 26342745.

Changes in text:

Abstract

After a 1:5 propensity score matching, we observed an elevated risk of developing allergic diseases associated with COVID-19, beyond the 30 days after COVID-19 diagnosis (HR, 1.20; 95% CI, 1.13-1.27). This association was specifically shown 
[revised manuscript text omitted]
 $< 140 \text{ mmHg}$ and diastolic blood pressure $< 90 \text{ mmHg}$, systolic blood pressure $\geq 140 \text{ mmHg}$ or diastolic blood pressure $\geq 90 \text{ mmHg}$, and unknown); fasting blood glucose (<100 , $\geq 100 \text{ mg/dL}$, and unknown); smoking status (non- and current smoker, and unknown); alcohol consumption (every day, sometimes, rarely days per week, and unknown); aerobic physical activity (low, moderate, high, and unknown); previous history of cardiovascular disease, chronic kidney disease, and chronic obstructive pulmonary disease; history of medication use for diabetes mellitus, dyslipidemia, and hypertension; and missing indicators (household income missing indicator [yes or no], townsend deprivation index missing indicator [yes or no], ethnicity missing indicator [yes or no], education levels [yes or no], obesity missing indicator [yes or no], blood pressure missing indicator [yes or no], fasting blood glucose missing indicator [yes or no], serum total cholesterol missing indicator [yes or no], glomerular filtration rate missing indicator [yes or no], smoking status missing indicator [yes or no], alcoholic drinks missing indicator [yes or no], and aerobic physical activity missing indicator [yes or no]).

Table 3. Time attenuation effect on the development of allergic diseases after SARS-CoV-2 infection (model 2 adjusted HR with 95% CI).

	HR (95% CI)	
	Main cohort (South Korea)	Replication cohort A (Japan)
Allergic diseases		
<3 months	1.42 (1.29 to 1.56)	3.30 (3.24 to 3.36)
3–6 months	1.14 (1.01 to 1.29)	1.77 (1.70 to 1.84)
≥6 months	1.00 (0.91 to 1.11)	1.61 (1.56 to 1.67)

BMI, body mass index; CI, confidence interval; HR, hazard ratio; SARS-CoV-2, severe acute respiratory syndrome coronavirus.

The data in bold indicate significant differences ($P < 0.05$).

Model 2 (main cohort): Adjusted for age (20–39, 40–59, and ≥60 years); sex; household income (low income, middle income, and high income); region of residence (urban and rural); Charlson comorbidity index (0, 1, and ≥2); BMI (underweight [$<18.5 \text{ kg/m}^2$], normal [$18.5\text{--}23.0 \text{ kg/m}^2$], overweight [$23.0\text{--}25.0 \text{ kg/m}^2$], obese [$\geq 25.0 \text{ kg/m}^2$], and unknown); blood pressure (systolic blood pressure $<140 \text{ mmHg}$ and diastolic blood pressure $<90 \text{ mmHg}$, systolic blood pressure $\geq 140 \text{ mmHg}$ or diastolic blood pressure $\geq 90 \text{ mmHg}$, and unknown); fasting blood glucose (<100 , $\geq 100 \text{ mg/dL}$, and unknown); serum total cholesterol (<200 , $200\text{--}240$, $\geq 240 \text{ mg/dL}$, and unknown); glomerular filtration rate (<60 , $60\text{--}90$, $\geq 90 \text{ mL/min/1.73 m}^2$, and unknown); smoking status (non-, ex-, current smoker, and unknown); alcoholic drinks (<1 , $1\text{--}2$, $3\text{--}4$, ≥ 5 days per week, and unknown); aerobic physical activity (sufficient, insufficient, and unknown); previous history of cardiovascular disease, chronic kidney disease, and chronic obstructive pulmonary disease; history of medication use for diabetes mellitus, dyslipidemia, and hypertension; and missing indicators (BMI missing indicator [yes or no], blood pressure missing indicator [yes or no], fasting blood glucose missing indicator [yes or no], serum total cholesterol missing indicator [yes or no], glomerular filtration rate missing indicator [yes or no], smoking status missing indicator [yes or no], alcoholic drinks missing indicator [yes or no], and aerobic physical activity missing indicator [yes or no]).

[revised manuscript text omitted]

Table S1. Definitions of diseases.

Diseases	ICD-10 codes
Asthma	J45 or J46 with ≥ 2 claims and the use of asthma related medications (inhaled corticosteroid and systemic corticosteroids, long- and short-acting $\beta 2$ -agonists, and/or leukotriene antagonists)
Allergic asthma	Asthma with at least one of the following conditions: allergic rhinitis, atopic dermatitis, or food allergy.
Non-allergic asthma	Asthma occurring without any of the following conditions: allergic rhinitis, atopic dermatitis, and food allergy.
Allergic rhinitis	J30.1, J30.2, J30.3, or J30.4 with ≥ 2 claims and the use of allergic rhinitis related medications (antihistamines, corticosteroid nasal sprays, leukotriene antagonists, and/or mast cell stabilizers).
Atopic dermatitis	L20 with ≥ 2 claims and the use of atopic dermatitis related medications (topical and systemic corticosteroids, calcineurin inhibitors, and systemic immunosuppressants [azathioprine, cyclosporine, mycophenolate mofetil, and/or methotrexate]).
Food allergy	K52.2, K52.3, K52.8, K52.9, L27.2, L23.6, T78.0, T78.1, or Z91.0 with ≥ 2 claims
History of cardiovascular disease	I00-I02, I05-I09, I50, I51, I52, or I20-I45
History of chronic kidney disease	E10.2, E11.2, E13.2, E14.2, I12.0, M20.0, M31.3, M31.9, M32.1B, N02-N08, N11, N12, N14, N15.8, N15.9, N16.0, N16.2, N16.3, N16.4, N16.8, N18, N19, N26, Q61.2, Q61.3, or Q61.5 with ≥ 2 claims
History of chronic obstructive pulmonary disease	J44.0, J44.1, or J44.9 with ≥ 2 claims
Dyspnea	R06.0 with ≥ 1 claims
Tympanic membrane perforation	H72 with ≥ 2 claims

ICD-10, International Classification of Diseases, 10th edition.

Table S21. The HR with 95% CI for the long-term sequelae risk of incident allergic disease in negative control analysis using non-COVID-19 disease (tympanic membrane perforation) in the propensity score-matched main cohort (South Korea), replication cohort A (Japan), and replication cohort B (UK).

Cohort	Exposure	Incidence rate*	HR (95% CI)	
			Model 1 ^a	Model 2 ^b
Main cohort	None	0.5	1.00 (reference)	1.00 (reference)
Main cohort	Patients with COVID-19	0.4	0.85 (0.60 to 1.20)	0.85 (0.60 to 1.20)
Replication cohort A	None	0.0	1.00 (reference)	1.00 (reference)
Replication cohort A	Patients with COVID-19	0.1	1.09 (0.68 to 1.74)	0.81 (0.39 to 1.68)
Replication cohort B	None	0.39	1.00 (reference)	1.00 (reference)
Replication cohort B	Patients with COVID-19	0.38	0.97 (0.72 to 1.30)	0.98 (0.73 to 1.32)

BMI, body mass index; CI, confidence interval; HR, hazard ratio.

* Incidence rate is expressed as per 1,000 person-years.

^a **Model 1:** Adjusted for age (20–39, 40–59, and ≥60 years) and sex.

^b **Model 2 (main cohort):** Adjusted for age (20–39, 40–59, and ≥60 years); sex; household income (low income, middle income, and high income); region of residence (urban and rural); Charlson comorbidity index (0, 1, and ≥2); BMI (underweight [$<18.5 \text{ kg/m}^2$], normal [$18.5\text{--}23.0 \text{ kg/m}^2$], overweight [$23.0\text{--}25.0 \text{ kg/m}^2$], obese [$\geq 25.0 \text{ kg/m}^2$], and unknown); blood pressure (systolic blood pressure $<140 \text{ mmHg}$ and diastolic blood pressure $<90 \text{ mmHg}$, systolic blood pressure $\geq 140 \text{ mmHg}$ or diastolic blood pressure $\geq 90 \text{ mmHg}$, and unknown); fasting blood glucose (<100 , $\geq 100 \text{ mg/dL}$, and unknown); serum total cholesterol (<200 , $200\text{--}240$, $\geq 240 \text{ mg/dL}$, and unknown); glomerular filtration rate (<60 , $60\text{--}90$, $\geq 90 \text{ mL/min/1.73 m}^2$, and unknown); smoking status (non-, ex-, current smoker, and unknown); alcoholic drinks (<1 , $1\text{--}2$, $3\text{--}4$, ≥ 5 days per week, and unknown); aerobic physical activity (sufficient, insufficient, and unknown); previous history of cardiovascular disease, chronic kidney disease, and chronic obstructive pulmonary disease; history of medication use for diabetes mellitus, dyslipidemia, and hypertension; and missing indicators (BMI missing indicator [yes or no], blood pressure missing indicator [yes or no], fasting blood glucose

missing indicator [yes or no], serum total cholesterol missing indicator [yes or no], glomerular filtration rate missing indicator [yes or no], smoking status missing indicator [yes or no], alcoholic drinks missing indicator [yes or no], and aerobic physical activity missing indicator [yes or no]).

^b Model 2 (replication cohort A): Adjusted for age (20–39, 40–59, and ≥ 60 years); sex; Charlson comorbidity index (0, 1, and ≥ 2); BMI (underweight [<18.5 kg/m²], normal [18.5–23.0 kg/m²], overweight [23.0–25.0 kg/m²], obese [≥ 25.0 kg/m²], and unknown); blood pressure (systolic blood pressure <140 mmHg and diastolic blood pressure <90 mmHg, systolic blood pressure ≥ 140 mmHg or diastolic blood pressure ≥ 90 mmHg, and unknown); fasting blood glucose (<100 , ≥ 100 mg/dL, and unknown); serum total cholesterol (<200 , 200–240, ≥ 240 mg/dL, and unknown); glomerular filtration rate (<60 , 60–90, ≥ 90 mL/min/1.73 m², and unknown); smoking status (non- and current smoker, and unknown); alcoholic drinks (drinks; <1 , 1–2, 3–4, ≥ 5 days per week, and unknown); aerobic physical activity (sufficient, insufficient, and unknown); previous history of cardiovascular disease, chronic kidney disease, and chronic obstructive pulmonary disease; history of medication use for diabetes mellitus, dyslipidemia, and hypertension; and missing indicators (BMI missing indicator [yes or no], blood pressure missing indicator [yes or no], fasting blood glucose missing indicator [yes or no], serum total cholesterol missing indicator [yes or no], glomerular filtration rate missing indicator [yes or no], smoking status missing indicator [yes or no], alcoholic drinks missing indicator [yes or no], and aerobic physical activity missing indicator [yes or no]).

^b Model 2 (replication cohort B): Adjusted for age (20–39, 40–59, and ≥ 60 years); sex; household income ($<£18,000$, £18,000–£30,999, £31,000–£51,999, £52,000–£100,000, $>£100,000$), and unknown); region of residence (urban and rural); townsend deprivation index (T1 [last deprived], T2, T3 [most deprived], and unknown); race (white, mixed, Asian, black, others, and unknown); Charlson comorbidity index (0, 1, and ≥ 2); BMI (normal [<25.0 kg/m²], overweight [25.0–30.0 kg/m²], obese [≥ 30.0 kg/m²], and unknown); education levels (≤ 10 , 11–12, >12 , and unknown); blood pressure (systolic blood pressure < 140 mmHg and diastolic blood pressure < 90 mmHg, systolic blood pressure ≥ 140 mmHg or diastolic blood pressure ≥ 90 mmHg, and unknown); fasting blood glucose (<100 , ≥ 100 mg/dL, and unknown); smoking status (non- and current smoker, and unknown); alcohol consumption (every day, sometimes, rarely days per week, and unknown); aerobic physical activity (low, moderate, high, and unknown); previous history of cardiovascular disease, chronic kidney disease, and chronic obstructive pulmonary disease; history of medication use for diabetes mellitus, dyslipidemia, and hypertension; and missing indicators (household income missing indicator [yes or no], townsend deprivation index missing indicator [yes or no], race missing indicator [yes or no], education levels [yes or no], obesity missing indicator [yes or no], blood pressure missing indicator [yes or no], fasting blood glucose missing indicator [yes or no], serum total cholesterol missing indicator [yes or no], glomerular filtration rate missing indicator [yes or no], smoking status missing indicator [yes or no], alcoholic drinks missing indicator [yes or no], and aerobic physical activity missing indicator [yes or no]).

Table S22. The HR with 95% CI for the long-term sequelae risk of incident asthma without dyspnea following COVID-19 diagnosis of patients in the propensity score-matched main cohort (South Korea), replication cohort A (Japan), and replication cohort B (UK).

Cohort	Exposure	n (%)	Incidence rate*	HR (95% CI)	
				Model 1 ^a	Model 2 ^b
Allergic diseases					
Main cohort	None	675,760 (82.48)	23.7	1.00 (reference)	1.00 (reference)
Main cohort	Patients with COVID-19	143,563 (17.52)	27.6	1.16 (1.10 to 1.23)	1.16 (1.10 to 1.24)
Replication cohort A	None	1,989,799 (78.94)	27.3	1.00 (reference)	1.00 (reference)
Replication cohort A	Patients with COVID-19	530,924 (21.06)	74.5	2.64 (2.60 to 2.68)	2.54 (2.51 to 2.58)
Replication cohort B	None	248,936 (79.40)	7.2	1.00 (reference)	1.00 (reference)
Replication cohort B	Patients with COVID-19	76,886 (23.60)	8.2	1.13 (1.06 to 1.22)	1.12 (1.04 to 1.20)
Asthma					
Main cohort	None	675,760 (82.48)	0.8	1.00 (reference)	1.00 (reference)
Main cohort	Patients with COVID-19	143,563 (17.52)	1.8	2.13 (1.65 to 2.75)	2.12 (1.65 to 2.74)
Replication cohort A	None	1,989,799 (78.94)	2.6	1.00 (reference)	1.00 (reference)
Replication cohort A	Patients with COVID-19	530,924 (21.06)	7.5	2.74 (2.62 to 2.87)	2.61 (2.49 to 2.74)
Replication cohort B	None	248,936 (79.40)	2.7	1.00 (reference)	1.00 (reference)
Replication cohort B	Patients with COVID-19	76,886 (23.60)	3.7	1.15 (1.04 to 1.28)	1.13 (1.02 to 1.26)

BMI, body mass index; CI, confidence interval; HR, hazard ratio.

* Incidence rate is expressed as per 1,000 person-years.

The data in bold indicate significant differences ($P < 0.05$).

^a **Model 1:** Adjusted for age (20–39, 40–59, and ≥ 60 years) and sex.

^b **Model 2 (main cohort):** Adjusted for age (20–39, 40–59, and ≥ 60 years); sex; household income (low income, middle income, and high income); region of residence (urban and rural); Charlson comorbidity index (0, 1, and ≥ 2); BMI (underweight [$< 18.5 \text{ kg/m}^2$], normal [18.5 –

23.0 kg/m²], overweight [23.0–25.0 kg/m²], obese [\geq 25.0 kg/m²], and unknown); blood pressure (systolic blood pressure <140 mmHg and diastolic blood pressure <90 mmHg, systolic blood pressure \geq 140 mmHg or diastolic blood pressure \geq 90 mmHg, and unknown); fasting blood glucose (<100, \geq 100 mg/dL, and unknown); serum total cholesterol (<200, 200–240, \geq 240 mg/dL, and unknown); glomerular filtration rate (<60, 60–90, \geq 90 mL/min/1.73 m², and unknown); smoking status (non-, ex-, current smoker, and unknown); alcoholic drinks (<1, 1–2, 3–4, \geq 5 days per week, and unknown); aerobic physical activity (sufficient, insufficient, and unknown); previous history of cardiovascular disease, chronic kidney disease, and chronic obstructive pulmonary disease; history of medication use for diabetes mellitus, dyslipidemia, and hypertension; and missing indicators (BMI missing indicator [yes or no], blood pressure missing indicator [yes or no], fasting blood glucose missing indicator [yes or no], serum total cholesterol missing indicator [yes or no], glomerular filtration rate missing indicator [yes or no], smoking status missing indicator [yes or no], alcoholic drinks missing indicator [yes or no], and aerobic physical activity missing indicator [yes or no]).

^b **Model 2 (replication cohort A):** Adjusted for age (20–39, 40–59, and \geq 60 years); sex; Charlson comorbidity index (0, 1, and \geq 2); BMI (underweight [<18.5 kg/m²], normal [18.5–23.0 kg/m²], overweight [23.0–25.0 kg/m²], obese [\geq 25.0 kg/m²], and unknown); blood pressure (systolic blood pressure <140 mmHg and diastolic blood pressure <90 mmHg, systolic blood pressure \geq 140 mmHg or diastolic blood pressure \geq 90 mmHg, and unknown); fasting blood glucose (<100, \geq 100 mg/dL, and unknown); serum total cholesterol (<200, 200–240, \geq 240 mg/dL, and unknown); glomerular filtration rate (<60, 60–90, \geq 90 mL/min/1.73 m², and unknown); smoking status (non- and current smoker, and unknown); alcoholic drinks (drinks; <1, 1–2, 3–4, \geq 5 days per week, and unknown); aerobic physical activity (sufficient, insufficient, and unknown); previous history of cardiovascular disease, chronic kidney disease, and chronic obstructive pulmonary disease; history of medication use for diabetes mellitus, dyslipidemia, and hypertension; and missing indicators (BMI missing indicator [yes or no], blood pressure missing indicator [yes or no], fasting blood glucose missing indicator [yes or no], serum total cholesterol missing indicator [yes or no], glomerular filtration rate missing indicator [yes or no], smoking status missing indicator [yes or no], alcoholic drinks missing indicator [yes or no], and aerobic physical activity missing indicator [yes or no]).

^b **Model 2 (replication cohort B):** Adjusted for age (20–39, 40–59, and \geq 60 years); sex; household income (<£18,000, £18,000–£30,999, £31,000–£51,999, £52,000–£100,000, >£100,000), and unknown); region of residence (urban and rural); townsend deprivation index (T1 [least deprived], T2, T3 [most deprived], and unknown); race (white, mixed, Asian, black, others, and unknown); Charlson comorbidity index (0, 1, and \geq 2); BMI (normal [<25.0 kg/m²], overweight [25.0–30.0 kg/m²], obese [\geq 30.0 kg/m²], and unknown); education levels (\leq 10, 11–12, >12,

and unknown); blood pressure (systolic blood pressure < 140 mmHg and diastolic blood pressure < 90 mmHg, systolic blood pressure \geq 140 mmHg or diastolic blood pressure \geq 90 mmHg, and unknown); fasting blood glucose (<100, \geq 100 mg/dL, and unknown); smoking status (non- and current smoker, and unknown); alcohol consumption (every day, sometimes, rarely days per week, and unknown); aerobic physical activity (low, moderate, high, and unknown); previous history of cardiovascular disease, chronic kidney disease, and chronic obstructive pulmonary disease; history of medication use for diabetes mellitus, dyslipidemia, and hypertension; and missing indicators (household income missing indicator [yes or no], townsend deprivation index missing indicator [yes or no], race missing indicator [yes or no], education levels [yes or no], obesity missing indicator [yes or no], blood pressure missing indicator [yes or no], fasting blood glucose missing indicator [yes or no], serum total cholesterol missing indicator [yes or no], glomerular filtration rate missing indicator [yes or no], smoking status missing indicator [yes or no], alcoholic drinks missing indicator [yes or no], and aerobic physical activity missing indicator [yes or no]).

Table S23. The HR with 95% CI for the long-term sequelae risk of incident asthma with EDHO following COVID-19 diagnosis of patients in the propensity score-matched main cohort (South Korea).

Exposure	n (%)	Incidence rate*	HR (95% CI)	
			Model 1 ^a	Model 2 ^b
Allergic diseases				
None	688,340 (82.32)	24.0	1.00 (reference)	1.00 (reference)
Patients with COVID-19	147,824 (17.68)	28.7	1.20 (1.13 to 1.27)	1.20 (1.13 to 1.27)
Asthma				
None	688,340 (82.32)	0.4	1.00 (reference)	1.00 (reference)
Patients with COVID-19	147,824 (17.68)	1.2	2.86 (2.08 to 3.94)	2.74 (1.99 to 3.78)

BMI, body mass index; CI, confidence interval; EDHO, history of emergency department visits or hospitalizations; HR, hazard ratio.

* Incidence rate is expressed as per 1,000 person-years.

The data in bold indicate significant differences ($P < 0.05$).

^a **Model 1:** Adjusted for age (20–39, 40–59, and ≥ 60 years) and sex.

^b **Model 2 (South Korea):** Adjusted for age (20–39, 40–59, and ≥ 60 years); sex; household income (low income, middle income, and high income); region of residence (urban and rural); Charlson comorbidity index (0, 1, and ≥ 2); BMI (underweight [$< 18.5 \text{ kg/m}^2$], normal [$18.5\text{--}23.0 \text{ kg/m}^2$], overweight [$23.0\text{--}25.0 \text{ kg/m}^2$], obese [$\geq 25.0 \text{ kg/m}^2$], and unknown); blood pressure (systolic blood pressure $< 140 \text{ mmHg}$ and diastolic blood pressure $< 90 \text{ mmHg}$, systolic blood pressure $\geq 140 \text{ mmHg}$ or diastolic blood pressure $\geq 90 \text{ mmHg}$, and unknown); fasting blood glucose (< 100 , $\geq 100 \text{ mg/dL}$, and unknown); serum total cholesterol (< 200 , $200\text{--}240$, $\geq 240 \text{ mg/dL}$, and unknown); glomerular filtration rate (< 60 , $60\text{--}90$, $\geq 90 \text{ mL/min/1.73 m}^2$, and unknown); smoking status (non-, ex-, current smoker, and unknown); alcoholic drinks (< 1 , 1–2, 3–4, ≥ 5 days per week, and unknown); aerobic physical activity (sufficient, insufficient, and unknown); previous history of cardiovascular disease, chronic kidney disease, and chronic obstructive pulmonary disease; history of medication use for diabetes mellitus, dyslipidemia, and

hypertension; and missing indicators (BMI missing indicator [yes or no], blood pressure missing indicator [yes or no], fasting blood glucose missing indicator [yes or no], serum total cholesterol missing indicator [yes or no], glomerular filtration rate missing indicator [yes or no], smoking status missing indicator [yes or no], alcoholic drinks missing indicator [yes or no], and aerobic physical activity missing indicator [yes or no]).

Comment 2.

The incidence in the non-COVID groups varies between the three cohorts. Especially UK Biobank reports a much higher onset of diseases. Due to an older population it should be less. This fact is likely due to different data collection (patient reported diseases vs. physician coded diseases). The Clinical Practice Research Datalink (CPRD) or THIN would be the comparably cohort for the United Kingdom.

Response:

Thank you for your insightful comments on the fundamental issues. As you mentioned, there are significant differences in data collection methodology, with the K-COV-N and JMDC data relating to insurance claims (physician-coded disease), whereas the UK Biobank data consists of self-reported information (patient-reported disease). It is crucial to recognize the difference, especially in the context of conducting multinational population studies, and we appreciate your acknowledgment of this important point.

We are aware of the impact that the reporting format of diseases could have on the observed incidence rates. Therefore, although we reviewed the suggested datasets (Clinical Practice Research Datalink, CPRD, or THIN), we did not obtain access to these specific datasets. We would like to request your understanding as the acquisition and analysis of this data in the subsequent phases are impractical within the confines of the revision period. Nevertheless, we believe that the current results are meaningful. Despite different methods of data collecting, the results were aligned, suggesting concordance in their outcomes. It may help to reinforce the robustness of the current study. Also, a prior study conducted analyses using two independent cohorts with different reporting formats and reached similar conclusions.

- Woo A, Lee SW, Koh HY, Kim MA, Han MY, Yon DK. Incidence of cancer after asthma development: **2 independent population-based cohort studies.** J Allergy Clin

Immunol. 2021 Jan;147(1):135-143. doi: 10.1016/j.jaci.2020.04.041. Epub 2020 May 15. PMID: 32417133.

Data: National Health Insurance Service–National Sample Cohort (claims-based data); the Ansan-Ansung cohort (interview-based data).

→ **Interpretation:** The claims-based definition of asthma in the NHIS-NSC led to an incidence of 1.47%, and the interview-based (physician-diagnosed) definition in the Ansan-Ansung cohort led to an incidence of 0.94%. **Thus, these 2 cohorts, which used different asthma definitions, had similar incidences of asthma.**

However, given the evident constraints in comprehensively interpreting the results, we addressed this discussion in the "Limitations and Strengths" section and included more comprehensive information about the main cohort (K-COV-N) in method and replication cohorts (JMDC and UK Biobank) in the supplementary material.

Changes in text:

Methods, *Data source*

Both the K-COV-N and JMDC cohorts employ a universal health insurance system. The UKB, meanwhile, is a dataset comprised of voluntary participation, including biomedical samples and health information. Detailed explanations of the JMDC and UKB cohorts can be found in supplemental material section.

Methods, *K-COV-N cohort (main)*

The K-COV-N cohort is a large-scale, nationwide, general population-based cohort in South Korea, covering 98% of the South Korean population.¹ The cohort was developed and provided by the National Health Insurance Service of South Korea (NHIS) and Korea Disease Control

and Prevention Agency (KDCA) focused on individuals aged ≥ 20 years between January 1, 2018, and December 31, 2021.

Discussion, *Limitations and strengths*

Seventh, we used three cohorts with different reporting formats (self-report for the UK Biobank cohort and insurance claims for the K-COV-N and JMDC cohorts). However, the results were aligned with one another, which rather strengthens the robustness of the study.

Supplement material. Explanation of replication cohorts (Japan and the UK).

JMDC (replication cohort A)

Data source

Japan has health insurance provided by the universal insurance system.¹ The JMDC has contracts with over 60 insurance providers and includes health insurance claims records data of insured individuals who are primarily employees of relatively large companies in Japan.² JMDC has created a database, using data collected from medical institutions in Japan, consisting of patient-level data (unique identifier, family identifiers, relationship to the insured individual, age, sex) and claims for inpatient and outpatient treatment (disease class according to International Classification of Diseases [ICD]-10 code, prescribed drugs based on Anatomical Therapeutic Chemical class, and drug dosage form), diagnosis or therapeutic procedure, institutional information (hospital character), and health checkup (i.e., body mass index, blood pressure, clinical laboratory test, medication status, and self-administered questionnaire such as smoking status, alcohol consumption, and physical activity).³

For more information, please see this webpage (<https://www.jmdc.co.jp/en/jmdc-claims->

database/).

UK Biobank (replication cohort B)

Data source

The UK Biobank, which is one of the datasets that rely on voluntary participation, is a globally recognized institution providing unique and extensive data for large-scale research through biomedical samples and health information.⁴ Launched in 2006 in the UK, this project has engaged numerous researchers who utilize participant health data and biological samples for diverse medical studies.⁵ With over 500,000 participants recruited nationwide, the cohort represents rich diversity in terms of age, race, gender, and geographical location, ensuring the applicability of research outcomes to diverse populations.⁶ Participants have provided a range of health-related data, including health surveys, physical measurements, blood pressure readings, and the provision of blood and urine samples. This invaluable data is employed in research across various fields, such as genomic analysis, cardiovascular and metabolic diseases, and cancer research. Genomic data collected from participants contribute to studying genetic traits and variations, providing crucial information for understanding genetic contributions and exploring interactions between genes and environmental factors.⁷

For more information, please see this webpage (<https://www.ukbiobank.ac.uk/learn-more-about-uk-biobank>).

[REDACTED]

Supplement references

1. Reich MR, Ikegami N, Shibuya K, Takemi K. 50 years of pursuing a healthy society in Japan. *Lancet* 378, 1051-1053 (2011).
2. Setogawa N, Ohbe H, Isogai T, Matsui H, Yasunaga H. Characteristics and short-term outcomes of outpatient and inpatient cardiac catheterizations: A descriptive study using a nationwide claim database in Japan. *J Cardiol* 82, 201-206 (2023).
3. Kaneko H, et al. Medication-Naïve Blood Pressure and Incident Cancers: Analysis of 2 Nationwide Population-Based Databases. *Am J Hypertens* 35, 731-739 (2022).
4. Keyes KM, Westreich D. UK Biobank, big data, and the consequences of non-representativeness. *Lancet (London, England)* 393, 1297 (2019).

5. Backman JD, et al. Exome sequencing and analysis of 454,787 UK Biobank participants. *Nature* 599, 628-634 (2021).

6. Bycroft C, et al. The UK Biobank resource with deep phenotyping and genomic data. *Nature* 562, 203-209 (2018).

7. Ganna A, Ingelsson E. 5 year mortality predictors in 498,103 UK Biobank participants: a prospective population-based study. *Lancet (London, England)* 386, 533-540 (2015).

Comment 3.

The definition of “obese” is odd, as obese is defined by the WHO as an BMI above 30 not above 25. Also, overweight would be an BMI above 25 not above 23. This classification could likely result in residual confounding.

Response:

We would like to express our sincere gratitude for your invaluable feedback. Taking your comment into consideration, we have re-evaluated the classification criteria for body mass index (BMI) in adults. The average BMI for each country is as follows: South Korea (male, 24.2; female, 23.2 [kg/m²]), Japan (male, 23.6; female, 21.7 [kg/m²]), and the UK (male, 27.4; female, 27.0 [kg/m²])¹. BMI is a variable that can be influenced by ethnic characteristics and regional differences in each country. Therefore, to better reflect the biological characteristics of the study participants, we decided to utilize BMI criteria based on the Asia Pacific guidelines for South Korea and Japan and the World Health Organization guidelines for the UK.

- The Asia-Pacific guidelines [South Korea and Japan]: underweight, <18.5; normal, 18.5–23.0; overweight, 23.0–25.0; and obese, ≥ 25.0 (kg/m²)^{2,3}
- The World Health Organization guidelines [the UK]: underweight, <18.5; normal, 18.5–25.0; overweight, 25.0–30.0; and obese, ≥ 30.0 (kg/m²)⁴

Using the criteria outlined above, we refined the BMI standards, conducted a reanalysis, and incorporated all pertinent changes into the manuscript. We believe this correction enhances the precision and clarity of our manuscript.

<Reference>

1. NCD Risk Factor Collaboration (NCD-RisC). Trends in adult body-mass index in 200 countries from 1975 to 2014: a pooled analysis of 1698 population-based measurement studies

with 19.2 million participants. *Lancet*. 2016 Apr 2;387(10026):1377-1396. doi: 10.1016/S0140-6736(16)30054-X. Erratum in: *Lancet*. 2016 May 14;387(10032):1998. PMID: 27115820; PMCID: PMC7615134.

2. Park S, Kim HJ, Kim S, Rhee SY, Woo HG, Lim H, Cho W, Yon DK. National Trends in Physical Activity Among Adults in South Korea Before and During the COVID-19 Pandemic, 2009-2021. *JAMA Netw Open*. 2023 Jun 1;6(6):e2316930. doi: 10.1001/jamanetworkopen.2023.16930. PMID: 37273204; PMCID: PMC10242425.

3. Oze I, Ito H, Koyanagi YN, Abe SK, Rahman MS, Islam MR, Saito E, Gupta PC, Sawada N, Tamakoshi A, Shu XO, Sakata R, Malekzadeh R, Tsuji I, Kim J, Nagata C, You SL, Park SK, Yuan JM, Shin MH, Kweon SS, Pednekar MS, Tsugane S, Kimura T, Gao YT, Cai H, Pourshams A, Lu Y, Kanemura S, Wada K, Sugawara Y, Chen CJ, Chen Y, Shin A, Wang R, Ahn YO, Shin MH, Ahsan H, Boffetta P, Chia KS, Qiao YL, Rothman N, Zheng W, Inoue M, Kang D, Matsuo K. Obesity is associated with biliary tract cancer mortality and incidence: A pooled analysis of 21 cohort studies in the Asia Cohort Consortium. *Int J Cancer*. 2023 Nov 15. doi: 10.1002/ijc.34794. Epub ahead of print. PMID: 37966009.

4. Zhu J, Ge F, Zeng Y, Qu Y, Chen W, Yang H, Yang L, Fang F, Song H. Physical and Mental Activity, Disease Susceptibility, and Risk of Dementia: A Prospective Cohort Study Based on UK Biobank. *Neurology*. 2022 Aug 23;99(8):e799-e813. doi: 10.1212/WNL.0000000000200701. Epub 2022 Jul 27. PMID: 35896434; PMCID: PMC9484730.

Changes in text:

Table 2. The HR with 95% CI for the long-term sequelae risk of incident allergic diseases following COVID-19 diagnosis of patients in the propensity score-matched cohorts in **main cohort** (South Korea), **replication cohort A** (Japan), and **replication cohort B** (UK).

Cohort	Exposure	n (%)	Incidence rate*	HR (95% CI)	
				Model 1 ^a	Model 1 ^a
Allergic diseases					
Main cohort	None	688,340 (82.32)	24.0	1.00 (reference)	1.00 (reference)
Main cohort	Patients with COVID-19	147,824 (17.68)	28.7	1.20 (1.13 to 1.27)	1.20 (1.13 to 1.27)
Replication cohort A	None	1,998,524 (78.65)	27.4	1.00 (reference)	1.00 (reference)
Replication cohort A	Patients with COVID-19	542,497 (21.35)	75.1	2.65 (2.61 to 2.69)	2.56 (2.52 to 2.59)
Replication cohort B	None	248,949 (76.40)	7.2	1.00 (reference)	1.00 (reference)
Replication cohort B	Patients with COVID-19	76,894 (23.60)	8.3	1.14 (1.06 to 1.22)	1.12 (1.04 to 1.20)
Asthma					
Main cohort	None	688,340 (82.32)	1.0	1.00 (reference)	1.00 (reference)
Main cohort	Patients with COVID-19	147,824 (17.68)	2.2	2.27 (1.81 to 2.85)	2.25 (1.80 to 2.83)
Replication cohort A	None	1,998,524 (78.65)	2.7	1.00 (reference)	1.00 (reference)
Replication cohort A	Patients with COVID-19	542,497 (21.35)	7.6	2.77 (2.65 to 2.90)	2.63 (2.51 to 2.75)
Replication cohort B	None	248,949 (76.40)	3.2	1.00 (reference)	1.00 (reference)
Replication cohort B	Patients with COVID-19	76,894 (23.60)	3.8	1.16 (1.05 to 1.29)	1.14 (1.03 to 1.27)
Allergic rhinitis					
Main cohort	None	688,340 (82.32)	17.3	1.00 (reference)	1.00 (reference)
Main cohort	Patients with COVID-19	147,824 (17.68)	21.3	1.23 (1.15 to 1.32)	1.23 (1.15 to 1.32)

Replication cohort A	None	1,998,524 (78.65)	20.4	1.00 (reference)	1.00 (reference)
Replication cohort A	Patients with COVID-19	542,497 (21.35)	62.6	2.98 (2.93 to 3.03)	2.88 (2.83 to 2.93)
Replication cohort B	None	248,949 (76.40)	1.1	1.00 (reference)	1.00 (reference)
Replication cohort B	Patients with COVID-19	76,894 (23.60)	1.4	1.21 (1.02 to 1.44)	1.20 (1.01 to 1.43)

Atopic dermatitis

Main cohort	None	688,340 (82.32)	2.6	1.00 (reference)	1.00 (reference)
Main cohort	Patients with COVID-19	147,824 (17.68)	3.0	1.15 (0.96 to 1.37)	1.15 (0.96 to 1.37)
Replication cohort A	None	1,998,524 (78.65)	5.2	1.00 (reference)	1.00 (reference)
Replication cohort A	Patients with COVID-19	542,497 (21.35)	6.8	1.25 (1.20 to 1.30)	1.21 (1.16 to 1.26)
Replication cohort B	None	248,949 (76.40)	0.04	1.00 (reference)	1.00 (reference)
Replication cohort B	Patients with COVID-19	76,894 (23.60)	0.05	1.18 (0.46 to 2.99)	1.19 (0.47 to 3.01)

Food allergy

Main cohort	None	688,340 (82.32)	3.6	1.00 (reference)	1.00 (reference)
Main cohort	Patients with COVID-19	147,824 (17.68)	3.0	0.85 (0.71 to 1.01)	0.85 (0.71 to 1.00)
Replication cohort A	None	1,998,524 (78.65)	1.3	1.00 (reference)	1.00 (reference)
Replication cohort A	Patients with COVID-19	542,497 (21.35)	2.7	2.10 (1.95 to 2.25)	1.84 (1.71 to 1.98)
Replication cohort B	None	248,949 (76.40)	3.1	1.00 (reference)	1.00 (reference)
Replication cohort B	Patients with COVID-19	76,894 (23.60)	3.4	1.09 (0.97 to 1.21)	1.07 (0.96 to 1.20)

BMI, body mass index; CI, confidence interval; HR, hazard ratio.

* Incidence rate is expressed as per 1,000 person-years.

The data in bold indicate significant differences ($P < 0.05$).

^a **Model 1**: Adjusted for age (20–39, 40–59, and ≥ 60 years) and sex.

^b **Model 2 (main cohort)**: Adjusted for age (20–39, 40–59, and ≥ 60 years); sex; household income (low income, middle income, and high income); region of residence (urban and rural); Charlson comorbidity index (0, 1, and ≥ 2); BMI (underweight [$< 18.5 \text{ kg/m}^2$], normal [$18.5\text{--}23.0 \text{ kg/m}^2$], overweight [$23.0\text{--}25.0 \text{ kg/m}^2$], obese [$\geq 25.0 \text{ kg/m}^2$], and unknown); blood pressure (systolic blood pressure $< 140 \text{ mmHg}$ and diastolic blood pressure $< 90 \text{ mmHg}$, systolic blood pressure $\geq 140 \text{ mmHg}$ or diastolic blood pressure $\geq 90 \text{ mmHg}$, and unknown); fasting blood glucose (< 100 , $\geq 100 \text{ mg/dL}$, and unknown); serum total cholesterol (< 200 , $200\text{--}240$, $\geq 240 \text{ mg/dL}$, and unknown); glomerular filtration rate (< 60 , $60\text{--}90$, $\geq 90 \text{ mL/min/1.73 m}^2$, and unknown); smoking status (non-, ex-, current smoker, and unknown); alcoholic drinks (< 1 , $1\text{--}2$, $3\text{--}4$, ≥ 5 days per week, and unknown); aerobic physical activity (sufficient, insufficient, and unknown); previous history of cardiovascular disease, chronic kidney disease, and chronic obstructive pulmonary disease; history of medication use for diabetes mellitus, dyslipidemia, and hypertension; and missing indicators (BMI missing indicator [yes or no], blood pressure missing indicator [yes or no], fasting blood glucose missing indicator [yes or no], serum total cholesterol missing indicator [yes or no], glomerular filtration rate missing indicator [yes or no], smoking status missing indicator [yes or no], alcoholic drinks missing indicator [yes or no], and aerobic physical activity missing indicator [yes or no]).

^b **Model 2 (replication cohort A)**: Adjusted for age (20–39, 40–59, and ≥ 60 years); sex; Charlson comorbidity index (0, 1, and ≥ 2); BMI (underweight [$< 18.5 \text{ kg/m}^2$], normal [$18.5\text{--}23.0 \text{ kg/m}^2$], overweight [$23.0\text{--}25.0 \text{ kg/m}^2$], obese [$\geq 25.0 \text{ kg/m}^2$], and unknown); blood pressure (systolic blood pressure $< 140 \text{ mmHg}$ and diastolic blood pressure $< 90 \text{ mmHg}$, systolic blood pressure $\geq 140 \text{ mmHg}$ or diastolic blood pressure $\geq 90 \text{ mmHg}$, and unknown); fasting blood glucose (< 100 , $\geq 100 \text{ mg/dL}$, and unknown); serum total cholesterol (< 200 , $200\text{--}240$, $\geq 240 \text{ mg/dL}$, and unknown); glomerular filtration rate (< 60 , $60\text{--}90$, $\geq 90 \text{ mL/min/1.73 m}^2$, and unknown); smoking status (non- and current smoker, and unknown); alcoholic drinks (drinks; < 1 , $1\text{--}2$, $3\text{--}4$, ≥ 5 days per week, and unknown); aerobic physical activity (sufficient, insufficient, and unknown); previous history of cardiovascular disease, chronic kidney disease, and chronic obstructive pulmonary disease; history of medication use for diabetes mellitus, dyslipidemia, and hypertension; and missing indicators (BMI missing indicator [yes or no], blood pressure missing indicator [yes or no], fasting blood glucose missing indicator [yes or no], serum total cholesterol missing indicator [yes or no], glomerular

filtration rate missing indicator [yes or no], smoking status missing indicator [yes or no], alcoholic drinks missing indicator [yes or no], and aerobic physical activity missing indicator [yes or no]).

^b **Model 2 (replication cohort B):** Adjusted for age (20–39, 40–59, and ≥ 60 years); sex; household income (<£18,000, £18,000–£30,999, £31,000–£51,999, £52,000–£100,000, >£100,000), and unknown); region of residence (urban and rural); townsend deprivation index (T1 [least deprived], T2, T3 [most deprived], and unknown); ethnicity (white, mixed, Asian, black, others, and unknown); Charlson comorbidity index (0, 1, and ≥ 2); BMI (normal [$<25.0 \text{ kg/m}^2$], overweight [$25.0\text{--}30.0 \text{ kg/m}^2$], obese [$\geq 30.0 \text{ kg/m}^2$], and unknown); education levels (≤ 10 , 11–12, >12 , and unknown); blood pressure (systolic blood pressure $< 140 \text{ mmHg}$ and diastolic blood pressure $< 90 \text{ mmHg}$, systolic blood pressure $\geq 140 \text{ mmHg}$ or diastolic blood pressure $\geq 90 \text{ mmHg}$, and unknown); fasting blood glucose (<100 , $\geq 100 \text{ mg/dL}$, and unknown); smoking status (non- and current smoker, and unknown); alcohol consumption (every day, sometimes, rarely days per week, and unknown); aerobic physical activity (low, moderate, high, and unknown); previous history of cardiovascular disease, chronic kidney disease, and chronic obstructive pulmonary disease; history of medication use for diabetes mellitus, dyslipidemia, and hypertension; and missing indicators (household income missing indicator [yes or no], townsend deprivation index missing indicator [yes or no], ethnicity missing indicator [yes or no], education levels [yes or no], obesity missing indicator [yes or no], blood pressure missing indicator [yes or no], fasting blood glucose missing indicator [yes or no], serum total cholesterol missing indicator [yes or no], glomerular filtration rate missing indicator [yes or no], smoking status missing indicator [yes or no], alcoholic drinks missing indicator [yes or no], and aerobic physical activity missing indicator [yes or no]).

Figure S3. Density plot and box plot for before matching and after matching in patients with COVID-19 and non- COVID-19 in replication cohort B (UK)

Table S5. Baseline characteristics for 1:5 propensity score-matched cohort in replication cohort B (UK, n=325,843).

Covariates	1:5 matched cohort (n=325,843)			SMD
	Total	COVID-19 (n=76,894)	Non-COVID-19 (n=248,949)	
Age, years, mean (SD)	71.08 (7.87)	71.33 (7.58)	71.00 (7.95)	0.048
Age, years, n (%)				0.048
40–59	30,577 (9.38)	6,726 (8.75)	23,851 (9.58)	
≥ 60	295,266 (90.62)	70,168 (91.25)	225,098 (90.42)	
Sex, n (%)				0.043
Male	186,007 (57.08)	31,764 (41.31)	108,072 (43.41)	
Female	139,836 (42.92)	45,130 (58.69)	140,877 (56.59)	
Region of residence				0.014
Urban	277,445 (85.15)	65,179 (84.76)	212,266 (85.26)	
Rural	48,398 (14.85)	11,715 (15.24)	36,683 (14.74)	
Household income, n (%)				0.186
Level1 (<£18,000)	53,572 (16.44)	10,311 (13.41)	43,261 (17.38)	
Level2 (£18,000-£30,999)	70,524 (21.64)	16,209 (21.08)	54,315 (21.82)	
Level3 (£31,000-£51,999)	76,821 (23.58)	19,028 (24.75)	57,793 (23.21)	
Level4 (£52,000-£100,000)	62,610 (19.21)	16,916 (22)	45,694 (18.35)	
Level5 (>£100,000)	17,314 (5.31)	5,177 (6.73)	12,137 (4.88)	
Unknown	45,002 (13.81)	9,253 (12.03)	35,749 (14.36)	

Townsend deprivation index, mean (SD)	-1.43 (3.02)	-1.73 (2.83)	-1.35 (3.06)	0.126
Townsend deprivation index tertile				0.143
T1 (last deprived)	118,547 (36.38)	28,477 (37.03)	90,070 (36.18)	
T2	112,207 (34.44)	26,375 (34.30)	85,832 (34.48)	
T3 (most deprived)	94,822 (29.1)	21,973 (28.58)	72,849 (29.26)	
Unknown	267 (0.08)	69 (0.09)	198 (0.08)	
Charlson comorbidity index, n (%)				0.064
0	311,545 (95.61)	72,715 (94.57)	238,830 (95.94)	
1	6,762 (2.08)	2,008 (2.61)	4,754 (1.91)	
≥2	7,536 (2.31)	2,171 (2.82)	5,365 (2.16)	
Race, n (%)				<0.001
White	313,685 (96.27)	74,072 (96.33)	239,613 (96.25)	
Mixed	2,495 (0.77)	599 (0.78)	1,896 (0.76)	
Asian	3,874 (1.19)	915 (1.19)	2,959 (1.19)	
Black	2,710 (0.83)	589 (0.77)	2,121 (0.85)	
Others	1,957 (0.60)	449 (0.58)	1,508 (0.61)	
Unknown	1,122 (0.34)	270 (0.35)	852 (0.34)	
Education levels, years, n (%)				0.142
≤10	176 (0.05)	44 (0.06)	132 (0.05)	
11-12	9,090 (2.79)	2,261 (2.94)	6,829 (2.74)	
>12	314,338 (96.47)	74,063 (96.32)	240,275 (96.52)	
Unknown	2,239 (0.69)	526 (0.68)	1,713 (0.69)	
History of cardiovascular disease, n (%)	32,179 (9.88)	8,264 (10.75)	23,915 (9.61)	0.038
History of chronic kidney disease, n (%)	17,274 (5.30)	4,849 (6.31)	12,425 (4.99)	0.057
History of chronic obstructive pulmonary disease, n (%)	6,330 (1.94)	2,051 (2.67)	4,279 (1.72)	0.065
History of medication use for diabetes	586 (0.18)	148 (0.19)	438 (0.18)	0.004
History of medication use for hyperlipidemia	10,609 (3.26)	2,551 (3.32)	8,058 (3.24)	0.005

History of medication use for hypertension	30,180 (9.26)	7,164 (9.32)	23,016 (9.25)	0.003
Body mass index, kg/m ² , n (%)				0.022
Normal (<25.0)	121,657 (37.34)	29,580 (38.47)	92,077 (36.99)	
Overweight (25.0–30.0)	140,093 (42.99)	32,341 (42.06)	107,752 (43.28)	
Obese (≥30.0)	63,108 (19.37)	14,740 (19.17)	48,368 (19.43)	
Unknown	985 (0.30)	233 (0.30)	752 (0.30)	
Blood pressure, n (%)				<0.001
SBP <140 mmHg and DBP <90 mmHg	167 (0.05)	42 (0.05)	125 (0.05)	
SBP ≥140 mmHg or DBP ≥90 mmHg	308,228 (94.59)	72,857 (94.75)	235,371 (94.55)	
Unknown	17,448 (5.35)	3,995 (5.20)	13,453 (5.40)	
Fasting blood glucose, mg/dL, n (%)				0.039
<100	241,863 (74.23)	56,841 (73.92)	185,022 (74.32)	
≥100	40,329 (12.38)	9688 (12.60)	30,641 (12.31)	
Unknown	43,651 (13.4)	10,365 (13.48)	33,286 (13.37)	
Smoking status, n (%)				<0.001
Current smoker	187,176 (57.44)	44,105 (57.36)	143,071 (57.47)	
Non-smoker	137,877 (42.31)	32,598 (42.39)	105,279 (42.29)	
Unknown	790 (0.24)	191 (0.25)	599 (0.24)	
Alcoholic drinks, days per week, n (%)				0.029
Every day	50,535 (15.51)	11,813 (15.36)	38,722 (15.55)	
Sometimes	120,935 (37.11)	28,101 (36.55)	92,834 (37.29)	
Rarely	154,024 (47.27)	36,901 (47.99)	117,123 (47.05)	
Unknown	349 (0.11)	79 (0.10)	270 (0.11)	
Aerobic physical activity, n (%)				0.066
Low	94,783 (29.09)	22,269 (28.96)	72,514 (29.13)	
Moderate	113,035 (23.07)	27,018 (35.14)	86,017 (34.55)	

High	108,836 (34.69)	25,414 (33.05)	83,422 (33.51)
Unknown	9189 (33.4)	2193 (2.85)	6996 (2.81)

DBP, diastolic blood pressure; SBP, systolic blood pressure; SD, standard deviation; SMD, standardized mean difference.

Supplement material. Explanation of replication cohorts (Japan and the UK).

JMDC (replication cohort A)

Covariates

The demographic characteristics of the participants were obtained from the health JMDC database as followings: age (20–39, 40–59, and ≥ 60 years), sex, Charlson comorbidity index (0, 1, and ≥ 2), BMI (underweight [$< 18.5 \text{ kg/m}^2$], normal [$18.5\text{--}23.0 \text{ kg/m}^2$], overweight [$23.0\text{--}25.0 \text{ kg/m}^2$], obese [$\geq 25.0 \text{ kg/m}^2$], and unknown), blood pressure (systolic blood pressure < 140 mmHg and diastolic blood pressure < 90 mmHg, systolic blood pressure ≥ 140 mmHg or diastolic blood pressure ≥ 90 mmHg, and unknown), fasting blood glucose (< 100 , ≥ 100 mg/dL, and unknown), serum total cholesterol (< 200 , $200\text{--}240$, ≥ 240 mg/dL, and unknown), glomerular filtration rate (< 60 , $60\text{--}90$, ≥ 90 mL/min/1.73 m², and unknown), smoking status (non- and current smoker, and unknown), alcoholic drinks (drinks; < 1 , $1\text{--}2$, $3\text{--}4$, ≥ 5 days per week, and unknown), aerobic physical activity (sufficient, insufficient, and unknown), previous history of cardiovascular disease, chronic kidney disease, and chronic obstructive pulmonary disease, history of medication use for diabetes mellitus, dyslipidemia, and hypertension, and missing indicators (BMI missing indicator [yes or no], blood pressure missing indicator [yes or no], fasting blood glucose missing indicator [yes or no], serum total cholesterol missing indicator [yes or no], glomerular filtration rate missing indicator [yes or no], smoking status missing indicator [yes or no], alcoholic drinks missing indicator [yes or no], and aerobic physical activity missing indicator [yes or no]).

UK Biobank (replication cohort B)

Covariates

The demographic characteristics of the participants were obtained from the health UK biobank database as followings: age (20–39, 40–59, and ≥ 60 years), sex, household income (<£18,000, £18,000–£30,999, £31,000–£51,999, £52,000–£100,000, >£100,000), and unknown), region of residence (urban and rural), townsend deprivation index (T1[last deprived], T2, T3 [most deprived], and unknown), race (white, mixed, Asian, black, others, and unknown), Charlson comorbidity index (0, 1, and ≥ 2), BMI (normal [$<25.0 \text{ kg/m}^2$], overweight [$25.0\text{--}30.0 \text{ kg/m}^2$], obese [$\geq 30.0 \text{ kg/m}^2$], and unknown), education levels (≤ 10 , 11-12, >12 , and unknown), blood pressure (systolic blood pressure $< 140 \text{ mmHg}$ and diastolic blood pressure $< 90 \text{ mmHg}$, systolic blood pressure $\geq 140 \text{ mmHg}$ or diastolic blood pressure $\geq 90 \text{ mmHg}$, and unknown), fasting blood glucose (<100 , $\geq 100 \text{ mg/dL}$, and unknown), smoking status (non- and current smoker, and unknown), alcohol consumption (every day, sometimes, rarely days per week, and unknown), aerobic physical activity (low, moderate, high, and unknown), previous history of cardiovascular disease, chronic kidney disease, and chronic obstructive pulmonary disease, history of medication use for diabetes mellitus, dyslipidemia, and hypertension, and missing indicators (household income missing indicator [yes or no], townsend deprivation index missing indicator [yes or no], race missing indicator [yes or no], education levels [yes or no], obesity missing indicator [yes or no], blood pressure missing indicator [yes or no], fasting blood glucose missing indicator [yes or no], serum total cholesterol missing indicator [yes or no], glomerular filtration rate missing indicator [yes or no], smoking status missing indicator [yes or no], alcoholic drinks missing indicator [yes or no], and aerobic physical activity missing indicator [yes or no]).

Comment 4.

For the matching process the authors used age, sex, and region type. It is unclear why matching was not exact on these variable and if not why the introduction of other variable in the matching process would induce immortal time bias. “To reduce the immortal bias, variables requiring index date, such as previous history of diseases, are not included as matching covariates” If immortal time bias is an issue, this should also apply to adjustment of the variables. In so far it is not understandable why certain variables were selected in the matching process.

Response:

We appreciate the insightful and constructive feedback. First, we sincerely apologize for the insufficient understanding in the process of establishing matching parameters. The purpose of matching is to make the characteristics of two groups as similar as possible, excluding exposure status, therefore it is advisable to match with all relevant confounding factors included. However, the reason for excluding variables, such as the previous history of diseases, for matching was that we defined these variables based on the index date. The current study analyzed that the use of variables necessitating the index date could inadvertently exert a temporally related positive impact on the exposed group, thereby counterfeiting the role of the exposure variable and potentially disconcerting the matching process.¹ This concern prompted our decision to exclude variables such as previous history of diseases. However, this is not a direct variable that influences exposure status and serves as a metric that can better represent medical characteristics.

Therefore, we reconstructed the matching covariates to align the purpose of matching as follows (South Korea): age (20–39, 40–59, and ≥ 60 years), sex, household income (low income, middle income, and high income), region of residence (urban and rural), Charlson comorbidity index (CCI; 0, 1, and ≥ 2), obesity (underweight [<18.5 kg/m²], normal [18.5–23.0

kg/m²]; overweight [23.0–25.0 kg/m²], obese [\geq 25.0 kg/m²], and unknown), blood pressure (systolic blood pressure <140 mmHg and diastolic blood pressure <90 mmHg, systolic blood pressure \geq 140 mmHg or diastolic blood pressure \geq 90 mmHg, and unknown), fasting blood glucose (<100, \geq 100 mg/dL, and unknown), serum total cholesterol (<200, 200–240, \geq 240 mg/dL, and unknown), glomerular filtration rate (<60, 60–90, \geq 90 mL/min/1.73 m², and unknown), smoking status (non-, ex-, current smoker, and unknown), alcoholic drinks (<1, 1–2, 3–4, \geq 5 days per week, and unknown), aerobic physical activity (sufficient, insufficient, and unknown), and history of medication use for diabetes mellitus, dyslipidemia, and hypertension. For the replication cohorts of Japan and the UK, we also used all covariates as matching variables except for the variable corresponding to previous history of diseases (cardiovascular disease, chronic kidney disease, and chronic obstructive pulmonary disease). Moreover, even though the variable concerning the history of diseases was not included as the matching variables, the SMD values for these variables were less than 0.1 (Table 1, Table S4, and Table S5). This likely indicates that the inclusion of the CCI as a matching variable has adjusted the aspect you were concerned about. We also obtained results for models that included all of the variables we considered as confounders in calculating the hazard ratio of the outcome (model 2 [South Korea]; adjusted for age, sex, household income, region of residence, CCI, BMI, blood pressure, fasting blood glucose, serum total cholesterol, glomerular filtration rate, smoking status, alcoholic drinks, aerobic physical activity, previous history of cardiovascular disease, chronic kidney disease, and chronic obstructive pulmonary disease, history of medication use for diabetes mellitus, dyslipidemia, and hypertension, and missing indicators [BMI, blood pressure, fasting blood glucose, serum total cholesterol, glomerular filtration rate, smoking status, alcoholic drinks, and aerobic physical activity]).

We believe all of this will make the association between exposure and outcome more

reliable. We have revised all of our manuscripts and tables in this regard and appreciate your feedback and insights to help us improve the quality of our study.

<Reference>

1. Lévesque LE, Hanley JA, Kezouh A, Suissa S. Problem of immortal time bias in cohort studies: example using statins for preventing progression of diabetes. *BMJ*. 2010 Mar 12;340:b5087. doi: 10.1136/bmj.b5087. PMID: 20228141.

Changes in text:

Abstract

After a 1:5 propensity score matching, we observed an elevated risk of developing allergic diseases associated with COVID-19, beyond the 30 days after COVID-19 diagnosis (HR, 1.20; 95% CI, 1.13-1.27). This association was specifically shown in asthma (HR, 2.25; 95% CI, 1.80-2.83) and allergic rhinitis (HR, 1.23; 95% CI, 1.15-1.32).

Notably, COVID-19 vaccination, at least two doses, had a protective effect against the incidence of allergic diseases following COVID-19 (HR, 0.81; 95% CI, 0.68-0.96).

Methods, *Propensity score matching*

~~To reduce the immortal bias, variables requiring index date, such as previous history of diseases, are not included as matching covariates.~~

[revised manuscript text omitted]

Cohort	Exposure	n (%)	Incidence rate*	HR (95% CI)	
				Model 1 ^a	Model 1 ^a
Allergic diseases					
Main cohort	None	688,340 (82.32)	24.0	1.00 (reference)	1.00 (reference)
Main cohort	Patients with COVID-19	147,824 (17.68)	28.7	1.20 (1.13 to 1.27)	1.20 (1.13 to 1.27)
Replication cohort A	None	1,998,524 (78.65)	27.4	1.00 (reference)	1.00 (reference)
Replication cohort A	Patients with COVID-19	542,497 (21.35)	75.1	2.65 (2.61 to 2.69)	2.56 (2.52 to 2.59)
Replication cohort B	None	248,949 (76.40)	7.2	1.00 (reference)	1.00 (reference)
Replication cohort B	Patients with COVID-19	76,894 (23.60)	8.3	1.14 (1.06 to 1.22)	1.12 (1.04 to 1.20)
Asthma					
Main cohort	None	688,340 (82.32)	1.0	1.00 (reference)	1.00 (reference)
Main cohort	Patients with COVID-19	147,824 (17.68)	2.2	2.27 (1.81 to 2.85)	2.25 (1.80 to 2.83)
Replication cohort A	None	1,998,524 (78.65)	2.7	1.00 (reference)	1.00 (reference)
Replication cohort A	Patients with COVID-19	542,497 (21.35)	7.6	2.77 (2.65 to 2.90)	2.63 (2.51 to 2.75)
Replication cohort B	None	248,949 (76.40)	3.2	1.00 (reference)	1.00 (reference)
Replication cohort B	Patients with COVID-19	76,894 (23.60)	3.8	1.16 (1.05 to 1.29)	1.14 (1.03 to 1.27)
Allergic rhinitis					
Main cohort	None	688,340 (82.32)	17.3	1.00 (reference)	1.00 (reference)
Main cohort	Patients with COVID-19	147,824 (17.68)	21.3	1.23 (1.15 to 1.32)	1.23 (1.15 to 1.32)
Replication cohort A	None	1,998,524 (78.65)	20.4	1.00 (reference)	1.00 (reference)
Replication cohort A	Patients with COVID-19	542,497 (21.35)	62.6	2.98 (2.93 to 3.03)	2.88 (2.83 to 2.93)
Replication cohort B	None	248,949 (76.40)	1.1	1.00 (reference)	1.00 (reference)
Replication cohort B	Patients with COVID-19	76,894 (23.60)	1.4	1.21 (1.02 to 1.44)	1.20 (1.01 to 1.43)
Atopic dermatitis					
Main cohort	None	688,340 (82.32)	2.6	1.00 (reference)	1.00 (reference)
Main cohort	Patients with COVID-19	147,824 (17.68)	3.0	1.15 (0.96 to 1.37)	1.15 (0.96 to 1.37)
Replication cohort A	None	1,998,524 (78.65)	5.2	1.00 (reference)	1.00 (reference)
Replication cohort A	Patients with COVID-19	542,497 (21.35)	6.8	1.25 (1.20 to 1.30)	1.21 (1.16 to 1.26)
Replication cohort B	None	248,949 (76.40)	0.04	1.00 (reference)	1.00 (reference)

Replication cohort B	Patients with COVID-19	76,894 (23.60)	0.05	1.18 (0.46 to 2.99)	1.19 (0.47 to 3.01)
Food allergy					
Main cohort	None	688,340 (82.32)	3.6	1.00 (reference)	1.00 (reference)
Main cohort	Patients with COVID-19	147,824 (17.68)	3.0	0.85 (0.71 to 1.01)	0.85 (0.71 to 1.00)
Replication cohort A	None	1,998,524 (78.65)	1.3	1.00 (reference)	1.00 (reference)
Replication cohort A	Patients with COVID-19	542,497 (21.35)	2.7	2.10 (1.95 to 2.25)	1.84 (1.71 to 1.98)
Replication cohort B	None	248,949 (76.40)	3.1	1.00 (reference)	1.00 (reference)
Replication cohort B	Patients with COVID-19	76,894 (23.60)	3.4	1.09 (0.97 to 1.21)	1.07 (0.96 to 1.20)

BMI, body mass index; CI, confidence interval; HR, hazard ratio.

* Incidence rate is expressed as per 1,000 person-years.

The data in bold indicate significant differences ($P < 0.05$).

^a **Model 1:** Adjusted for age (20–39, 40–59, and ≥ 60 years) and sex.

^b **Model 2 (main cohort):** Adjusted for age (20–39, 40–59, and ≥ 60 years); sex; household income (low income, middle income, and high income); region of residence (urban and rural); Charlson comorbidity index (0, 1, and ≥ 2); BMI (underweight [$< 18.5 \text{ kg/m}^2$], normal [$18.5\text{--}23.0 \text{ kg/m}^2$], overweight [$23.0\text{--}25.0 \text{ kg/m}^2$], obese [$\geq 25.0 \text{ kg/m}^2$], and unknown); blood pressure (systolic blood pressure $< 140 \text{ mmHg}$ and diastolic blood pressure $< 90 \text{ mmHg}$, systolic blood pressure $\geq 140 \text{ mmHg}$ or diastolic blood pressure $\geq 90 \text{ mmHg}$, and unknown); fasting blood glucose (< 100 , $\geq 100 \text{ mg/dL}$, and unknown); serum total cholesterol (< 200 , $200\text{--}240$, $\geq 240 \text{ mg/dL}$, and unknown); glomerular filtration rate (< 60 , $60\text{--}90$, $\geq 90 \text{ mL/min/1.73 m}^2$, and unknown); smoking status (non-, ex-, current smoker, and unknown); alcoholic drinks (< 1 , $1\text{--}2$, $3\text{--}4$, ≥ 5 days per week, and unknown); aerobic physical activity (sufficient, insufficient, and unknown); previous history of cardiovascular disease, chronic kidney disease, and chronic obstructive pulmonary disease; history of medication use for diabetes mellitus, dyslipidemia, and hypertension; and missing indicators (BMI missing indicator [yes or no], blood pressure missing indicator [yes or no], fasting blood glucose missing indicator [yes or no], serum total cholesterol missing indicator [yes or no], glomerular filtration rate missing indicator [yes or no], smoking status missing indicator [yes or no], alcoholic drinks missing indicator [yes or no], and aerobic physical activity missing indicator [yes or no]).

^b Model 2 (replication cohort A): Adjusted for age (20–39, 40–59, and ≥ 60 years); sex; Charlson comorbidity index (0, 1, and ≥ 2); BMI (underweight [<18.5 kg/m²], normal [18.5–23.0 kg/m²], overweight [23.0–25.0 kg/m²], obese [≥ 25.0 kg/m²], and unknown); blood pressure (systolic blood pressure <140 mmHg and diastolic blood pressure <90 mmHg, systolic blood pressure ≥ 140 mmHg or diastolic blood pressure ≥ 90 mmHg, and unknown); fasting blood glucose (<100 , ≥ 100 mg/dL, and unknown); serum total cholesterol (<200 , 200–240, ≥ 240 mg/dL, and unknown); glomerular filtration rate (<60 , 60–90, ≥ 90 mL/min/1.73 m², and unknown); smoking status (non- and current smoker, and unknown); alcoholic drinks (drinks; <1 , 1–2, 3–4, ≥ 5 days per week, and unknown); aerobic physical activity (sufficient, insufficient, and unknown); previous history of cardiovascular disease, chronic kidney disease, and chronic obstructive pulmonary disease; history of medication use for diabetes mellitus, dyslipidemia, and hypertension; and missing indicators (BMI missing indicator [yes or no], blood pressure missing indicator [yes or no], fasting blood glucose missing indicator [yes or no], serum total cholesterol missing indicator [yes or no], glomerular filtration rate missing indicator [yes or no], smoking status missing indicator [yes or no], alcoholic drinks missing indicator [yes or no], and aerobic physical activity missing indicator [yes or no]).

^b Model 2 (replication cohort B): Adjusted for age (20–39, 40–59, and ≥ 60 years); sex; household income ($<£18,000$, £18,000–£30,999, £31,000–£51,999, £52,000–£100,000, $>£100,000$), and unknown); region of residence (urban and rural); townsend deprivation index (T1 [least deprived], T2, T3 [most deprived], and unknown); ethnicity (white, mixed, Asian, black, others, and unknown); Charlson comorbidity index (0, 1, and ≥ 2); BMI (normal [<25.0 kg/m²], overweight [25.0–30.0 kg/m²], obese [≥ 30.0 kg/m²], and unknown); education levels (≤ 10 , 11–12, >12 , and unknown); blood pressure (systolic blood pressure < 140 mmHg and diastolic blood pressure < 90 mmHg, systolic blood pressure ≥ 140 mmHg or diastolic blood pressure ≥ 90 mmHg, and unknown); fasting blood glucose (<100 , ≥ 100 mg/dL, and unknown); smoking status (non- and current smoker, and unknown); alcohol consumption (every day, sometimes, rarely days per week, and unknown); aerobic physical activity (low, moderate, high, and unknown); previous history of cardiovascular disease, chronic kidney disease, and chronic obstructive pulmonary disease; history of medication use for diabetes mellitus, dyslipidemia, and hypertension; and missing indicators (household income missing indicator [yes or no], townsend deprivation index missing indicator [yes or no], ethnicity missing indicator [yes or no], education levels [yes or no], obesity missing indicator [yes or no], blood pressure missing indicator [yes or no], fasting blood glucose missing indicator [yes or no], serum total cholesterol missing indicator [yes or no], glomerular filtration rate missing indicator [yes or no], smoking status missing indicator [yes or no], alcoholic drinks missing indicator [yes or no], and aerobic physical activity missing indicator [yes or no]).

Table 3. Time attenuation effect on the development of allergic diseases after SARS-CoV-2 infection (model 2 adjusted HR with 95% CI).

	HR (95% CI)	
	Main cohort (South Korea)	Replication cohort A (Japan)
Allergic diseases		
<3 months	1.42 (1.29 to 1.56)	3.30 (3.24 to 3.36)
3–6 months	1.14 (1.01 to 1.29)	1.77 (1.70 to 1.84)
≥6 months	1.00 (0.91 to 1.11)	1.61 (1.56 to 1.67)

BMI, body mass index; CI, confidence interval; HR, hazard ratio; SARS-CoV-2, severe acute respiratory syndrome coronavirus.

The data in bold indicate significant differences ($P < 0.05$).

Model 2 (main cohort): Adjusted for age (20–39, 40–59, and ≥60 years); sex; household income (low income, middle income, and high income); region of residence (urban and rural); Charlson comorbidity index (0, 1, and ≥2); BMI (underweight [$<18.5 \text{ kg/m}^2$], normal [$18.5\text{--}23.0 \text{ kg/m}^2$], overweight [$23.0\text{--}25.0 \text{ kg/m}^2$], obese [$\geq 25.0 \text{ kg/m}^2$], and unknown); blood pressure (systolic blood pressure $<140 \text{ mmHg}$ and diastolic blood pressure $<90 \text{ mmHg}$, systolic blood pressure $\geq 140 \text{ mmHg}$ or diastolic blood pressure $\geq 90 \text{ mmHg}$, and unknown); fasting blood glucose (<100 , $\geq 100 \text{ mg/dL}$, and unknown); serum total cholesterol (<200 , $200\text{--}240$, $\geq 240 \text{ mg/dL}$, and unknown); glomerular filtration rate (<60 , $60\text{--}90$, $\geq 90 \text{ mL/min/1.73 m}^2$, and unknown); smoking status (non-, ex-, current smoker, and unknown); alcoholic drinks (<1 , $1\text{--}2$, $3\text{--}4$, ≥ 5 days per week, and unknown); aerobic physical activity (sufficient, insufficient, and unknown); previous history of cardiovascular disease, chronic kidney disease, and chronic obstructive pulmonary disease; history of medication use for diabetes mellitus, dyslipidemia, and hypertension; and missing indicators (BMI missing indicator [yes or no], blood pressure missing indicator [yes or no], fasting blood glucose missing indicator [yes or no], serum total cholesterol missing indicator [yes or no], glomerular filtration rate missing indicator [yes or no], smoking status missing indicator [yes or no], alcoholic drinks missing indicator [yes or no], and aerobic physical activity missing indicator [yes or no]).

Model 2 (replication cohort A): Adjusted for age (20–39, 40–59, and ≥60 years); sex; Charlson comorbidity index (0, 1, and ≥ 2); BMI (underweight [$<18.5 \text{ kg/m}^2$], normal [$18.5\text{--}23.0 \text{ kg/m}^2$], overweight [$23.0\text{--}25.0 \text{ kg/m}^2$], obese [$\geq 25.0 \text{ kg/m}^2$], and unknown); blood pressure

(systolic blood pressure <140 mmHg and diastolic blood pressure <90 mmHg, systolic blood pressure \geq 140 mmHg or diastolic blood pressure \geq 90 mmHg, and unknown); fasting blood glucose (<100, \geq 100 mg/dL, and unknown); serum total cholesterol (<200, 200–240, \geq 240 mg/dL, and unknown); glomerular filtration rate (<60, 60–90, \geq 90 mL/min/1.73 m², and unknown); smoking status (non- and current smoker, and unknown); alcoholic drinks (drinks; <1, 1–2, 3–4, \geq 5 days per week, and unknown); aerobic physical activity (sufficient, insufficient, and unknown); previous history of cardiovascular disease, chronic kidney disease, and chronic obstructive pulmonary disease; history of medication use for diabetes mellitus, dyslipidemia, and hypertension; and missing indicators (BMI missing indicator [yes or no], blood pressure missing indicator [yes or no], fasting blood glucose missing indicator [yes or no], serum total cholesterol missing indicator [yes or no], glomerular filtration rate missing indicator [yes or no], smoking status missing indicator [yes or no], alcoholic drinks missing indicator [yes or no], and aerobic physical activity missing indicator [yes or no]).

Table 4. Propensity-score-matched subgroup analysis of HR (95% CI) of allergic diseases following COVID-19 diagnosis stratified by COVID-19 severity, SARS-CoV-2 strain type, and number of vaccinations in **main cohort** (South Korea).

Factors	Group	Exposure	Events/total number (%)	HR (95% CI)	
				Model 1 ^a	Model 2 ^b
COVID-19 severity	Total	Non-infected control	5,633/688,340 (0.82)	1.00 (reference)	1.00 (reference)
		Mild COVID-19	1,125/127,641 (0.88)	1.13 (1.06 to 1.20)	1.14 (1.07 to 1.21)
		Moderate to severe COVID-19	293/20,183 (1.45)	1.55 (1.38 to 1.74)	1.48 (1.31 to 1.66)
Strain type (original)	Total	Non-infected control at the same index date [‡]	4,174/220,904 (1.89)	1.00 (reference)	1.00 (reference)
		Original SARS-CoV-2 infection	1,050/46,900 (2.24)	1.20 (1.12 to 1.29)	1.20 (1.12 to 1.29)
Strain type (delta)	Total	Non-infected control at the same index date [‡]	1,459/467,436 (0.31)	1.00 (reference)	1.00 (reference)
		Delta SARS-CoV-2 infection	368/100,924 (0.36)	1.18 (1.05 to 1.32)	1.18 (1.05 to 1.32)
Number of SARS-CoV-2 vaccinations	Patients with COVID-19	Non-infected control	5,633/688,340 (0.82)	1.00 (reference)	1.00 (reference)
		Without vaccination	1,133/68,456 (1.66)	1.24 (1.16 to 1.32)	1.24 (1.16 to 1.32)
		Vaccination 1 time	148/14,125 (1.05)	1.43 (1.22 to 1.69)	1.44 (1.22 to 1.69)
		Vaccination ≥ 2 times	137/65,243 (0.21)	0.81 (0.68 to 0.96)	0.81 (0.68 to 0.96)

BMI, body mass index; CI, confidence interval; HR, hazard ratio; SARS-CoV-2, severe acute respiratory syndrome coronavirus.

The data in bold indicate significant differences ($P < 0.05$).

^a **Model 1:** Adjusted for age (20–39, 40–59, and ≥ 60 years) and sex.

^b **Model 2 (main cohort):** Adjusted for age (20–39, 40–59, and ≥ 60 years); sex; household income (low income, middle income, and high income); region of residence (urban and rural); Charlson comorbidity index (0, 1, and ≥ 2); BMI (underweight [$< 18.5 \text{ kg/m}^2$], normal [$18.5\text{--}23.0 \text{ kg/m}^2$], overweight [$23.0\text{--}25.0 \text{ kg/m}^2$], obese [$\geq 25.0 \text{ kg/m}^2$], and unknown); blood pressure (systolic blood pressure $< 140 \text{ mmHg}$ and diastolic blood pressure $< 90 \text{ mmHg}$, systolic blood pressure $\geq 140 \text{ mmHg}$ or diastolic blood pressure $\geq 90 \text{ mmHg}$, and unknown); fasting blood glucose (< 100 , $\geq 100 \text{ mg/dL}$, and unknown); serum total cholesterol (< 200 , $200\text{--}240$, $\geq 240 \text{ mg/dL}$, and unknown); glomerular filtration rate (< 60 , $60\text{--}90$, $\geq 90 \text{ mL/min/1.73 m}^2$, and unknown); smoking status (non-, ex-, current smoker, and unknown); alcoholic drinks (< 1 , $1\text{--}2$, $3\text{--}4$, ≥ 5 days per week, and unknown); aerobic physical activity (sufficient, insufficient, and unknown); previous history of cardiovascular disease, chronic kidney disease, and chronic obstructive pulmonary disease; history of medication use for diabetes mellitus, dyslipidemia, and hypertension; and missing indicators (BMI missing indicator [yes or no], blood pressure missing indicator [yes or no], fasting blood glucose missing indicator [yes or no], serum total cholesterol missing indicator [yes or no], glomerular filtration rate missing indicator [yes or no], smoking status missing indicator [yes or no], alcoholic drinks missing indicator [yes or no], and aerobic physical activity missing indicator [yes or no]).

Figure S1. Density plot and box plot for before matching and after matching in patients with COVID-19 and non- COVID-19 in main cohort (South Korea)

Figure S2. Density plot and box plot for before matching and after matching in patients with COVID-19 and non- COVID-19 in replication cohort A (Japan).

Figure S3. Density plot and box plot for before matching and after matching in patients with COVID-19 and non- COVID-19 in replication cohort B (UK)

Table S4. Baseline characteristics for 1:5 propensity score-matched cohort in replication cohort A (Japan, n=2,541,021).

Covariates	1:5 matched cohort (n= 2,541,021)			SMD
	Total	COVID-19 (n=542,497)	Non-COVID-19 (n= 1,998,524)	
Age, years, mean (SD)	46.91 (11.41)	44.95 (11.96)	47.44 (11.20)	0.215
Age, years, n (%)				0.221
20–39	677,584 (26.67)	187,828 (34.62)	489,756 (24.51)	
40–59	1,527,989 (60.13)	289,616 (53.39)	1,238,373 (61.96)	
≥ 60	335,448 (13.20)	65,053 (11.99)	270,395 (13.53)	
Sex, n (%)				0.030
Male	1,712,733 (67.40)	359,753 (66.31)	1,352,980 (67.70)	
Female	828,288 (32.60)	182,744 (33.69)	645,544 (32.30)	
Charlson comorbidity index, n (%)				0.203
0	2,497,253 (98.28)	521,111 (96.06)	1,976,142 (98.88)	
1	13,724 (0.54)	6,583 (1.21)	7,141 (0.36)	
≥2	30,044 (1.18)	14,803 (2.73)	15,241 (0.76)	
History of cardiovascular disease, n (%)	237,077 (9.33)	72,945 (13.45)	164,132 (8.21)	0.169
History of chronic kidney disease, n (%)	111,533 (4.39)	28,843 (5.32)	82,690 (4.14)	0.056
History of chronic obstructive pulmonary disease, n (%)	5,964 (0.23)	2,407 (0.44)	3,557 (0.18)	0.048
History of medication use for diabetes	67,783 (2.67)	14,605 (2.69)	53,178 (2.66)	0.002
History of medication use for hyperlipidemia	155,190 (6.11)	31,890 (5.88)	123,300 (6.17)	0.012
History of medication use for hypertension	204,823 (8.06)	43,409 (8.00)	161,414 (8.08)	0.003
Body mass index, kg/m ² , n (%)				0.149
Underweight (<18.5)	189,039 (7.44)	42,662 (7.86)	146,377 (7.32)	
Normal (18.5–23.0)	1,103,232 (43.42)	238,313 (43.93)	864,919 (43.28)	

Overweight (23.0–25.0)	500,160 (19.68)	103,239 (19.03)	396,921 (19.86)	
Obese (≥ 25.0)	736,406 (28.98)	156,101 (28.77)	580,305 (29.04)	
Unknown	12,184 (0.48)	2,182 (0.40)	10,002 (0.50)	
Blood pressure, n (%)				0.056
SBP <140 mmHg and DBP <90 mmHg	2,130,467 (83.84)	463,355 (85.41)	1,667,112 (83.42)	
SBP ≥ 140 mmHg or DBP ≥ 90 mmHg	397,463 (15.64)	76,778 (14.15)	320,685 (16.05)	
Unknown	13,091 (0.52)	2,364 (0.44)	10,727 (0.54)	
Fasting blood glucose, mg/dL, n (%)				0.075
<100	1,710,497 (67.32)	372,569 (68.68)	1,337,928 (66.95)	
≥ 100	616,724 (24.27)	121,909 (22.47)	494,815 (24.76)	
Unknown	213,800 (8.41)	48,019 (8.85)	165,781 (8.30)	
Serum total cholesterol, mg/dL, n (%)				0.084
<200	1,111,656 (43.75)	254,222 (46.86)	857,434 (42.90)	
200–240	960,246 (37.79)	195,032 (35.95)	765,214 (38.29)	
≥ 240	440,673 (17.34)	86,712 (15.98)	353,961 (17.71)	
Unknown	28,446 (1.12)	6,531 (1.20)	21,915 (1.10)	
Glomerular filtration rate, mL/min/1.73 m ² , n (%)				0.079
<60	63,408 (2.48)	15,244 (2.81)	48,164 (2.41)	
60–90	1,080,930 (42.54)	231,158 (42.61)	849,772 (42.52)	
≥ 90	1,362,670 (53.64)	288,663 (53.21)	1,074,007 (53.74)	
Unknown	34,013 (1.34)	7,432 (1.37)	26,581 (1.33)	
Smoking status, n (%)				0.023
Current smoker	675,763 (26.59)	140,478 (25.89)	535,285 (26.78)	
Non-smoker	1,783,816 (70.20)	383,597 (70.71)	1,400,219 (70.06)	
Unknown	81,442 (3.21)	18,422 (3.40)	63,020 (3.15)	
Alcoholic drinks, days per week, n (%)				0.044
<1	587,680 (23.13)	118,476 (21.84)	469,204 (23.48)	

1–2	826,498 (32.53)	182,406 (33.62)	644,092 (32.23)	
3–4	982,194 (38.65)	207,717 (38.29)	774,477 (38.75)	
≥5	144,649 (5.69)	33,898 (6.25)	110,751 (5.54)	
Unknown	587,680 (23.13)	118,476 (21.84)	469,204 (23.48)	
Aerobic physical activity, n (%)				0.050
Sufficient	546,224 (21.50)	110,731 (20.41)	435,493 (21.79)	
Insufficient	1,821,694 (71.69)	391,479 (72.16)	1,430,215 (71.56)	
Unknown	173,103 (6.81)	40,287 (7.43)	132,816 (6.65)	

DBP, diastolic blood pressure; SBP, systolic blood pressure; SD, standard deviation; SMD, standardized mean difference.

Table S5. Baseline characteristics for 1:5 propensity score-matched cohort in replication cohort B (UK, n=325,843).

Covariates	1:5 matched cohort (n=325,843)			SMD
	Total	COVID-19 (n=76,894)	Non-COVID-19 (n=248,949)	
Age, years, mean (SD)	71.08 (7.87)	71.33 (7.58)	71.00 (7.95)	0.048
Age, years, n (%)				0.048
40–59	30,577 (9.38)	6,726 (8.75)	23,851 (9.58)	
≥ 60	295,266 (90.62)	70,168 (91.25)	225,098 (90.42)	
Sex, n (%)				0.043
Male	186,007 (57.08)	31,764 (41.31)	108,072 (43.41)	
Female	139,836 (42.92)	45,130 (58.69)	140,877 (56.59)	
Region of residence				0.014
Urban	277,445 (85.15)	65,179 (84.76)	212,266 (85.26)	
Rural	48,398 (14.85)	11,715 (15.24)	36,683 (14.74)	
Household income, n (%)				0.186
Level1 (<£18,000)	53,572 (16.44)	10,311 (13.41)	43,261 (17.38)	
Level2 (£18,000-£30,999)	70,524 (21.64)	16,209 (21.08)	54,315 (21.82)	
Level3 (£31,000-£51,999)	76,821 (23.58)	19,028 (24.75)	57,793 (23.21)	
Level4 (£52,000-£100,000)	62,610 (19.21)	16,916 (22)	45,694 (18.35)	
Level5 (>£100,000)	17,314 (5.31)	5,177 (6.73)	12,137 (4.88)	
Unknown	45,002 (13.81)	9,253 (12.03)	35,749 (14.36)	
Townsend deprivation index, mean (SD)	-1.43 (3.02)	-1.73 (2.83)	-1.35 (3.06)	0.126
Townsend deprivation index tertile				0.143
T1 (last deprived)	118,547 (36.38)	28,477 (37.03)	90,070 (36.18)	

T2	112,207 (34.44)	26,375 (34.30)	85,832 (34.48)	
T3 (most deprived)	94,822 (29.1)	21,973 (28.58)	72,849 (29.26)	
Unknown	267 (0.08)	69 (0.09)	198 (0.08)	
Charlson comorbidity index, n (%)				0.064
0	311,545 (95.61)	72,715 (94.57)	238,830 (95.94)	
1	6,762 (2.08)	2,008 (2.61)	4,754 (1.91)	
≥2	7,536 (2.31)	2,171 (2.82)	5,365 (2.16)	
Race, n (%)				<0.001
White	313,685 (96.27)	74,072 (96.33)	239,613 (96.25)	
Mixed	2,495 (0.77)	599 (0.78)	1,896 (0.76)	
Asian	3,874 (1.19)	915 (1.19)	2,959 (1.19)	
Black	2,710 (0.83)	589 (0.77)	2,121 (0.85)	
Others	1,957 (0.60)	449 (0.58)	1,508 (0.61)	
Unknown	1,122 (0.34)	270 (0.35)	852 (0.34)	
Education levels, years, n (%)				0.142
≤10	176 (0.05)	44 (0.06)	132 (0.05)	
11-12	9,090 (2.79)	2,261 (2.94)	6,829 (2.74)	
>12	314,338 (96.47)	74,063 (96.32)	240,275 (96.52)	
Unknown	2,239 (0.69)	526 (0.68)	1,713 (0.69)	
History of cardiovascular disease, n (%)	32,179 (9.88)	8,264 (10.75)	23,915 (9.61)	0.038
History of chronic kidney disease, n (%)	17,274 (5.30)	4,849 (6.31)	12,425 (4.99)	0.057
History of chronic obstructive pulmonary disease, n (%)	6,330 (1.94)	2,051 (2.67)	4,279 (1.72)	0.065
History of medication use for diabetes	586 (0.18)	148 (0.19)	438 (0.18)	0.004
History of medication use for hyperlipidemia	10,609 (3.26)	2,551 (3.32)	8,058 (3.24)	0.005
History of medication use for hypertension	30,180 (9.26)	7,164 (9.32)	23,016 (9.25)	0.003
Body mass index, kg/m ² , n (%)				0.022
Normal (<25.0)	121,657 (37.34)	29,580 (38.47)	92,077 (36.99)	
Overweight (25.0–30.0)	140,093 (42.99)	32,341 (42.06)	107,752 (43.28)	
Obese (≥30.0)	63,108 (19.37)	14,740 (19.17)	48,368 (19.43)	

Unknown	985 (0.30)	233 (0.30)	752 (0.30)	
Blood pressure, n (%)				<0.001
SBP <140 mmHg and DBP <90 mmHg	167 (0.05)	42 (0.05)	125 (0.05)	
SBP ≥140 mmHg or DBP ≥90 mmHg	308,228 (94.59)	72,857 (94.75)	235,371 (94.55)	
Unknown	17,448 (5.35)	3,995 (5.20)	13,453 (5.40)	
Fasting blood glucose, mg/dL, n (%)				0.039
<100	241,863 (74.23)	56,841 (73.92)	185,022 (74.32)	
≥100	40,329 (12.38)	9688 (12.60)	30,641 (12.31)	
Unknown	43,651 (13.4)	10,365 (13.48)	33,286 (13.37)	
Smoking status, n (%)				<0.001
Current smoker	187,176 (57.44)	44,105 (57.36)	143,071 (57.47)	
Non-smoker	137,877 (42.31)	32,598 (42.39)	105,279 (42.29)	
Unknown	790 (0.24)	191 (0.25)	599 (0.24)	
Alcoholic drinks, days per week, n (%)				0.029
Every day	50,535 (15.51)	11,813 (15.36)	38,722 (15.55)	
Sometimes	120,935 (37.11)	28,101 (36.55)	92,834 (37.29)	
Rarely	154,024 (47.27)	36,901 (47.99)	117,123 (47.05)	
Unknown	349 (0.11)	79 (0.10)	270 (0.11)	
Aerobic physical activity, n (%)				0.066
Low	94,783 (29.09)	22,269 (28.96)	72,514 (29.13)	
Moderate	113,035 (34.69)	27,018 (35.14)	86,017 (34.55)	
High	108,836 (34.69)	25,414 (33.05)	83,422 (33.51)	

[revised manuscript text omitted]

Parameter	Exposure	n (%)	Incidence rate*	HR (95% CI)	
				Model 1 ^a	Model 2 ^b
Total	None	688,340 (82.32)	24.0	1.00 (reference)	1.00 (reference)
	Patients with COVID-19	147,824 (17.68)	28.7	1.20 (1.13 to 1.27)	1.20 (1.13 to 1.27)
Male	None	306,165 (82.10)	28.1	1.00 (reference)	1.00 (reference)
	Patients with COVID-19	66,749 (17.90)	33.1	1.18 (1.09 to 1.28)	1.18 (1.09 to 1.28)
Female	None	299,203 (82.51)	21.0	1.00 (reference)	1.00 (reference)
	Patients with COVID-19	63,421 (17.49)	25.7	1.22 (1.12 to 1.34)	1.22 (1.12 to 1.34)
Age 20–39 years	None	173,290 (82.39)	23.5	1.00 (reference)	1.00 (reference)
	Patients with COVID-19	37,030 (17.61)	26.1	1.11 (0.99 to 1.25)	1.11 (0.99 to 1.25)
Age 40–59 years	None	299,203 (82.51)	21.0	1.00 (reference)	1.00 (reference)
	Patients with COVID-19	63,421 (17.49)	25.7	1.22 (1.12 to 1.34)	1.22 (1.12 to 1.34)
Age ≥60 years	None	215,847 (82.00)	29.3	1.00 (reference)	1.00 (reference)

	Patients with COVID-19	47,373 (18.00)	36.1	1.23 (1.11 to 1.36)	1.23 (1.11 to 1.36)
Low income					
	None	298,798 (82.41)	24.6	1.00 (reference)	1.00 (reference)
	Patients with COVID-19	63,769 (17.59)	28.9	1.17 (1.07 to 1.28)	1.17 (1.07 to 1.28)
Middle income					
	None	262,108 (82.38)	23.7	1.00 (reference)	1.00 (reference)
	Patients with COVID-19	56,080 (17.62)	29.7	1.26 (1.14 to 1.38)	1.25 (1.14 to 1.38)
High income					
	None	127,434 (82.00)	23.2	1.00 (reference)	1.00 (reference)
	Patients with COVID-19	27,975 (18.00)	26.3	1.13 (0.98 to 1.30)	1.13 (0.98 to 1.30)
CCI, 0 score					
	None	597,115 (82.47)	22.8	1.00 (reference)	1.00 (reference)
	Patients with COVID-19	126,952 (17.53)	28.0	1.23 (1.15 to 1.31)	1.23 (1.15 to 1.31)
CCI, 1 scores					
	None	63,117 (82.15)	31.6	1.00 (reference)	1.00 (reference)
	Patients with COVID-19	13,712 (17.85)	31.6	1.00 (0.85 to 1.19)	1.01 (0.86 to 1.20)
CCI, ≥2 scores					
	None	28,108 (79.70)	30.1	1.00 (reference)	1.00 (reference)
	Patients with COVID-19	7,160 (20.30)	33.9	1.13 (0.88 to 1.45)	1.13 (0.88 to 1.45)
BMI, <18.5 kg/m²					
	None	17,997 (81.42)	23.5	1.00 (reference)	1.00 (reference)

BMI, 18.5-23.0 kg/m²	Patients with COVID-19	4,108 (18.58)	31.5	1.34 (0.95 to 1.89)	1.34 (0.95 to 1.89)
	None	220,919 (82.24)	24.3	1.00 (reference)	1.00 (reference)
BMI, 23.0-25.0 kg/m²	Patients with COVID-19	47,707 (17.76)	27.9	1.15 (1.03 to 1.27)	1.15 (1.03 to 1.27)
	None	159,861 (82.35)	24.0	1.00 (reference)	1.00 (reference)
BMI, ≥25.0 kg/m²	Patients with COVID-19	34,255 (17.65)	29.3	1.22 (1.08 to 1.37)	1.22 (1.08 to 1.38)
	None	289,452 (82.43)	23.8	1.00 (reference)	1.00 (reference)
Drinker	Patients with COVID-19	61,718 (17.57)	28.7	1.21 (1.11 to 1.33)	1.21 (1.11 to 1.33)
	None	293,797 (82.42)	22.2	1.00 (reference)	1.00 (reference)
Non-drinker	Patients with COVID-19	62,665 (17.58)	25.6	1.16 (1.05 to 1.27)	1.15 (1.05 to 1.27)
	None	394,032 (82.24)	25.3	1.00 (reference)	1.00 (reference)
Sufficient physical activity	Patients with COVID-19	85,120 (17.76)	31.0	1.22 (1.14 to 1.32)	1.22 (1.14 to 1.32)
	None	340,507 (82.39)	23.6	1.00 (reference)	1.00 (reference)
Insufficient physical activity	Patients with COVID-19	72,760 (17.61)	27.8	1.18 (1.08 to 1.28)	1.18 (1.08 to 1.28)
	None	347,204 (82.24)	24.4	1.00 (reference)	1.00 (reference)

	Patients with COVID-19	74,978 (17.76)	29.6	1.21 (1.12 to 1.31)	1.21 (1.12 to 1.31)
Smoker	None	251,296 (82.46)	21.3	1.00 (reference)	1.00 (reference)
	Patients with COVID-19	53,445 (17.54)	23.3	1.10 (0.99 to 1.22)	1.10 (0.99 to 1.22)
Non-smoker	None	436,584 (82.23)	25.5	1.00 (reference)	1.00 (reference)
	Patients with COVID-19	94,341 (17.77)	31.7	1.24 (1.16 to 1.33)	1.24 (1.16 to 1.33)
Rural residence	None	368,924 (82.35)	25.3	1.00 (reference)	1.00 (reference)
	Patients with COVID-19	79,097 (17.65)	29.9	1.18 (1.09 to 1.28)	1.18 (1.09 to 1.28)
Urban residence	None	319,416 (82.29)	22.6	1.00 (reference)	1.00 (reference)
	Patients with COVID-19	68,727 (17.71)	27.4	1.21 (1.11 to 1.32)	1.21 (1.11 to 1.32)
Strain type (original)	None	220,904 (82.49)	25.3	1.00 (reference)	1.00 (reference)
	Patients with COVID-19	46,900 (17.51)	30.4	1.20 (1.12 to 1.29)	1.20 (1.12 to 1.29)
Strain type (delta)	None	467,436 (82.24)	20.9	1.00 (reference)	1.00 (reference)
	Patients with COVID-19	100,924 (17.76)	24.7	1.18 (1.05 to 1.32)	1.18 (1.05 to 1.32)

BMI, body mass index; CCI, charlson comorbidity index; CI, confidence interval; HR, hazard ratio.

* Incidence rate is expressed as per 1,000 person-years.

The data in bold indicate significant differences ($P < 0.05$).

^a **Model 1:** Adjusted for age (20–39, 40–59, and ≥ 60 years) and sex.

^b **Model 2:** Adjusted for age (20–39, 40–59, and ≥ 60 years); sex; household income (low income, middle income, and high income); region of residence (urban and rural); Charlson comorbidity index (0, 1, and ≥ 2); obesity (underweight [< 18.5 kg/m²], normal [18.5–23.0 kg/m²], overweight [23.0–25.0 kg/m²], obese [≥ 25.0 kg/m²], and unknown); blood pressure (systolic blood pressure < 140 mmHg and diastolic blood pressure < 90 mmHg, systolic blood pressure ≥ 140 mmHg or diastolic blood pressure ≥ 90 mmHg, and unknown); fasting blood glucose (< 100 , ≥ 100 mg/dL, and unknown); serum total cholesterol (< 200 , 200–240, ≥ 240 mg/dL, and unknown); glomerular filtration rate (< 60 , 60–90, ≥ 90 mL/min/1.73 m², and unknown); smoking status (non-, ex-, current smoker, and unknown); alcoholic drinks (< 1 , 1–2, 3–4, ≥ 5 days per week, and unknown); aerobic physical activity (sufficient, insufficient, and unknown); previous history of cardiovascular disease, chronic kidney disease, and chronic obstructive pulmonary disease; history of medication use for diabetes mellitus, dyslipidemia, and hypertension; and missing indicators (obesity missing indicator [yes or no], blood pressure missing indicator [yes or no], fasting blood glucose missing indicator [yes or no], serum total cholesterol missing indicator [yes or no], glomerular filtration rate missing indicator [yes or no], smoking status missing indicator [yes or no], alcoholic drinks missing indicator [yes or no], and aerobic physical activity missing indicator [yes or no]).

Table S11. The HR with 95% CI for the long-term sequelae risk of incident asthma following COVID-19 diagnosis in the propensity score-matched main cohort (South Korea).

Parameter	Exposure	n (%)	Incidence rate*	HR (95% CI)	
				Model 1 ^a	Model 2 ^b
Total					
	None	688,340 (82.32)	1.0	1.00 (reference)	1.00 (reference)
	Patients with COVID-19	147,824 (17.68)	2.2	2.27 (1.81 to 2.85)	2.25 (1.80 to 2.83)
Male					
	None	382,175 (82.50)	0.8	1.00 (reference)	1.00 (reference)
	Patients with COVID-19	81,075 (17.50)	1.9	2.25 (1.62 to 3.14)	2.25 (1.61 to 3.14)
Female					
	None	306,165 (82.10)	1.1	1.0 (reference)	1.0 (reference)
	Patients with COVID-19	66,749 (17.90)	2.6	2.29 (1.67 to 3.12)	2.26 (1.66 to 3.09)
Age 20–39 years					
	None	173,290 (82.39)	0.8	1.00 (reference)	1.00 (reference)
	Patients with COVID-19	37,030 (17.61)	2.3	2.88 (1.83 to 4.54)	2.87 (1.82 to 4.53)
Age 40–59 years					
	None	299,203 (82.51)	0.7	1.00 (reference)	1.00 (reference)
	Patients with COVID-19	63,421 (17.49)	1.8	2.60 (1.79 to 3.79)	2.61 (1.79 to 3.80)

Age ≥60 years	None	215,847 (82.00)	1.6	1.00 (reference)	1.00 (reference)
	Patients with COVID-19	47,373 (18.00)	2.7	1.75 (1.20 to 2.53)	1.70 (1.17 to 2.46)
					
Low income	None	298,798 (82.41)	1.0	1.00 (reference)	1.00 (reference)
	Patients with COVID-19	63,769 (17.59)	2.0	2.02 (1.42 to 2.88)	2.03 (1.43 to 2.89)
Middle income	None	262,108 (82.38)	1.0	1.00 (reference)	1.00 (reference)
	Patients with COVID-19	56,080 (17.62)	2.2	2.32 (1.61 to 3.36)	2.27 (1.57 to 3.29)
High income	None	127,434 (82.00)	0.9	1.00 (reference)	1.00 (reference)
	Patients with COVID-19	27,975 (18.00)	2.5	2.77 (1.67 to 4.60)	2.67 (1.61 to 4.43)
					
CCI, 0 score	None	597,115 (82.47)	0.7	1.00 (reference)	1.00 (reference)
	Patients with COVID-19	126,952 (17.53)	2.1	2.82 (2.17 to 3.67)	2.82 (2.16 to 3.66)
CCI, 1 scores	None	63,117 (82.15)	2.2	1.00 (reference)	1.00 (reference)
	Patients with COVID-19	13,712 (17.85)	2.8	1.29 (0.73 to 2.28)	1.33 (0.75 to 2.36)

CCI, ≥ 2 scores					
	None	28,108 (79.70)	2.7	1.00 (reference)	1.00 (reference)
	Patients with COVID-19	7,160 (20.30)	3.0	1.13 (0.49 to 2.60)	1.03 (0.45 to 2.39)
					
BMI, < 18.5 kg/m²					
	None	17,997 (81.42)	1.0	1.00 (reference)	1.00 (reference)
	Patients with COVID-19	4,108 (18.58)	1.5	1.46 (0.29 to 7.23)	1.57 (0.31 to 8.04)
BMI, 18.5-23.0 kg/m²					
	None	220,919 (82.24)	0.9	1.00 (reference)	1.00 (reference)
	Patients with COVID-19	47,707 (17.76)	1.5	1.62 (1.02 to 2.58)	1.60 (1.01 to 2.54)
BMI, 23.0-25.0 kg/m²					
	None	159,861 (82.35)	1.1	1.00 (reference)	1.00 (reference)
	Patients with COVID-19	34,255 (17.65)	2.2	2.06 (1.30 to 3.26)	2.03 (1.28 to 3.22)
BMI, ≥ 25.0 kg/m²					
	None	289,452 (82.43)	0.9	1.00 (reference)	1.00 (reference)
	Patients with COVID-19	61,718 (17.57)	2.8	2.95 (2.13 to 4.09)	2.94 (2.12 to 4.08)
					
Drinker					
	None	293,797 (82.42)	0.8	1.00 (reference)	1.00 (reference)

	Patients with COVID-19	62,665 (17.58)	2.3	3.03 (2.12 to 4.32)	3.03 (2.13 to 4.33)
Non-drinker	None	394,032 (82.24)	1.1	1.00 (reference)	1.00 (reference)
	Patients with COVID-19	85,120 (17.76)	2.1	1.88 (1.39 to 2.53)	1.87 (1.38 to 2.52)
					
Sufficient physical activity	None	340,507 (82.39)	0.8	1.00 (reference)	1.00 (reference)
	Patients with COVID-19	72,760 (17.61)	1.8	2.19 (1.54 to 3.12)	2.21 (1.55 to 3.14)
Insufficient physical activity	None	347,204 (82.24)	1.1	1.00 (reference)	1.00 (reference)
	Patients with COVID-19	74,978 (17.76)	2.6	2.34 (1.74 to 3.16)	2.31 (1.71 to 3.11)
					
Smoker	None	251,296 (82.46)	0.9	1.00 (reference)	1.00 (reference)
	Patients with COVID-19	53,445 (17.54)	1.6	1.78 (1.15 to 2.74)	1.78 (1.15 to 2.75)
Non-smoker	None	436,584 (82.23)	1.0	1.00 (reference)	1.00 (reference)
	Patients with COVID-19	94,341 (17.77)	2.6	2.50 (1.92 to 3.27)	2.48 (1.90 to 3.24)

Rural residence					
	None	368,924 (82.35)	1.0	1.00 (reference)	1.00 (reference)
	Patients with COVID-19	79,097 (17.65)	2.1	2.04 (1.49 to 2.80)	2.03 (1.48 to 2.79)
Urban residence					
	None	319,416 (82.29)	0.9	1.00 (reference)	1.00 (reference)
	Patients with COVID-19	68,727 (17.71)	2.3	2.56 (1.85 to 3.56)	2.53 (1.82 to 3.51)
					
Strain type (original)					
	None	220,904 (82.49)	1.1	1.00 (reference)	1.00 (reference)
	Patients with COVID-19	46,900 (17.51)	2.3	2.08 (1.59 to 2.70)	2.07 (1.59 to 2.69)
Strain type (delta)					
	None	467,436 (82.24)	0.7	1.00 (reference)	1.00 (reference)
	Patients with COVID-19	100,924 (17.76)	2.1	3.01 (1.91 to 4.72)	2.97 (1.89 to 4.66)

BMI, body mass index; CCI, charlson comorbidity index; CI, confidence interval; HR, hazard ratio.

* Incidence rate is expressed as per 1,000 person-years.

The data in bold indicate significant differences ($P < 0.05$).

^a **Model 1:** Adjusted for age (20–39, 40–59, and ≥ 60 years) and sex.

^b **Model 2:** Adjusted for age (20–39, 40–59, and ≥ 60 years); sex; household income (low income, middle income, and high income); region of

residence (urban and rural); Charlson comorbidity index (0, 1, and ≥ 2); obesity (underweight [<18.5 kg/m²], normal [18.5–23.0 kg/m²], overweight [23.0–25.0 kg/m²], obese [≥ 25.0 kg/m²], and unknown); blood pressure (systolic blood pressure <140 mmHg and diastolic blood pressure <90 mmHg, systolic blood pressure ≥ 140 mmHg or diastolic blood pressure ≥ 90 mmHg, and unknown); fasting blood glucose (<100 , ≥ 100 mg/dL, and unknown); serum total cholesterol (<200 , 200–240, ≥ 240 mg/dL, and unknown); glomerular filtration rate (<60 , 60–90, ≥ 90 mL/min/1.73 m², and unknown); smoking status (non-, ex-, current smoker, and unknown); alcoholic drinks (<1 , 1–2, 3–4, ≥ 5 days per week, and unknown); aerobic physical activity (sufficient, insufficient, and unknown); previous history of cardiovascular disease, chronic kidney disease, and chronic obstructive pulmonary disease; history of medication use for diabetes mellitus, dyslipidemia, and hypertension; and missing indicators (obesity missing indicator [yes or no], blood pressure missing indicator [yes or no], fasting blood glucose missing indicator [yes or no], serum total cholesterol missing indicator [yes or no], glomerular filtration rate missing indicator [yes or no], smoking status missing indicator [yes or no], alcoholic drinks missing indicator [yes or no], and aerobic physical activity missing indicator [yes or no]).

Table S12. The HR with 95% CI for the long-term sequelae risk of incident allergic rhinitis following COVID-19 diagnosis in the propensity score-matched **main cohort (South Korea).**

Parameter	Exposure	n (%)	Incidence rate*	HR (95% CI)	
				Model 1 ^a	Model 2 ^b
Total					
	None	688,340 (82.32)	17.3	1.00 (reference)	1.00 (reference)
	Patients with COVID-19	147,824 (17.68)	21.3	1.23 (1.15 to 1.32)	1.23 (1.15 to 1.32)
Male					
	None	382,175 (82.50)	14.9	1.00 (reference)	1.00 (reference)
	Patients with COVID-19	81,075 (17.50)	18.9	1.27 (1.16 to 1.41)	1.27 (1.16 to 1.40)
Female					
	None	173,290 (82.39)	17.9	1.00 (reference)	1.00 (reference)
	Patients with COVID-19	37,030 (17.61)	20.2	1.13 (0.98 to 1.29)	1.13 (0.98 to 1.29)
Age 20–39 years					
	None	299,203 (82.51)	15.0	1.00 (reference)	1.00 (reference)
	Patients with COVID-19	63,421 (17.49)	19.5	1.30 (1.17 to 1.44)	1.30 (1.17 to 1.45)
Age 40–59 years					
	None	299,203 (82.51)	15.0	1.00 (reference)	1.00 (reference)
	Patients with COVID-19	63,421 (17.49)	19.5	1.30 (1.17 to 1.44)	1.30 (1.17 to 1.45)
Age ≥60 years					
	None	215,847 (82.00)	20.5	1.00 (reference)	1.00 (reference)

	Patients with COVID-19	47,373 (18.00)	25.2	1.23 (1.09 to 1.38)	1.23 (1.10 to 1.39)
Low income					
	None	298,798 (82.41)	17.6	1.00 (reference)	1.00 (reference)
	Patients with COVID-19	63,769 (17.59)	21.3	1.21 (1.09 to 1.34)	1.21 (1.09 to 1.34)
Middle income					
	None	262,108 (82.38)	17.2	1.00 (reference)	1.00 (reference)
	Patients with COVID-19	56,080 (17.62)	23.0	1.34 (1.20 to 1.49)	1.34 (1.20 to 1.49)
High income					
	None	127,434 (82.00)	16.5	1.00 (reference)	1.00 (reference)
	Patients with COVID-19	27,975 (18.00)	17.6	1.06 (0.90 to 1.26)	1.07 (0.90 to 1.26)
CCI, 0 score					
	None	597,115 (82.47)	16.7	1.00 (reference)	1.00 (reference)
	Patients with COVID-19	126,952 (17.53)	21.1	1.26 (1.17 to 1.36)	1.26 (1.17 to 1.36)
CCI, 1 scores					
	None	63,117 (82.15)	21.0	1.00 (reference)	1.00 (reference)
	Patients with COVID-19	13,712 (17.85)	21.8	1.04 (0.85 to 1.28)	1.06 (0.86 to 1.29)
CCI, ≥2 scores					
	None	28,108 (79.70)	19.9	1.00 (reference)	1.00 (reference)
	Patients with COVID-19	7,160 (20.30)	23.0	1.15 (0.85 to 1.56)	1.17 (0.86 to 1.59)
BMI, <18.5 kg/m²					
	None	17,997 (81.42)	17.4	1.00 (reference)	1.00 (reference)
	Patients with COVID-19	4,108 (18.58)	21.7	1.24 (0.83 to 1.88)	1.24 (0.82 to 1.87)

BMI, 18.5-23.0 kg/m²					
	None	220,919 (82.24)	17.8	1.00 (reference)	1.00 (reference)
	Patients with COVID-19	47,707 (17.76)	21.1	1.18 (1.05 to 1.33)	1.18 (1.04 to 1.33)
BMI, 23.0-25.0 kg/m²					
	None	159,861 (82.35)	17.1	1.00 (reference)	1.00 (reference)
	Patients with COVID-19	34,255 (17.65)	22.0	1.29 (1.12 to 1.48)	1.29 (1.12 to 1.48)
BMI, ≥25.0 kg/m²					
	None	289,452 (82.43)	16.9	1.00 (reference)	1.00 (reference)
	Patients with COVID-19	61,718 (17.57)	21.0	1.24 (1.12 to 1.38)	1.24 (1.12 to 1.38)
					
Drinker					
	None	293,797 (82.42)	16.1	1.00 (reference)	1.00 (reference)
	Patients with COVID-19	62,665 (17.58)	19.2	1.19 (1.07 to 1.33)	1.19 (1.07 to 1.33)
Non-drinker					
	None	394,032 (82.24)	18.1	1.00 (reference)	1.00 (reference)
	Patients with COVID-19	85,120 (17.76)	22.8	1.26 (1.15 to 1.37)	1.26 (1.15 to 1.37)
					
Sufficient physical activity					
	None	340,507 (82.39)	16.9	1.00 (reference)	1.00 (reference)
	Patients with COVID-19	72,760 (17.61)	20.8	1.23 (1.12 to 1.36)	1.23 (1.12 to 1.36)
Insufficient physical activity					
	None	347,204 (82.24)	17.6	1.00 (reference)	1.00 (reference)

	Patients with COVID-19	74,978 (17.76)	21.7	1.23 (1.12 to 1.35)	1.23 (1.12 to 1.35)
Smoker					
	None	251,296 (82.46)	15.5	1.00 (reference)	1.00 (reference)
	Patients with COVID-19	53,445 (17.54)	17.7	1.14 (1.01 to 1.29)	1.14 (1.01 to 1.29)
Non-smoker					
	None	436,584 (82.23)	18.3	1.00 (reference)	1.00 (reference)
	Patients with COVID-19	94,341 (17.77)	23.3	1.27 (1.17 to 1.38)	1.27 (1.17 to 1.38)
Rural residence					
	None	368,924 (82.35)	18.2	1.00 (reference)	1.00 (reference)
	Patients with COVID-19	79,097 (17.65)	21.8	1.20 (1.09 to 1.32)	1.20 (1.09 to 1.32)
Urban residence					
	None	319,416 (82.29)	16.3	1.00 (reference)	1.00 (reference)
	Patients with COVID-19	68,727 (17.71)	20.7	1.27 (1.15 to 1.40)	1.27 (1.15 to 1.40)
Strain type (original)					
	None	220,904 (82.49)	17.9	1.00 (reference)	1.00 (reference)
	Patients with COVID-19	46,900 (17.51)	22.2	1.24 (1.14 to 1.34)	1.24 (1.14 to 1.34)
Strain type (delta)					
	None	467,436 (82.24)	15.7	1.00 (reference)	1.00 (reference)
	Patients with COVID-19	100,924 (17.76)	19.2	1.22 (1.07 to 1.39)	1.22 (1.07 to 1.39)

BMI, body mass index; CCI, charlson comorbidity index; CI, confidence interval; HR, hazard ratio.

* Incidence rate is expressed as per 1,000 person-years.

The data in bold indicate significant differences ($P < 0.05$).

^a **Model 1:** Adjusted for age (20–39, 40–59, and ≥ 60 years) and sex.

^b **Model 2:** Adjusted for age (20–39, 40–59, and ≥ 60 years); sex; household income (low income, middle income, and high income); region of residence (urban and rural); Charlson comorbidity index (0, 1, and ≥ 2); obesity (underweight [< 18.5 kg/m²], normal [18.5–23.0 kg/m²], overweight [23.0–25.0 kg/m²], obese [≥ 25.0 kg/m²], and unknown); blood pressure (systolic blood pressure < 140 mmHg and diastolic blood pressure < 90 mmHg, systolic blood pressure ≥ 140 mmHg or diastolic blood pressure ≥ 90 mmHg, and unknown); fasting blood glucose (< 100 , ≥ 100 mg/dL, and unknown); serum total cholesterol (< 200 , 200–240, ≥ 240 mg/dL, and unknown); glomerular filtration rate (< 60 , 60–90, ≥ 90 mL/min/1.73 m², and unknown); smoking status (non-, ex-, current smoker, and unknown); alcoholic drinks (< 1 , 1–2, 3–4, ≥ 5 days per week, and unknown); aerobic physical activity (sufficient, insufficient, and unknown); previous history of cardiovascular disease, chronic kidney disease, and chronic obstructive pulmonary disease; history of medication use for diabetes mellitus, dyslipidemia, and hypertension; and missing indicators (obesity missing indicator [yes or no], blood pressure missing indicator [yes or no], fasting blood glucose missing indicator [yes or no], serum total cholesterol missing indicator [yes or no], glomerular filtration rate missing indicator [yes or no], smoking status missing indicator [yes or no], alcoholic drinks missing indicator [yes or no], and aerobic physical activity missing indicator [yes or no]).

Table S13. The HR with 95% CI for the long-term sequelae risk of incident atopic dermatitis following COVID-19 diagnosis in the propensity score-matched **main cohort (South Korea).**

Parameter	Exposure	n (%)	Incidence rate*	HR (95% CI)	
				Model 1 ^a	Model 2 ^b
Total					
	None	688,340 (82.32)	2.6	1.00 (reference)	1.00 (reference)
	Patients with COVID-19	147,824 (17.68)	3.0	1.15 (0.96 to 1.37)	1.15 (0.96 to 1.37)
Male					
	None	382,175 (82.50)	2.2	1.00 (reference)	1.00 (reference)
	Patients with COVID-19	81,075 (17.50)	2.2	1.02 (0.77 to 1.34)	1.01 (0.77 to 1.33)
Female					
	None	306,165 (82.10)	3.0	1.00 (reference)	1.00 (reference)
	Patients with COVID-19	66,749 (17.90)	3.8	1.27 (1.00 to 1.61)	1.27 (1.01 to 1.61)
Age 20–39 years					
	None	173,290 (82.39)	3.2	1.00 (reference)	1.00 (reference)
	Patients with COVID-19	37,030 (17.61)	3.0	0.95 (0.67 to 1.33)	0.95 (0.67 to 1.34)
Age 40–59 years					
	None	299,203 (82.51)	2.1	1.00 (reference)	1.00 (reference)
	Patients with COVID-19	63,421 (17.49)	2.4	1.13 (0.85 to 1.52)	1.13 (0.85 to 1.52)
Age ≥60 years					
	None	215,847 (82.00)	2.7	1.00 (reference)	1.00 (reference)

	Patients with COVID-19	47,373 (18.00)	3.8	1.39 (1.03 to 1.89)	1.38 (1.02 to 1.88)
Low income					
	None	298,798 (82.41)	2.7	1.00 (reference)	1.00 (reference)
	Patients with COVID-19	63,769 (17.59)	2.7	1.01 (0.76 to 1.34)	1.01 (0.76 to 1.34)
Middle income					
	None	262,108 (82.38)	2.3	1.00 (reference)	1.00 (reference)
	Patients with COVID-19	56,080 (17.62)	2.4	1.06 (0.77 to 1.46)	1.06 (0.77 to 1.46)
High income					
	None	127,434 (82.00)	2.9	1.00 (reference)	1.00 (reference)
	Patients with COVID-19	27,975 (18.00)	4.6	1.57 (1.12 to 2.21)	1.57 (1.11 to 2.21)
CCI, 0 score					
	None	597,115 (82.47)	2.5	1.00 (reference)	1.00 (reference)
	Patients with COVID-19	126,952 (17.53)	2.9	1.16 (0.95 to 1.41)	1.16 (0.95 to 1.41)
CCI, 1 scores					
	None	63,117 (82.15)	3.1	1.00 (reference)	1.00 (reference)
	Patients with COVID-19	13,712 (17.85)	3.6	1.17 (0.71 to 1.93)	1.17 (0.71 to 1.94)
CCI, ≥2 scores					
	None	28,108 (79.70)	3.5	1.00 (reference)	1.00 (reference)
	Patients with COVID-19	7,160 (20.30)	3.4	0.98 (0.45 to 2.11)	0.96 (0.44 to 2.08)
BMI, <18.5 kg/m²					
	None	17,997 (81.42)	2.3	1.00 (reference)	1.00 (reference)
	Patients with COVID-19	4,108 (18.58)	3.0	1.32 (0.44 to 4.02)	1.34 (0.44 to 4.09)

BMI, 18.5-23.0 kg/m²					
	None	220,919 (82.24)	2.8	1.00 (reference)	1.00 (reference)
	Patients with COVID-19	47,707 (17.76)	3.3	1.18 (0.87 to 1.60)	1.18 (0.87 to 1.59)
BMI, 23.0-25.0 kg/m²					
	None	159,861 (82.35)	2.4	1.00 (reference)	1.00 (reference)
	Patients with COVID-19	34,255 (17.65)	2.8	1.14 (0.78 to 1.68)	1.15 (0.78 to 1.69)
BMI, ≥25.0 kg/m²					
	None	289,452 (82.43)	2.5	1.00 (reference)	1.00 (reference)
	Patients with COVID-19	61,718 (17.57)	2.9	1.12 (0.85 to 1.49)	1.11 (0.84 to 1.47)
					
Drinker					
	None	293,797 (82.42)	2.4	1.00 (reference)	1.00 (reference)
	Patients with COVID-19	62,665 (17.58)	2.6	1.08 (0.81 to 1.45)	1.08 (0.80 to 1.44)
Non-drinker					
	None	394,032 (82.24)	2.7	1.00 (reference)	1.00 (reference)
	Patients with COVID-19	85,120 (17.76)	3.3	1.19 (0.95 to 1.50)	1.19 (0.95 to 1.50)
					
Sufficient physical activity					
	None	340,507 (82.39)	2.6	1.00 (reference)	1.00 (reference)
	Patients with COVID-19	72,760 (17.61)	2.6	0.98 (0.75 to 1.29)	0.98 (0.75 to 1.29)
Insufficient physical activity					
	None	347,204 (82.24)	2.5	1.00 (reference)	1.00 (reference)

	Patients with COVID-19	74,978 (17.76)	3.3	1.32 (1.03 to 1.68)	1.31 (1.03 to 1.67)
Smoker					
	None	251,296 (82.46)	2.1	1.00 (reference)	1.00 (reference)
	Patients with COVID-19	53,445 (17.54)	2.1	1.00 (0.71 to 1.42)	1.00 (0.71 to 1.42)
Non-smoker					
	None	436,584 (82.23)	2.8	1.00 (reference)	1.00 (reference)
	Patients with COVID-19	94,341 (17.77)	3.4	1.21 (0.98 to 1.49)	1.21 (0.98 to 1.49)
Rural residence					
	None	368,924 (82.35)	2.9	1.00 (reference)	1.00 (reference)
	Patients with COVID-19	79,097 (17.65)	3.4	1.18 (0.93 to 1.48)	1.17 (0.93 to 1.48)
Urban residence					
	None	319,416 (82.29)	2.2	1.00 (reference)	1.00 (reference)
	Patients with COVID-19	68,727 (17.71)	2.5	1.11 (0.84 to 1.48)	1.11 (0.84 to 1.47)
Strain type (original)					
	None	220,904 (82.49)	2.9	1.00 (reference)	1.00 (reference)
	Patients with COVID-19	46,900 (17.51)	3.5	1.22 (1.00 to 1.48)	1.21 (0.99 to 1.48)
Strain type (delta)					
	None	467,436 (82.24)	1.9	1.00 (reference)	1.00 (reference)
	Patients with COVID-19	100,924 (17.76)	1.7	0.92 (0.60 to 1.40)	0.92 (0.60 to 1.40)

BMI, body mass index; CCI, charlson comorbidity index; CI, confidence interval; HR, hazard ratio.

* Incidence rate is expressed as per 1,000 person-years.

The data in bold indicate significant differences ($P < 0.05$).

^a **Model 1:** Adjusted for age (20–39, 40–59, and ≥ 60 years) and sex.

^b **Model 2:** Adjusted for age (20–39, 40–59, and ≥ 60 years); sex; household income (low income, middle income, and high income); region of residence (urban and rural); Charlson comorbidity index (0, 1, and ≥ 2); obesity (underweight [< 18.5 kg/m²], normal [18.5–23.0 kg/m²], overweight [23.0–25.0 kg/m²], obese [≥ 25.0 kg/m²], and unknown); blood pressure (systolic blood pressure < 140 mmHg and diastolic blood pressure < 90 mmHg, systolic blood pressure ≥ 140 mmHg or diastolic blood pressure ≥ 90 mmHg, and unknown); fasting blood glucose (< 100 , ≥ 100 mg/dL, and unknown); serum total cholesterol (< 200 , 200–240, ≥ 240 mg/dL, and unknown); glomerular filtration rate (< 60 , 60–90, ≥ 90 mL/min/1.73 m², and unknown); smoking status (non-, ex-, current smoker, and unknown); alcoholic drinks (< 1 , 1–2, 3–4, ≥ 5 days per week, and unknown); aerobic physical activity (sufficient, insufficient, and unknown); previous history of cardiovascular disease, chronic kidney disease, and chronic obstructive pulmonary disease; history of medication use for diabetes mellitus, dyslipidemia, and hypertension; and missing indicators (obesity missing indicator [yes or no], blood pressure missing indicator [yes or no], fasting blood glucose missing indicator [yes or no], serum total cholesterol missing indicator [yes or no], glomerular filtration rate missing indicator [yes or no], smoking status missing indicator [yes or no], alcoholic drinks missing indicator [yes or no], and aerobic physical activity missing indicator [yes or no]).

Table S14. The HR with 95% CI for the long-term sequelae risk of incident food allergy following COVID-19 diagnosis in the propensity score-matched **main cohort (South Korea).**

Parameter	Exposure	n (%)	Incidence rate*	HR (95% CI)	
				Model 1 ^a	Model 2 ^b
Total					
	None	688,340 (82.32)	3.6	1.00 (reference)	1.00 (reference)
	Patients with COVID-19	147,824 (17.68)	3.0	0.85 (0.71 to 1.01)	0.85 (0.71 to 1.00)
Male					
	None	382,175 (82.50)	3.2	1.00 (reference)	1.00 (reference)
	Patients with COVID-19	81,075 (17.50)	2.5	0.80 (0.62 to 1.04)	0.80 (0.62 to 1.04)
Female					
	None	306,165 (82.10)	4.1	1.00 (reference)	1.00 (reference)
	Patients with COVID-19	66,749 (17.90)	3.7	0.88 (0.70 to 1.12)	0.88 (0.70 to 1.12)
Age 20–39 years					
	None	173,290 (82.39)	2.1	1.00 (reference)	1.00 (reference)
	Patients with COVID-19	37,030 (17.61)	1.6	0.77 (0.49 to 1.22)	0.77 (0.49 to 1.22)
Age 40–59 years					
	None	299,203 (82.51)	3.4	1.00 (reference)	1.00 (reference)
	Patients with COVID-19	63,421 (17.49)	2.6	0.75 (0.57 to 0.99)	0.75 (0.57 to 0.98)
Age ≥60 years					
	None	215,847 (82.00)	5.3	1.00 (reference)	1.00 (reference)

	Patients with COVID-19	47,373 (18.00)	5.2	0.97 (0.76 to 1.26)	0.98 (0.76 to 1.26)
Low income					
	None	298,798 (82.41)	3.8	1.00 (reference)	1.00 (reference)
	Patients with COVID-19	63,769 (17.59)	3.5	0.93 (0.73 to 1.20)	0.93 (0.73 to 1.19)
Middle income					
	None	262,108 (82.38)	3.5	1.00 (reference)	1.00 (reference)
	Patients with COVID-19	56,080 (17.62)	2.8	0.78 (0.58 to 1.05)	0.78 (0.58 to 1.04)
High income					
	None	127,434 (82.00)	3.3	1.00 (reference)	1.00 (reference)
	Patients with COVID-19	27,975 (18.00)	2.5	0.76 (0.49 to 1.16)	0.75 (0.48 to 1.15)
CCI, 0 score					
	None	597,115 (82.47)	3.3	1.00 (reference)	1.00 (reference)
	Patients with COVID-19	126,952 (17.53)	2.9	0.88 (0.72 to 1.06)	0.88 (0.72 to 1.06)
CCI, 1 scores					
	None	63,117 (82.15)	5.9	1.00 (reference)	1.00 (reference)
	Patients with COVID-19	13,712 (17.85)	3.8	0.65 (0.41 to 1.04)	0.65 (0.41 to 1.04)
CCI, ≥2 scores					
	None	28,108 (79.70)	4.8	1.00 (reference)	1.00 (reference)
	Patients with COVID-19	7,160 (20.30)	4.7	0.99 (0.52 to 1.91)	0.95 (0.49 to 1.84)
BMI, <18.5 kg/m²					
	None	17,997 (81.42)	2.9	1.00 (reference)	1.00 (reference)
	Patients with COVID-19	4,108 (18.58)	5.9	2.01 (0.87 to 4.62)	2.14 (0.92 to 4.99)

BMI, 18.5-23.0 kg/m²					
	None	220,919 (82.24)	3.3	1.00 (reference)	1.00 (reference)
	Patients with COVID-19	47,707 (17.76)	2.9	0.89 (0.65 to 1.21)	0.88 (0.65 to 1.20)
BMI, 23.0-25.0 kg/m²					
	None	159,861 (82.35)	3.9	1.00 (reference)	1.00 (reference)
	Patients with COVID-19	34,255 (17.65)	3.0	0.77 (0.54 to 1.10)	0.77 (0.54 to 1.10)
BMI, ≥25.0 kg/m²					
	None	289,452 (82.43)	3.7	1.00 (reference)	1.00 (reference)
	Patients with COVID-19	61,718 (17.57)	2.9	0.80 (0.61 to 1.05)	0.80 (0.61 to 1.05)
					
Drinker					
	None	293,797 (82.42)	3.2	1.00 (reference)	1.00 (reference)
	Patients with COVID-19	62,665 (17.58)	2.5	0.78 (0.58 to 1.04)	0.77 (0.58 to 1.03)
Non-drinker					
	None	394,032 (82.24)	3.9	1.00 (reference)	1.00 (reference)
	Patients with COVID-19	85,120 (17.76)	3.4	0.89 (0.72 to 1.10)	0.89 (0.71 to 1.10)
					
Sufficient physical activity					
	None	340,507 (82.39)	3.6	1.00 (reference)	1.00 (reference)
	Patients with COVID-19	72,760 (17.61)	3.2	0.87 (0.68 to 1.11)	0.87 (0.68 to 1.11)
Insufficient physical activity					
	None	347,204 (82.24)	3.6	1.00 (reference)	1.00 (reference)

	Patients with COVID-19	74,978 (17.76)	2.9	0.82 (0.64 to 1.06)	0.82 (0.64 to 1.05)
Smoker					
	None	251,296 (82.46)	3.3	1.00 (reference)	1.00 (reference)
	Patients with COVID-19	53,445 (17.54)	2.5	0.78 (0.57 to 1.06)	0.78 (0.57 to 1.06)
Non-smoker					
	None	436,584 (82.23)	3.8	1.00 (reference)	1.00 (reference)
	Patients with COVID-19	94,341 (17.77)	3.3	0.88 (0.72 to 1.08)	0.88 (0.72 to 1.08)
Rural residence					
	None	368,924 (82.35)	3.7	1.00 (reference)	1.00 (reference)
	Patients with COVID-19	79,097 (17.65)	3.4	0.92 (0.73 to 1.16)	0.92 (0.73 to 1.15)
Urban residence					
	None	319,416 (82.29)	3.5	1.00 (reference)	1.00 (reference)
	Patients with COVID-19	68,727 (17.71)	2.7	0.76 (0.58 to 0.99)	0.76 (0.58 to 0.99)
Strain type (original)					
	None	220,904 (82.49)	3.9	1.00 (reference)	1.00 (reference)
	Patients with COVID-19	46,900 (17.51)	3.4	0.88 (0.73 to 1.07)	0.88 (0.73 to 1.07)
Strain type (delta)					
	None	467,436 (82.24)	2.9	1.00 (reference)	1.00 (reference)
	Patients with COVID-19	100,924 (17.76)	2.1	0.73 (0.51 to 1.06)	0.73 (0.50 to 1.06)

BMI, body mass index; CCI, charlson comorbidity index; CI, confidence interval; HR, hazard ratio.

* Incidence rate is expressed as per 1,000 person-years.

The data in bold indicate significant differences ($P < 0.05$).

^a **Model 1:** Adjusted for age (20–39, 40–59, and ≥ 60 years) and sex.

^b **Model 2:** Adjusted for age (20–39, 40–59, and ≥ 60 years), sex, household income (low income, middle income, and high income), region of residence (urban and rural), Charlson comorbidity index (0, 1, and ≥ 2), obesity (underweight [$< 18.5 \text{ kg/m}^2$], normal [$18.5\text{--}23.0 \text{ kg/m}^2$], overweight [$23.0\text{--}25.0 \text{ kg/m}^2$], obese [$\geq 25.0 \text{ kg/m}^2$], and unknown), blood pressure (systolic blood pressure $< 140 \text{ mmHg}$ and diastolic blood pressure $< 90 \text{ mmHg}$, systolic blood pressure $\geq 140 \text{ mmHg}$ or diastolic blood pressure $\geq 90 \text{ mmHg}$, and unknown), fasting blood glucose (< 100 , $\geq 100 \text{ mg/dL}$, and unknown), serum total cholesterol (< 200 , $200\text{--}240$, $\geq 240 \text{ mg/dL}$, and unknown), glomerular filtration rate (< 60 , $60\text{--}90$, $\geq 90 \text{ mL/min/1.73 m}^2$, and unknown), smoking status (non-, ex-, current smoker, and unknown), alcoholic drinks (< 1 , $1\text{--}2$, $3\text{--}4$, ≥ 5 days per week, and unknown), aerobic physical activity (sufficient, insufficient, and unknown), previous history of cardiovascular disease, chronic kidney disease, and chronic obstructive pulmonary disease, history of medication use for diabetes mellitus, dyslipidemia, and hypertension, and missing indicators (obesity missing indicator [yes or no], blood pressure missing indicator [yes or no], fasting blood glucose missing indicator [yes or no], serum total cholesterol missing indicator [yes or no], glomerular filtration rate missing indicator [yes or no], smoking status missing indicator [yes or no], alcoholic drinks missing indicator [yes or no], and aerobic physical activity missing indicator [yes or no]).

Table S15. The HR with 95% CI for the long-term sequelae risk of incident allergic diseases following COVID-19 diagnosis in the propensity score-matched replication cohort A (Japan).

Parameter	Exposure	n (%)	Incidence rate*	HR (95% CI)	
				Model 1 ^a	Model 2 ^b
Total					
	None	1,998,524 (78.65)	27.4	1.00 (reference)	1.00 (reference)
	Patients with COVID-19	542,497 (21.35)	75.1	2.65 (2.61 to 2.69)	2.56 (2.52 to 2.59)
Male					
	None	1,352,980 (79.00)	23.8	1.00 (reference)	1.00 (reference)
	Patients with COVID-19	359,753 (21.00)	66.7	2.74 (2.69 to 2.80)	2.63 (2.58 to 2.68)
Female					
	None	645,544 (77.94)	35.0	1.00 (reference)	1.00 (reference)
	Patients with COVID-19	182,744 (22.06)	91.9	2.52 (2.46 to 2.58)	2.45 (2.39 to 2.50)
Age 20–39 years					
	None	489,756 (72.28)	30.9	1.00 (reference)	1.00 (reference)
	Patients with COVID-19	187,828 (27.72)	85.9	2.71 (2.64 to 2.78)	2.67 (2.60 to 2.74)
Age 40–59 years					
	None	1,238,373 (81.05)	25.3	1.00 (reference)	1.00 (reference)
	Patients with COVID-19	289,616 (18.95)	69.5	2.72 (2.67 to 2.77)	2.60 (2.54 to 2.65)
Age ≥60 years					
	None	270,395 (80.61)	30.4	1.00 (reference)	1.00 (reference)

	Patients with COVID-19	65,053 (19.39)	68.7	2.26 (2.17 to 2.35)	2.12 (2.03 to 2.21)
CCI, 0 score					
	None	1,976,142 (79.13)	27.3	1.00 (reference)	1.00 (reference)
	Patients with COVID-19	521,111 (20.87)	74.7	2.64 (2.60 to 2.68)	2.56 (2.52 to 2.60)
CCI, 1 scores					
	None	7,141 (52.03)	36.6	1.00 (reference)	1.00 (reference)
	Patients with COVID-19	6,583 (47.97)	83.7	2.40 (2.05 to 2.80)	2.39 (2.04 to 2.80)
CCI, ≥2 scores					
	None	15,241 (50.73)	32.8	1.00 (reference)	1.00 (reference)
	Patients with COVID-19	14,803 (49.27)	85.0	2.56 (2.30 to 2.85)	2.40 (2.15 to 2.67)
BMI, <18.5 kg/m²					
	None	146,377 (77.43)	29.3	1.00 (reference)	1.00 (reference)
	Patients with COVID-19	42,662 (22.57)	83.3	2.72 (2.59 to 2.86)	2.63 (2.50 to 2.77)
BMI, 18.5-23.0 kg/m²					
	None	864,919 (78.40)	28.0	1.00 (reference)	1.00 (reference)
	Patients with COVID-19	238,313 (21.60)	76.6	2.61 (2.56 to 2.67)	2.54 (2.48 to 2.60)
BMI, 23.0-25.0 kg/m²					
	None	396,921 (79.36)	26.8	1.00 (reference)	1.00 (reference)
	Patients with COVID-19	103,239 (20.64)	72.6	2.64 (2.55 to 2.73)	2.55 (2.46 to 2.64)
BMI, ≥25.0 kg/m²					

	None	580,305 (78.80)	26.9	1.00 (reference)	1.00 (reference)
	Patients with COVID-19	156,101 (21.20)	72.2	2.65 (2.58 to 2.72)	2.53 (2.46 to 2.60)
Non-drinker					
	None	774,477 (78.85)	28.2	1.00 (reference)	1.00 (reference)
	Patients with COVID-19	207,717 (21.15)	81.4	2.80 (2.73 to 2.86)	2.70 (2.63 to 2.76)
Drinker					
	None	1,113,296 (78.72)	27.1	1.00 (reference)	1.00 (reference)
	Patients with COVID-19	300,882 (21.28)	71.0	2.54 (2.49 to 2.59)	2.45 (2.40 to 2.50)
Sufficient physical activity					
	None	435,493 (79.73)	26.8	1.00 (reference)	1.00 (reference)
	Patients with COVID-19	110,731 (20.27)	68.3	2.50 (2.42 to 2.58)	2.41 (2.33 to 2.50)
Insufficient physical activity					
	None	1,430,215 (78.51)	27.8	1.00 (reference)	1.00 (reference)
	Patients with COVID-19	391,479 (21.49)	77.1	2.68 (2.63 to 2.72)	2.58 (2.54 to 2.63)

BMI, body mass index; CCI, charlson comorbidity index; CI, confidence interval; HR, hazard ratio.

* Incidence rate is expressed as per 1,000 person-years.

The data in bold indicate significant differences ($P < 0.05$).

^a **Model 1:** Adjusted for age (20–39, 40–59, and ≥ 60 years) and sex.

^b **Model 2:** Adjusted for age (20–39, 40–59, and ≥ 60 years); sex; Charlson comorbidity index (0, 1, and ≥ 2); BMI (underweight [< 18.5 kg/m²], normal [18.5–23.0 kg/m²], overweight [23.0–25.0 kg/m²], obese [≥ 25.0 kg/m²], and unknown); blood pressure (systolic blood pressure < 140 mmHg and diastolic blood pressure < 90 mmHg, systolic blood pressure ≥ 140 mmHg or diastolic blood pressure ≥ 90 mmHg, and unknown); fasting blood glucose (< 100 , ≥ 100 mg/dL, and unknown); serum total cholesterol (< 200 , 200–240, ≥ 240 mg/dL, and unknown); glomerular filtration rate (< 60 , 60–90, ≥ 90 mL/min/1.73 m², and unknown); smoking status (non- and current smoker, and unknown); alcoholic drinks (drinks; < 1 , 1–2, 3–4, ≥ 5 days per week, and unknown); aerobic physical activity (sufficient, insufficient, and unknown); previous history of cardiovascular disease, chronic kidney disease, and chronic obstructive pulmonary disease; history of medication use for diabetes mellitus, dyslipidemia, and hypertension; and missing indicators (BMI missing indicator [yes or no], blood pressure missing indicator [yes or no], fasting blood glucose missing indicator [yes or no], serum total cholesterol missing indicator [yes or no], glomerular filtration rate missing indicator [yes or no], smoking status missing indicator [yes or no], alcoholic drinks missing indicator [yes or no], and aerobic physical activity missing indicator [yes or no]).

Table S16. The HR with 95% CI for the long-term sequelae risk of incident asthma following COVID-19 diagnosis in the propensity score-matched replication cohort A (Japan).

Parameter	Exposure	n (%)	Incidence rate*	HR (95% CI)	
				Model 1 ^a	Model 2 ^b
Total					
	None	1,998,524 (78.65)	2.7	1.00 (reference)	1.00 (reference)
	Patients with COVID-19	542,497 (21.35)	7.6	2.77 (2.65 to 2.90)	2.63 (2.51 to 2.75)
Male					
	None	1,352,980 (79.00)	2.3	1.00 (reference)	1.00 (reference)
	Patients with COVID-19	359,753 (21.00)	6.7	2.81 (2.65 to 2.98)	2.66 (2.51 to 2.83)
Female					
	None	645,544 (77.94)	3.4	1.00 (reference)	1.00 (reference)
	Patients with COVID-19	182,744 (22.06)	9.5	2.70 (2.52 to 2.90)	2.57 (2.39 to 2.76)
Age 20–39 years					
	None	489,756 (72.28)	3.0	1.00 (reference)	1.00 (reference)
	Patients with COVID-19	187,828 (27.72)	9.1	2.96 (2.74 to 3.20)	2.91 (2.70 to 3.15)
Age 40–59 years					
	None	1,238,373 (81.05)	2.6	1.00 (reference)	1.00 (reference)
	Patients with COVID-19	289,616 (18.95)	7.1	2.75 (2.58 to 2.92)	2.58 (2.42 to 2.74)
Age ≥60 years					
	None	270,395 (80.61)	2.6	1.00 (reference)	1.00 (reference)
	Patients with COVID-19	65,053 (19.39)	6.1	2.31 (2.02 to 2.64)	2.07 (1.80 to 2.38)
CCI, 0 score					

	None	1,976,142 (79.13)	2.7	1.00 (reference)	1.00 (reference)
	Patients with COVID-19	521,111 (20.87)	7.6	2.74 (2.62 to 2.87)	2.62 (2.51 to 2.75)
CCI, 1 scores	None	7,141 (52.03)	3.8	1.00 (reference)	1.00 (reference)
	Patients with COVID-19	6,583 (47.97)	10.7	3.07 (1.93 to 4.87)	2.93 (1.84 to 4.67)
CCI, ≥2 scores	None	15,241 (50.73)	2.6	1.00 (reference)	1.00 (reference)
	Patients with COVID-19	14,803 (49.27)	7.8	3.00 (2.07 to 4.34)	2.73 (1.87 to 3.98)
					
BMI, <18.5 kg/m²	None	146,377 (77.43)	2.9	1.00 (reference)	1.00 (reference)
	Patients with COVID-19	42,662 (22.57)	8.6	2.88 (2.47 to 3.36)	2.72 (2.33 to 3.18)
BMI, 18.5-23.0 kg/m²	None	864,919 (78.40)	2.5	1.00 (reference)	1.00 (reference)
	Patients with COVID-19	238,313 (21.60)	7.7	2.95 (2.76 to 3.17)	2.83 (2.64 to 3.04)
BMI, 23.0-25.0 kg/m²	None	396,921 (79.36)	2.6	1.00 (reference)	1.00 (reference)
	Patients with COVID-19	103,239 (20.64)	7.0	2.60 (2.34 to 2.89)	2.49 (2.24 to 2.77)
BMI, ≥25.0 kg/m²	None	580,305 (78.80)	2.9	1.00 (reference)	1.00 (reference)
	Patients with COVID-19	156,101 (21.20)	7.7	2.56 (2.36 to 2.78)	2.41 (2.21 to 2.61)
					
Non-drinker	None	774,477 (78.85)	2.9	1.00 (reference)	1.00 (reference)
	Patients with COVID-19	207,717 (21.15)	8.8	2.95 (2.76 to 3.16)	2.80 (2.61 to 3.00)

Drinker	None	1,113,296 (78.72)	2.6	1.00 (reference)	1.00 (reference)
	Patients with COVID-19	300,882 (21.28)	6.8	2.61 (2.45 to 2.78)	2.47 (2.32 to 2.63)
					
Sufficient physical activity	None	435,493 (79.73)	2.4	1.00 (reference)	1.00 (reference)
	Patients with COVID-19	110,731 (20.27)	5.8	2.42 (2.16 to 2.70)	2.29 (2.04 to 2.56)
Insufficient physical activity	None	1,430,215 (78.51)	2.8	1.00 (reference)	1.00 (reference)
	Patients with COVID-19	391,479 (21.49)	8.1	2.82 (2.68 to 2.97)	2.68 (2.54 to 2.83)

BMI, body mass index; CCI, charlson comorbidity index; CI, confidence interval; HR, hazard ratio.

* Incidence rate is expressed as per 1,000 person-years.

The data in bold indicate significant differences ($P < 0.05$).

^a **Model 1:** Adjusted for age (20–39, 40–59, and ≥ 60 years) and sex.

^b **Model 2:** Adjusted for age (20–39, 40–59, and ≥ 60 years); sex; Charlson comorbidity index (0, 1, and ≥ 2); BMI (underweight [$< 18.5 \text{ kg/m}^2$], normal [$18.5\text{--}23.0 \text{ kg/m}^2$], overweight [$23.0\text{--}25.0 \text{ kg/m}^2$], obese [$\geq 25.0 \text{ kg/m}^2$], and unknown); blood pressure (systolic blood pressure < 140 mmHg and diastolic blood pressure < 90 mmHg, systolic blood pressure ≥ 140 mmHg or diastolic blood pressure ≥ 90 mmHg, and unknown); fasting blood glucose (< 100 , ≥ 100 mg/dL, and unknown); serum total cholesterol (< 200 , $200\text{--}240$, ≥ 240 mg/dL, and unknown); glomerular filtration rate (< 60 , $60\text{--}90$, ≥ 90 mL/min/1.73 m^2 , and unknown); smoking status (non- and current smoker, and unknown); alcoholic drinks (drinks; < 1 , $1\text{--}2$, $3\text{--}4$, ≥ 5 days per week, and unknown); aerobic physical activity (sufficient, insufficient, and unknown); previous history of

cardiovascular disease, chronic kidney disease, and chronic obstructive pulmonary disease; history of medication use for diabetes mellitus, dyslipidemia, and hypertension; and missing indicators (BMI missing indicator [yes or no], blood pressure missing indicator [yes or no], fasting blood glucose missing indicator [yes or no], serum total cholesterol missing indicator [yes or no], glomerular filtration rate missing indicator [yes or no], smoking status missing indicator [yes or no], alcoholic drinks missing indicator [yes or no], and aerobic physical activity missing indicator [yes or no]).

Table S17. The HR with 95% CI for the long-term sequelae risk of incident allergic rhinitis following COVID-19 diagnosis in the propensity score-matched replication cohort A (Japan).

Parameter	Exposure	n (%)	Incidence rate*	HR (95% CI)	
				Model 1 ^a	Model 2 ^b
Total					
	None	1,998,524 (78.65)	20.4	1.00 (reference)	1.00 (reference)
	Patients with COVID-19	542,497 (21.35)	62.6	2.98 (2.93 to 3.03)	2.88 (2.83 to 2.93)
Male					
	None	1,352,980 (79.00)	17.9	1.00 (reference)	1.00 (reference)
	Patients with COVID-19	359,753 (21.00)	55.7	3.05 (2.99 to 3.12)	2.94 (2.88 to 3.01)
Female					
	None	645,544 (77.94)	25.6	1.00 (reference)	1.00 (reference)
	Patients with COVID-19	182,744 (22.06)	76.1	2.86 (2.79 to 2.94)	2.79 (2.72 to 2.86)
Age 20–39 years					
	None	489,756 (72.28)	21.9	1.00 (reference)	1.00 (reference)
	Patients with COVID-19	187,828 (27.72)	72.2	3.20 (3.11 to 3.29)	3.16 (3.07 to 3.25)
Age 40–59 years					
	None	1,238,373 (81.05)	18.9	1.00 (reference)	1.00 (reference)
	Patients with COVID-19	289,616 (18.95)	57.7	3.02 (2.95 to 3.09)	2.90 (2.83 to 2.96)
Age ≥60 years					
	None	270,395 (80.61)	24.2	1.00 (reference)	1.00 (reference)
	Patients with COVID-19	65,053 (19.39)	56.5	2.33 (2.23 to 2.44)	2.22 (2.12 to 2.32)
CCI, 0 score					

	None	1,976,142 (79.13)	20.4	1.00 (reference)	1.00 (reference)
	Patients with COVID-19	521,111 (20.87)	62.5	2.97 (2.92 to 3.02)	2.89 (2.84 to 2.94)
CCI, 1 scores					
	None	7,141 (52.03)	27.6	1.00 (reference)	1.00 (reference)
	Patients with COVID-19	6,583 (47.97)	67.2	2.55 (2.14 to 3.04)	2.56 (2.14 to 3.05)
CCI, ≥2 scores					
	None	15,241 (50.73)	23.3	1.00 (reference)	1.00 (reference)
	Patients with COVID-19	14,803 (49.27)	63.5	2.69 (2.37 to 3.05)	2.53 (2.22 to 2.88)
					
BMI, <18.5 kg/m²					
	None	146,377 (77.43)	20.1	1.00 (reference)	1.00 (reference)
	Patients with COVID-19	42,662 (22.57)	68.0	3.24 (3.06 to 3.43)	3.13 (2.96 to 3.32)
BMI, 18.5-23.0 kg/m²					
	None	864,919 (78.40)	20.5	1.00 (reference)	1.00 (reference)
	Patients with COVID-19	238,313 (21.60)	63.7	2.98 (2.91 to 3.05)	2.91 (2.83 to 2.98)
BMI, 23.0-25.0 kg/m²					
	None	396,921 (79.36)	20.3	1.00 (reference)	1.00 (reference)
	Patients with COVID-19	103,239 (20.64)	61.1	2.94 (2.83 to 3.05)	2.86 (2.75 to 2.97)
BMI, ≥25.0 kg/m²					
	None	580,305 (78.80)	20.8	1.00 (reference)	1.00 (reference)
	Patients with COVID-19	156,101 (21.20)	60.3	2.87 (2.78 to 2.95)	2.76 (2.67 to 2.84)
					
Non-drinker					

Drinker	None	774,477 (78.85)	20.5	1.00 (reference)	1.00 (reference)
	Patients with COVID-19	207,717 (21.15)	67.7	3.20 (3.12 to 3.28)	3.10 (3.02 to 3.18)
	None	1,113,296 (78.72)	20.6	1.00 (reference)	1.00 (reference)
	Patients with COVID-19	300,882 (21.28)	59.1	2.80 (2.74 to 2.86)	2.71 (2.65 to 2.77)
					
Sufficient physical activity					
	None	435,493 (79.73)	20.4	1.00 (reference)	1.00 (reference)
	Patients with COVID-19	110,731 (20.27)	57.2	2.76 (2.66 to 2.86)	2.68 (2.58 to 2.78)
Insufficient physical activity					
	None	1,430,215 (78.51)	20.6	1.00 (reference)	1.00 (reference)
	Patients with COVID-19	391,479 (21.49)	64.2	3.01 (2.95 to 3.07)	2.91 (2.86 to 2.97)

BMI, body mass index; CCI, charlson comorbidity index; CI, confidence interval; HR, hazard ratio.

* Incidence rate is expressed as per 1,000 person-years.

The data in bold indicate significant differences ($P < 0.05$).

^a **Model 1:** Adjusted for age (20–39, 40–59, and ≥ 60 years) and sex.

^b **Model 2:** Adjusted for age (20–39, 40–59, and ≥ 60 years); sex; Charlson comorbidity index (0, 1, and ≥ 2); BMI (underweight [$< 18.5 \text{ kg/m}^2$], normal [$18.5\text{--}23.0 \text{ kg/m}^2$], overweight [$23.0\text{--}25.0 \text{ kg/m}^2$], obese [$\geq 25.0 \text{ kg/m}^2$], and unknown); blood pressure (systolic blood pressure $< 140 \text{ mmHg}$ and diastolic blood pressure $< 90 \text{ mmHg}$, systolic blood pressure $\geq 140 \text{ mmHg}$ or diastolic blood pressure $\geq 90 \text{ mmHg}$, and unknown); fasting blood glucose (< 100 , $\geq 100 \text{ mg/dL}$, and unknown); serum total cholesterol (< 200 , $200\text{--}240$, $\geq 240 \text{ mg/dL}$, and unknown); glomerular

filtration rate (<60, 60–90, ≥90 mL/min/1.73 m², and unknown); smoking status (non- and current smoker, and unknown); alcoholic drinks (drinks; <1, 1–2, 3–4, ≥5 days per week, and unknown); aerobic physical activity (sufficient, insufficient, and unknown); previous history of cardiovascular disease, chronic kidney disease, and chronic obstructive pulmonary disease; history of medication use for diabetes mellitus, dyslipidemia, and hypertension; and missing indicators (BMI missing indicator [yes or no], blood pressure missing indicator [yes or no], fasting blood glucose missing indicator [yes or no], serum total cholesterol missing indicator [yes or no], glomerular filtration rate missing indicator [yes or no], smoking status missing indicator [yes or no], alcoholic drinks missing indicator [yes or no], and aerobic physical activity missing indicator [yes or no]).

Table S18. The HR with 95% CI for the long-term sequelae risk of incident atopic dermatitis following COVID-19 diagnosis in the propensity score-matched replication cohort A (Japan).

Parameter	Exposure	n (%)	Incidence rate*	HR (95% CI)	
				Model 1 ^a	Model 2 ^b
Total					
	None	1,998,524 (78.65)	5.2	1.00 (reference)	1.00 (reference)
	Patients with COVID-19	542,497 (21.35)	6.8	1.25 (1.20 to 1.30)	1.21 (1.16 to 1.26)
Male					
	None	1,352,980 (79.00)	4.2	1.00 (reference)	1.00 (reference)
	Patients with COVID-19	359,753 (21.00)	5.7	1.31 (1.24 to 1.38)	1.26 (1.19 to 1.33)
Female					
	None	645,544 (77.94)	7.2	1.00 (reference)	1.00 (reference)
	Patients with COVID-19	182,744 (22.06)	9.0	1.18 (1.11 to 1.26)	1.16 (1.09 to 1.23)
Age 20–39 years					
	None	489,756 (72.28)	7.5	1.00 (reference)	1.00 (reference)
	Patients with COVID-19	187,828 (27.72)	8.3	1.09 (1.02 to 1.16)	1.08 (1.01 to 1.15)
Age 40–59 years					
	None	1,238,373 (81.05)	4.6	1.00 (reference)	1.00 (reference)
	Patients with COVID-19	289,616 (18.95)	6.2	1.35 (1.27 to 1.44)	1.30 (1.23 to 1.38)
Age ≥60 years					
	None	270,395 (80.61)	3.7	1.00 (reference)	1.00 (reference)
	Patients with COVID-19	65,053 (19.39)	5.4	1.50 (1.31 to 1.71)	1.41 (1.23 to 1.62)
CCI, 0 score					

	None	1,976,142 (79.13)	5.2	1.00 (reference)	1.00 (reference)
	Patients with COVID-19	521,111 (20.87)	6.7	1.22 (1.17 to 1.27)	1.20 (1.15 to 1.25)
CCI, 1 scores					
	None	7,141 (52.03)	4.5	1.00 (reference)	1.00 (reference)
	Patients with COVID-19	6,583 (47.97)	7.1	1.70 (1.07 to 2.71)	1.66 (1.04 to 2.66)
CCI, ≥2 scores					
	None	15,241 (50.73)	6.0	1.00 (reference)	1.00 (reference)
	Patients with COVID-19	14,803 (49.27)	9.6	1.59 (1.21 to 2.08)	1.44 (1.10 to 1.90)
					
BMI, <18.5 kg/m²					
	None	146,377 (77.43)	7.4	1.00 (reference)	1.00 (reference)
	Patients with COVID-19	42,662 (22.57)	9.2	1.18 (1.03 to 1.34)	1.16 (1.02 to 1.33)
BMI, 18.5-23.0 kg/m²					
	None	864,919 (78.40)	5.8	1.00 (reference)	1.00 (reference)
	Patients with COVID-19	238,313 (21.60)	7.4	1.19 (1.12 to 1.26)	1.16 (1.09 to 1.24)
BMI, 23.0-25.0 kg/m²					
	None	396,921 (79.36)	4.7	1.00 (reference)	1.00 (reference)
	Patients with COVID-19	103,239 (20.64)	6.5	1.30 (1.18 to 1.44)	1.26 (1.14 to 1.39)
BMI, ≥25.0 kg/m²					
	None	580,305 (78.80)	4.0	1.00 (reference)	1.00 (reference)
	Patients with COVID-19	156,101 (21.20)	5.7	1.36 (1.25 to 1.48)	1.30 (1.19 to 1.41)
					
Non-drinker					

Drinker	None	774,477 (78.85)	5.6	1.00 (reference)	1.00 (reference)
	Patients with COVID-19	207,717 (21.15)	7.3	1.22 (1.15 to 1.31)	1.19 (1.12 to 1.27)
	None	1,113,296 (78.72)	4.9	1.00 (reference)	1.00 (reference)
	Patients with COVID-19	300,882 (21.28)	6.6	1.27 (1.20 to 1.35)	1.24 (1.17 to 1.31)
					
Sufficient physical activity					
	None	435,493 (79.73)	4.7	1.00 (reference)	1.00 (reference)
	Patients with COVID-19	110,731 (20.27)	6.1	1.23 (1.12 to 1.36)	1.20 (1.09 to 1.33)
Insufficient physical activity					
	None	1,430,215 (78.51)	5.3	1.00 (reference)	1.00 (reference)
	Patients with COVID-19	391,479 (21.49)	7.1	1.26 (1.20 to 1.32)	1.22 (1.17 to 1.29)

BMI, body mass index; CCI, charlson comorbidity index; CI, confidence interval; HR, hazard ratio.

* Incidence rate is expressed as per 1,000 person-years.

The data in bold indicate significant differences ($P < 0.05$).

^a **Model 1:** Adjusted for age (20–39, 40–59, and ≥ 60 years) and sex.

^b **Model 2:** Adjusted for age (20–39, 40–59, and ≥ 60 years); sex; Charlson comorbidity index (0, 1, and ≥ 2); BMI (underweight [$< 18.5 \text{ kg/m}^2$], normal [$18.5\text{--}23.0 \text{ kg/m}^2$], overweight [$23.0\text{--}25.0 \text{ kg/m}^2$], obese [$\geq 25.0 \text{ kg/m}^2$], and unknown); blood pressure (systolic blood pressure < 140 mmHg and diastolic blood pressure < 90 mmHg, systolic blood pressure ≥ 140 mmHg or diastolic blood pressure ≥ 90 mmHg, and unknown); fasting blood glucose (< 100 , ≥ 100 mg/dL, and unknown); serum total cholesterol (< 200 , $200\text{--}240$, ≥ 240 mg/dL, and unknown); glomerular

filtration rate (<60, 60–90, ≥90 mL/min/1.73 m², and unknown); smoking status (non- and current smoker, and unknown); alcoholic drinks (drinks; <1, 1–2, 3–4, ≥5 days per week, and unknown); aerobic physical activity (sufficient, insufficient, and unknown); previous history of cardiovascular disease, chronic kidney disease, and chronic obstructive pulmonary disease; history of medication use for diabetes mellitus, dyslipidemia, and hypertension; and missing indicators (BMI missing indicator [yes or no], blood pressure missing indicator [yes or no], fasting blood glucose missing indicator [yes or no], serum total cholesterol missing indicator [yes or no], glomerular filtration rate missing indicator [yes or no], smoking status missing indicator [yes or no], alcoholic drinks missing indicator [yes or no], and aerobic physical activity missing indicator [yes or no]).

Table S19. The HR with 95% CI for the long-term sequelae risk of incident food allergy following COVID-19 diagnosis in the propensity score-matched replication cohort A (Japan).

Parameter	Exposure	n (%)	Incidence rate*	HR (95% CI)	
				Model 1 ^a	Model 2 ^b
Total	None	1,998,524 (78.65)	1.3	1.00 (reference)	1.00 (reference)
	Patients with COVID-19	542,497 (21.35)	2.7	2.10 (1.95 to 2.25)	1.84 (1.71 to 1.98)
Male	None	1,352,980 (79.00)	1.0	1.00 (reference)	1.00 (reference)
	Patients with COVID-19	359,753 (21.00)	2.5	2.46 (2.24 to 2.70)	2.09 (1.89 to 2.30)
Female	None	645,544 (77.94)	1.9	1.00 (reference)	1.00 (reference)
	Patients with COVID-19	182,744 (22.06)	3.2	1.70 (1.52 to 1.90)	1.56 (1.40 to 1.75)
Age 20–39 years	None	489,756 (72.28)	1.3	1.00 (reference)	1.00 (reference)
	Patients with COVID-19	187,828 (27.72)	2.3	1.74 (1.52 to 1.99)	1.65 (1.44 to 1.88)
Age 40–59 years	None	1,238,373 (81.05)	1.2	1.00 (reference)	1.00 (reference)
	Patients with COVID-19	289,616 (18.95)	2.7	2.22 (2.02 to 2.45)	1.94 (1.76 to 2.14)
Age ≥60 years	None	270,395 (80.61)	1.6	1.00 (reference)	1.00 (reference)
	Patients with COVID-19	65,053 (19.39)	3.8	2.37 (2.00 to 2.81)	1.90 (1.59 to 2.28)
CCI, 0 score	None	1,976,142 (79.13)	1.3	1.00 (reference)	1.00 (reference)
	Patients with COVID-19	521,111 (20.87)	2.5	1.95 (1.80 to 2.10)	1.80 (1.67 to 1.95)

CCI, 1 scores					
	None	7,141 (52.03)	3.1	1.00 (reference)	1.00 (reference)
	Patients with COVID-19	6,583 (47.97)	4.7	1.54 (0.88 to 2.70)	1.48 (0.84 to 2.61)
CCI, ≥2 scores					
	None	15,241 (50.73)	2.9	1.00 (reference)	1.00 (reference)
	Patients with COVID-19	14,803 (49.27)	8.9	3.04 (2.15 to 4.31)	2.83 (1.99 to 4.03)
					
BMI, <18.5 kg/m²					
	None	146,377 (77.43)	1.7	1.00 (reference)	1.00 (reference)
	Patients with COVID-19	42,662 (22.57)	3.5	2.13 (1.70 to 2.66)	1.86 (1.48 to 2.33)
BMI, 18.5-23.0 kg/m²					
	None	864,919 (78.40)	1.4	1.00 (reference)	1.00 (reference)
	Patients with COVID-19	238,313 (21.60)	2.6	1.84 (1.65 to 2.04)	1.63 (1.46 to 1.82)
BMI, 23.0-25.0 kg/m²					
	None	396,921 (79.36)	1.2	1.00 (reference)	1.00 (reference)
	Patients with COVID-19	103,239 (20.64)	2.5	2.13 (1.80 to 2.53)	1.88 (1.58 to 2.24)
BMI, ≥25.0 kg/m²					
	None	580,305 (78.80)	1.1	1.00 (reference)	1.00 (reference)
	Patients with COVID-19	156,101 (21.20)	2.7	2.51 (2.19 to 2.87)	2.17 (1.89 to 2.49)
					
Non-drinker					
	None	774,477 (78.85)	1.5	1.00 (reference)	1.00 (reference)
	Patients with COVID-19	207,717 (21.15)	2.9	1.97 (1.77 to 2.20)	1.69 (1.51 to 1.89)
Drinker					
	None	1,113,296 (78.72)	1.2	1.00 (reference)	1.00 (reference)
	Patients with COVID-19	300,882 (21.28)	2.5	2.16 (1.96 to 2.39)	1.92 (1.74 to 2.13)
					
Sufficient physical activity					
	None	435,493 (79.73)	1.3	1.00 (reference)	1.00 (reference)

Insufficient physical activity	Patients with COVID-19	110,731 (20.27)	2.8	2.23 (1.90 to 2.60)	1.90 (1.62 to 2.23)
	None	1,430,215 (78.51)	1.3	1.00 (reference)	1.00 (reference)
	Patients with COVID-19	391,479 (21.49)	2.7	2.02 (1.86 to 2.20)	1.78 (1.64 to 1.94)

BMI, body mass index; CCI, charlson comorbidity index; CI, confidence interval; HR, hazard ratio.

* Incidence rate is expressed as per 1,000 person-years.

The data in bold indicate significant differences ($P < 0.05$).

^a **Model 1:** Adjusted for age (20–39, 40–59, and ≥ 60 years) and sex.

^b **Model 2:** Adjusted for age (20–39, 40–59, and ≥ 60 years); sex; Charlson comorbidity index (0, 1, and ≥ 2); BMI (underweight [$< 18.5 \text{ kg/m}^2$], normal [$18.5\text{--}23.0 \text{ kg/m}^2$], overweight [$23.0\text{--}25.0 \text{ kg/m}^2$], obese [$\geq 25.0 \text{ kg/m}^2$], and unknown); blood pressure (systolic blood pressure $< 140 \text{ mmHg}$ and diastolic blood pressure $< 90 \text{ mmHg}$, systolic blood pressure $\geq 140 \text{ mmHg}$ or diastolic blood pressure $\geq 90 \text{ mmHg}$, and unknown); fasting blood glucose (< 100 , $\geq 100 \text{ mg/dL}$, and unknown); serum total cholesterol (< 200 , $200\text{--}240$, $\geq 240 \text{ mg/dL}$, and unknown); glomerular filtration rate (< 60 , $60\text{--}90$, $\geq 90 \text{ mL/min/1.73 m}^2$, and unknown); smoking status (non- and current smoker, and unknown); alcoholic drinks (drinks; < 1 , $1\text{--}2$, $3\text{--}4$, ≥ 5 days per week, and unknown); aerobic physical activity (sufficient, insufficient, and unknown); previous history of cardiovascular disease, chronic kidney disease, and chronic obstructive pulmonary disease; history of medication use for diabetes mellitus, dyslipidemia, and hypertension; and missing indicators (BMI missing indicator [yes or no], blood pressure missing indicator [yes or no], fasting blood glucose missing indicator [yes or no], serum total cholesterol missing indicator [yes or no], glomerular filtration rate missing indicator [yes or no], smoking status missing indicator [yes or no], alcoholic drinks missing indicator [yes or no], and aerobic physical activity missing indicator [yes or no]).

**Comment 5.**

*For unknown reason in K-CoV-N nearly all cases with unknown body mass index, blood*
*pressure, fasting glucose, glomerular filtration rate, smoking status, alcoholic drinks, days per*
*week and aerobic physical activity were found the COVID-19 group. In JMDC on the other site*
*nearly are unknown information are in the control group.*

**Response:**

Thank you for your incisive and important comment. We have come to understand that merely
assigning an 'unknown' value for analysis is insufficient for accurately estimating the results.
The variables you pointed out, which exhibit a varied distribution of unknowns, are parameters
derived from health examination results. These were not employed as matching variables in
our initial analysis. Furthermore, it's typical in health examination data for not all values to be
available for each parameter. Hence, performing the matching without these parameters could
potentially lead to different distributions between the two groups. However, after re-matching
by reorganizing the matching variables in accordance with the suggestions in comment 4, we
observed a significant reduction in the number of missing values, and the proportion of these
missing values was found to be similar regardless of COVID-19 infection status (Table 1 and
Table S4). Nevertheless, due to concerns about bias arising from missingness, we generated
missing indicator variables for those parameters in each cohort and incorporated them as
adjusted variables in model 2 for the reanalysis.

- ● Missing indicators: BMI missing indicator (yes or no), blood pressure missing indicator
(yes or no), fasting blood glucose missing indicator (yes or no), serum total cholesterol
missing indicator (yes or no), glomerular filtration rate missing indicator (yes or no),
smoking status missing indicator (yes or no), alcoholic drinks missing indicator (yes or
no), and aerobic physical activity missing indicator (yes or no).

Our analysis had some inadequacies, but thanks to your comment, it seems we have
achieved more reliable data and results. We have revised the manuscript to reflect the content
of your comment, and once again, we deeply appreciate your valuable insights.

**Changes in text:**

**Methods, Covariates**

Additionally, to minimize bias related to missing data, we focused on the missing indicator
method, generating missing indicator variables and incorporating them into the adjustment
variables.²⁰

**Reference**

20 Sperrin, M. & Martin, G. P. Multiple imputation with missing indicators as proxies for
unmeasured variables: simulation study. *BMC Med Res Methodol* **20**, 185 (2020).
[https://doi.org:10.1186/s12874-020-01068-x](https://doi.org/10.1186/s12874-020-01068-x)

**Tables 1-4 and Tables S4-S19**

We have applied missing indicators to the adjustment variables in all tables that involved
matching. (**Changes in text:** please refer to the attached table for Comment 4)

43 ^b **Model 2 (main cohort)**: Adjusted for age (20–39, 40–59, and ≥60 years); sex; household
income (low income, middle income, and high income); region of residence (urban and rural);
Charlson comorbidity index (0, 1, and ≥2); BMI (underweight [$<18.5 \text{ kg/m}^2$], normal [18.5 –
23.0 kg/m^2], overweight [23.0 – 25.0 kg/m^2], obese [$\geq 25.0 \text{ kg/m}^2$], and unknown); blood
pressure (systolic blood pressure $<140 \text{ mmHg}$ and diastolic blood pressure $<90 \text{ mmHg}$, systolic
blood pressure $\geq 140 \text{ mmHg}$ or diastolic blood pressure $\geq 90 \text{ mmHg}$, and unknown); fasting
blood glucose (<100 , $\geq 100 \text{ mg/dL}$, and unknown); serum total cholesterol (<200 , 200 – 240 ,
$\geq 240 \text{ mg/dL}$, and unknown); glomerular filtration rate (<60 , 60 – 90 , $\geq 90 \text{ mL/min/1.73 m}^2$, and
unknown); smoking status (non-, ex-, current smoker, and unknown); alcoholic drinks (<1 , 1 –

[revised manuscript text omitted]

**Formal comments:**

**Comment 6.**

*The authors uses „race and ethnicity“ throughout the text. This is uncommon the should use*
*one of these term.*

**Response:**

*As per your suggestion, we have revised all sentences that contained these terms to use only*
*“race”. Thank you for improving the quality of our manuscript.*

**Changes in text:**

**Introduction**

As race and ethnicity **is** suggested to be novel risk factors for developing long COVID,¹ we
constructed the cohort consisting of over 22 million participants using multinational cohort
studies of South Korea, Japan, and the UK.

**Discussion, Key findings of this study**

These findings were consistent for all three different national cohorts, indicating the long-
COVID effect on allergic diseases regardless of race and ethnicity.

**Comment 7.**

*The Figure S1 and Figure S2 are identical and UK Biobank is missing. If the selection was*
*done the same way for all cohorts one figure would be enough.*

**Response:**

Firstly, we sincerely apologize for the omission of the UK Biobank in our figures. We also
agree with your comment that the flow of each cohort used in the study is identical. Therefore,
in the revised supplementary material, we have combined the data into a single supplementary
figure (Figure S4) that covers all three cohorts. Thank you for bringing this to our attention.

**Changes in text:**

**Figure S4. Study flow of cohorts**

**Comment 8.**

*At the first mention of the "pre-observation period" it should be written that the duration is two*
*years.*

**Response:**

We agree that clarifying the duration of the 'pre-observation period' is important for a better
understanding. We have incorporated the detail that this period covers two years in the sentence
where it is first mentioned. Thank you for your feedback.

**Changes in text:**

**Methods**

***K-COV-N cohort (main)***

We precluded those who meet the following criteria: (1) insufficient socioeconomic
information or died before; and (2) history of allergic diseases in **the pre-observation period,**
**defined as two years** (n=4,335,150).

**Comment 9.**

*On page 11 the reference to table S2 is wrong as the table has only the ICD-10 Code in it.*

**Response:**

The justification for the statistical analysis mentioned in the sentence referencing Table S2 is
found in Table S20 (Table S3 in original version). We apologize for our mistake and any
confusion caused, and we appreciate your attention to this detail.

**Changes in text:**

**Methods, *Statistical analysis***

We provided justification of the **statistical analyses in Table S20.**

**Comment 10.**

*On page 12 the effect of AD was reported as insignificant, but the corresponding table 2*
*reported otherwise.*

**Response:**

The initial analysis data for AD was statistically significant, but due to our oversight, incorrect
results were reported. Upon reanalysis, we found that the values in the Results section should
be revised. Therefore, we have taken extra care this time to prevent similar errors. We deeply
regret any confusion caused by this and sincerely apologize.

**Changes in text:**

**Abstract**

After a 1:5 propensity score matching, we observed an elevated risk of developing allergic
diseases associated with COVID-19, beyond the 30 days after COVID-19 diagnosis (HR, 1.20;
95% CI, 1.13-1.27). This association was specifically shown in asthma (HR, 2.25; 95% CI,
1.80-2.83) and allergic rhinitis (HR, 1.23; 95% CI, 1.15-1.32).

**Result**

According to the maximally adjusted model (model 2) in Table 2, the increased risks of incident
overall allergic diseases (HR, 1.20; 95% CI, 1.13 to 1.27), asthma (HR, 2.25; 95% CI, 1.80 to
2.83) and AR (HR, 1.23; 95% CI, 1.15 to 1.32) were associated with SARS-CoV-2 infection;
however, no significant risk was observed in AD (HR, 1.15; 95% CI, 0.96 to 1.37) and FA (HR,
0.85; 95% CI, 0.71 to 1.00).

**Comment 11.**

*Unclear what “(n=3477).” should mean for reference 23*

**Response:**

We apologize for the confusion caused by mentioning the sample size of the ‘reference 23’.

Therefore, we have excluded it from the sentence. Thank you for your recommendation.

**Changes in text:**

**Discussion, *Comparisons with previous studies***

One recent study identified an elevated risk of new-onset AD after COVID-19 (~~n=3477~~).

**Comment 12.**

*As there are not prognostic measures used, the term's discovery/validation are misplaced. It is*
*advised to use replication cohort for the other cohorts or refer to each cohort as separate*
*entities as K-CoV-N, JMDC and UK Biobank.*

**Response:**

Thank you for your insightful feedback. We have revised all manuscripts and tables to refer to
the groups as the main cohort (South Korea), the replication cohort A (Japan), and the
replication cohort B (UK) instead of the term "discovery/validation" to reflect your comments.

**Changes in text:**

**Abstract**

We used the Korean population-based cohort (K-CoV-N; n=10,027,506) as a **main cohort** and
the Japanese claims-based cohort (JMDC; n=12,218,680; **replication cohort A**) and the UK
biobank cohort (UKB; n=468,617; **replication cohort B**).

Among the 10,027,506 individuals from the **main cohort** (mean age, 48.4 years [SD,13.4];
5,000,621 [49.9%] women), 313,955 patients were infected with SARS-CoV-2 during the
follow-up period.

Similar patterns of association were reported in the **replication cohorts A and B**.

**Methods, Data source**

[revised manuscript text omitted]

Cohort	Exposure	n (%)	Incidence rate*	HR (95% CI)	
				Model 1 ^a	Model 1 ^a
Allergic diseases					
Main cohort	None	688,340 (82.32)	24.0	1.00 (reference)	1.00 (reference)
Main cohort	Patients with COVID-19	147,824 (17.68)	28.7	1.20 (1.13 to 1.27)	1.20 (1.13 to 1.27)
Replication cohort A	None	1,998,524 (78.65)	27.4	1.00 (reference)	1.00 (reference)
Replication cohort A	Patients with COVID-19	542,497 (21.35)	75.1	2.65 (2.61 to 2.69)	2.56 (2.52 to 2.59)
Replication cohort B	None	248,949 (76.40)	7.2	1.00 (reference)	1.00 (reference)
Replication cohort B	Patients with COVID-19	76,894 (23.60)	8.3	1.14 (1.06 to 1.22)	1.12 (1.04 to 1.20)
Asthma					
Main cohort	None	688,340 (82.32)	1.0	1.00 (reference)	1.00 (reference)
Main cohort	Patients with COVID-19	147,824 (17.68)	2.2	2.27 (1.81 to 2.85)	2.25 (1.80 to 2.83)
Replication cohort A	None	1,998,524 (78.65)	2.7	1.00 (reference)	1.00 (reference)
Replication cohort A	Patients with COVID-19	542,497 (21.35)	7.6	2.77 (2.65 to 2.90)	2.63 (2.51 to 2.75)
Replication cohort B	None	248,949 (76.40)	3.2	1.00 (reference)	1.00 (reference)
Replication cohort B	Patients with COVID-19	76,894 (23.60)	3.8	1.16 (1.05 to 1.29)	1.14 (1.03 to 1.27)
Allergic rhinitis					
Main cohort	None	688,340 (82.32)	17.3	1.00 (reference)	1.00 (reference)
Main cohort	Patients with COVID-19	147,824 (17.68)	21.3	1.23 (1.15 to 1.32)	1.23 (1.15 to 1.32)
Replication cohort A	None	1,998,524 (78.65)	20.4	1.00 (reference)	1.00 (reference)
Replication cohort A	Patients with COVID-19	542,497 (21.35)	62.6	2.98 (2.93 to 3.03)	2.88 (2.83 to 2.93)
Replication cohort B	None	248,949 (76.40)	1.1	1.00 (reference)	1.00 (reference)
Replication cohort B	Patients with COVID-19	76,894 (23.60)	1.4	1.21 (1.02 to 1.44)	1.20 (1.01 to 1.43)
Atopic dermatitis					
Main cohort	None	688,340 (82.32)	2.6	1.00 (reference)	1.00 (reference)
Main cohort	Patients with COVID-19	147,824 (17.68)	3.0	1.15 (0.96 to 1.37)	1.15 (0.96 to 1.37)
Replication cohort A	None	1,998,524 (78.65)	5.2	1.00 (reference)	1.00 (reference)
Replication cohort A	Patients with COVID-19	542,497 (21.35)	6.8	1.25 (1.20 to 1.30)	1.21 (1.16 to 1.26)
Replication cohort B	None	248,949 (76.40)	0.04	1.00 (reference)	1.00 (reference)

Replication cohort B	Patients with COVID-19	76,894 (23.60)	0.05	1.18 (0.46 to 2.99)	1.19 (0.47 to 3.01)
Food allergy					
Main cohort	None	688,340 (82.32)	3.6	1.00 (reference)	1.00 (reference)
Main cohort	Patients with COVID-19	147,824 (17.68)	3.0	0.85 (0.71 to 1.01)	0.85 (0.71 to 1.00)
Replication cohort A	None	1,998,524 (78.65)	1.3	1.00 (reference)	1.00 (reference)
Replication cohort A	Patients with COVID-19	542,497 (21.35)	2.7	2.10 (1.95 to 2.25)	1.84 (1.71 to 1.98)
Replication cohort B	None	248,949 (76.40)	3.1	1.00 (reference)	1.00 (reference)
Replication cohort B	Patients with COVID-19	76,894 (23.60)	3.4	1.09 (0.97 to 1.21)	1.07 (0.96 to 1.20)

BMI, body mass index; CI, confidence interval; HR, hazard ratio.

* Incidence rate is expressed as per 1,000 person-years.

The data in bold indicate significant differences ($P < 0.05$).

^a **Model 1:** Adjusted for age (20–39, 40–59, and ≥ 60 years) and sex.

^b **Model 2 (main cohort):** Adjusted for age (20–39, 40–59, and ≥ 60 years); sex; household income (low income, middle income, and high income); region of residence (urban and rural); Charlson comorbidity index (0, 1, and ≥ 2); BMI (underweight [$< 18.5 \text{ kg/m}^2$], normal [$18.5\text{--}23.0 \text{ kg/m}^2$], overweight [$23.0\text{--}25.0 \text{ kg/m}^2$], obese [$\geq 25.0 \text{ kg/m}^2$], and unknown); blood pressure (systolic blood pressure $< 140 \text{ mmHg}$ and diastolic blood pressure $< 90 \text{ mmHg}$, systolic blood pressure $\geq 140 \text{ mmHg}$ or diastolic blood pressure $\geq 90 \text{ mmHg}$, and unknown); fasting blood glucose (< 100 , $\geq 100 \text{ mg/dL}$, and unknown); serum total cholesterol (< 200 , $200\text{--}240$, $\geq 240 \text{ mg/dL}$, and unknown); glomerular filtration rate (< 60 , $60\text{--}90$, $\geq 90 \text{ mL/min/1.73 m}^2$, and unknown); smoking status (non-, ex-, current smoker, and unknown); alcoholic drinks (< 1 , $1\text{--}2$, $3\text{--}4$, ≥ 5 days per week, and unknown); aerobic physical activity (sufficient, insufficient, and unknown); previous history of cardiovascular disease, chronic kidney disease, and chronic obstructive pulmonary disease; history of medication use for diabetes mellitus, dyslipidemia, and hypertension; and missing indicators (BMI missing indicator [yes or no], blood pressure missing indicator [yes or no], fasting blood glucose missing indicator [yes or no], serum total cholesterol missing indicator [yes or no], glomerular filtration rate missing indicator [yes or no], smoking status missing indicator [yes or no], alcoholic drinks missing indicator [yes or no], and aerobic physical activity missing indicator [yes or no]).

^b Model 2 (replication cohort A): Adjusted for age (20–39, 40–59, and ≥ 60 years); sex; Charlson comorbidity index (0, 1, and ≥ 2); BMI (underweight [$<18.5 \text{ kg/m}^2$], normal [$18.5\text{--}23.0 \text{ kg/m}^2$], overweight [$23.0\text{--}25.0 \text{ kg/m}^2$], obese [$\geq 25.0 \text{ kg/m}^2$], and unknown); blood pressure (systolic blood pressure $<140 \text{ mmHg}$ and diastolic blood pressure $<90 \text{ mmHg}$, systolic blood pressure $\geq 140 \text{ mmHg}$ or diastolic blood pressure $\geq 90 \text{ mmHg}$, and unknown); fasting blood glucose (<100 , $\geq 100 \text{ mg/dL}$, and unknown); serum total cholesterol (<200 , $200\text{--}240$, $\geq 240 \text{ mg/dL}$, and unknown); glomerular filtration rate (<60 , $60\text{--}90$, $\geq 90 \text{ mL/min/1.73 m}^2$, and unknown); smoking status (non- and current smoker, and unknown); alcoholic drinks (drinks; <1 , $1\text{--}2$, $3\text{--}4$, ≥ 5 days per week, and unknown); aerobic physical activity (sufficient, insufficient, and unknown); previous history of cardiovascular disease, chronic kidney disease, and chronic obstructive pulmonary disease; history of medication use for diabetes mellitus, dyslipidemia, and hypertension; and missing indicators (BMI missing indicator [yes or no], blood pressure missing indicator [yes or no], fasting blood glucose missing indicator [yes or no], serum total cholesterol missing indicator [yes or no], glomerular filtration rate missing indicator [yes or no], smoking status missing indicator [yes or no], alcoholic drinks missing indicator [yes or no], and aerobic physical activity missing indicator [yes or no]).

^b Model 2 (replication cohort B): Adjusted for age (20–39, 40–59, and ≥ 60 years); sex; household income ($<£18,000$, $£18,000\text{--}£30,999$, $£31,000\text{--}£51,999$, $£52,000\text{--}£100,000$, $>£100,000$), and unknown); region of residence (urban and rural); townsend deprivation index (T1 [least deprived], T2, T3 [most deprived], and unknown); ethnicity (white, mixed, Asian, black, others, and unknown); Charlson comorbidity index (0, 1, and ≥ 2); BMI (normal [$<25.0 \text{ kg/m}^2$], overweight [$25.0\text{--}30.0 \text{ kg/m}^2$], obese [$\geq 30.0 \text{ kg/m}^2$], and unknown); education levels (≤ 10 , $11\text{--}12$, >12 , and unknown); blood pressure (systolic blood pressure $< 140 \text{ mmHg}$ and diastolic blood pressure $< 90 \text{ mmHg}$, systolic blood pressure $\geq 140 \text{ mmHg}$ or diastolic blood pressure $\geq 90 \text{ mmHg}$, and unknown); fasting blood glucose (<100 , $\geq 100 \text{ mg/dL}$, and unknown); smoking status (non- and current smoker, and unknown); alcohol consumption (every day, sometimes, rarely days per week, and unknown); aerobic physical activity (low, moderate, high, and unknown); previous history of cardiovascular disease, chronic kidney disease, and chronic obstructive pulmonary disease; history of medication use for diabetes mellitus, dyslipidemia, and hypertension; and missing indicators (household income missing indicator [yes or no], townsend deprivation index missing indicator [yes or no], ethnicity missing indicator [yes or no], education levels [yes or no], obesity missing indicator [yes or no], blood pressure missing indicator [yes or no], fasting blood glucose missing indicator [yes or no], serum total cholesterol missing indicator [yes or no], glomerular filtration rate missing indicator [yes or no], smoking status missing indicator [yes or no], alcoholic drinks missing indicator [yes or no], and aerobic physical activity missing indicator [yes or no]).

Table 3. Time attenuation effect on the development of allergic diseases after SARS-CoV-2 infection (model 2 adjusted HR with 95% CI).

	HR (95% CI)	
	Main cohort (South Korea)	Replication cohort A (Japan)
Allergic diseases		
<3 months	1.42 (1.29 to 1.56)	3.30 (3.24 to 3.36)
3–6 months	1.14 (1.01 to 1.29)	1.77 (1.70 to 1.84)
≥6 months	1.00 (0.91 to 1.11)	1.61 (1.56 to 1.67)

BMI, body mass index; CI, confidence interval; HR, hazard ratio; SARS-CoV-2, severe acute respiratory syndrome coronavirus.

The data in bold indicate significant differences ($P < 0.05$).

Model 2 (main cohort): Adjusted for age (20–39, 40–59, and ≥60 years); sex; household income (low income, middle income, and high income); region of residence (urban and rural); Charlson comorbidity index (0, 1, and ≥2); BMI (underweight [$<18.5 \text{ kg/m}^2$], normal [$18.5\text{--}23.0 \text{ kg/m}^2$], overweight [$23.0\text{--}25.0 \text{ kg/m}^2$], obese [$\geq 25.0 \text{ kg/m}^2$], and unknown); blood pressure (systolic blood pressure $<140 \text{ mmHg}$ and diastolic blood pressure $<90 \text{ mmHg}$, systolic blood pressure $\geq 140 \text{ mmHg}$ or diastolic blood pressure $\geq 90 \text{ mmHg}$, and unknown); fasting blood glucose (<100 , $\geq 100 \text{ mg/dL}$, and unknown); serum total cholesterol (<200 , $200\text{--}240$, $\geq 240 \text{ mg/dL}$, and unknown); glomerular filtration rate (<60 , $60\text{--}90$, $\geq 90 \text{ mL/min/1.73 m}^2$, and unknown); smoking status (non-, ex-, current smoker, and unknown); alcoholic drinks (<1 , $1\text{--}2$, $3\text{--}4$, ≥ 5 days per week, and unknown); aerobic physical activity (sufficient, insufficient, and unknown); previous history of cardiovascular disease, chronic kidney disease, and chronic obstructive pulmonary disease; history of medication use for diabetes mellitus, dyslipidemia, and hypertension; and missing indicators (BMI missing indicator [yes or no], blood pressure missing indicator [yes or no], fasting blood glucose missing indicator [yes or no], serum total cholesterol missing indicator [yes or no], glomerular filtration rate missing indicator [yes or no], smoking status missing indicator [yes or no], alcoholic drinks missing indicator [yes or no], and aerobic physical activity missing indicator [yes or no]).

Model 2 (replication cohort A): Adjusted for age (20–39, 40–59, and ≥60 years); sex; Charlson comorbidity index (0, 1, and ≥ 2); BMI (underweight [$<18.5 \text{ kg/m}^2$], normal [$18.5\text{--}23.0 \text{ kg/m}^2$], overweight [$23.0\text{--}25.0 \text{ kg/m}^2$], obese [$\geq 25.0 \text{ kg/m}^2$], and unknown); blood pressure

[revised manuscript text omitted]

Reviewer #3**Comment 0.**

This study presents interesting findings, with one of its primary strengths being the utilization of multiple extensive datasets. However, there are several critical aspects of the study that remain unaddressed, and addressing these would significantly enhance its depth and clarity.

Response:

Thank you for your valuable feedback on our study. We are grateful for your recognition of our effort in utilizing multiple extensive datasets, which is indeed a fundamental strength of our research. Your comment highlights some critical aspects that require further attention, and we acknowledge the importance of these issues for the depth and clarity of our study. Therefore, in response to your suggestions, we have incorporated additional analyses and provided supplementary materials about the datasets utilized. Through this process, we believe that the robustness of our study will be enhanced, and we hope that our modifications will align with your expectations.

Comment 1.

First and foremost, it is important to consider whether the diseases under investigation (the primary outcomes) are accurately captured across all datasets. Given that the use of ICD-10 codes typically signifies diagnoses in hospital settings, it raises questions about the potential exclusion of milder cases that may have been diagnosed in primary care settings. This might introduce a bias, as those who are generally less unwell (and therefore less likely to experience severe COVID-19 outcomes) may also be less prone to severe allergic issues. Can the authors undertake any sensitivity analysis to rule this out or elaborate on this in the discussion?

Response:

Thank you for your detailed comments and guidance. The National Health Insurance Service (NHIS) of South Korea operates a comprehensive health insurance system that includes data not only from hospitals but also from primary care settings. Therefore, the NHIS database suggests the possibility of encompassing a wide range of medical cases, including mild cases diagnosed in primary care.¹ Similarly, the JMDC claims database of Japan is an epidemiological database that accumulates receipts from various health insurance societies and health examination data. It is capable of capturing all medical records even if a patient visits multiple medical institutions, suggesting a high likelihood of including mild patients in primary care settings (<https://www.jmdc.co.jp/en/jmdc-claims-database/>).

[REDACTED]

However, the UK Biobank is a large-scale biomedicine database primarily used for research purposes, collecting data including health and genetic information. Therefore, there is a possibility that mild cases that could have been diagnosed in primary care settings may have been excluded. We admit that the use of ICD-10 codes to define primary outcomes may exclude some mild cases and may lead to bias.

As per your suggestion, we conducted sensitivity analyses and examined two distinct patient groups: mild COVID-19 patients and moderate to severe COVID-19 patients (Table S25-S26). According to the sensitivity analyses, we discovered significant associations between SARS-CoV-2 infection and allergic outcomes in most cases, with the results primarily showing higher incidences in patients with moderate to severe COVID-19. This observation can be attributed to the fact that moderate to severe COVID-19 is often associated with an elevated immune response and the release of various cytokines and inflammatory markers.² Such an excessive immune response can potentially induce or exacerbate allergic reactions.² However, despite these additional sensitivity analyses, we acknowledge the potential limitation of not being able to capture every detail regarding mild cases. Therefore, to enhance the completeness of the manuscript, we have supplemented the content in the “Limitations and Strengths” section. We hope our alternations are in line with your expectations.

<Reference>

1. Cheol Seong S, Kim YY, Khang YH, Heon Park J, Kang HJ, Lee H, Do CH, Song JS, Hyon Bang J, Ha S, Lee EJ, Ae Shin S. Data Resource Profile: The National Health Information Database of the National Health Insurance Service in South Korea. *Int J Epidemiol.* 2017 Jun 1;46(3):799-800. doi: 10.1093/ije/dyw253. PMID: 27794523; PMCID: PMC5837262.
2. Que Y, Hu C, Wan K, Hu P, Wang R, Luo J, Li T, Ping R, Hu Q, Sun Y, Wu X, Tu L,

Du Y, Chang C, Xu G. Cytokine release syndrome in COVID-19: a major mechanism of morbidity and mortality. *Int Rev Immunol.* 2022;41(2):217-230. doi: 10.1080/08830185.2021.1884248. Epub 2021 Feb 22. PMID: 33616462; PMCID: PMC7919105.

Changes in text:

Methods, *Sensitivity analysis*

Fifth, in order to examine the impact of COVID-19 severity on allergic diseases, the mild group and the moderate to severe group were analyzed as two separate cohorts (Table S25-S26).

Results

Additionally, we conducted sensitivity analyses to examine the impact of COVID-19 severity on allergic diseases. According to these analyses, the results primarily indicated higher incidences in patients with moderate to severe COVID-19.

Discussion, *Limitations and strengths*

Eighth, we conducted additional sensitivity analyses to capture mild cases of COVID-19 as comprehensively as possible (Tables S25-S26). However, the potential exclusion of milder cases still exists. Additionally, our data may be biased due to different treatment methods for COVID-19 patients based on the severity of their illness.

Table S25. The HR with 95% CI for the long-term sequelae risk of incident allergic diseases following mild COVID-19 diagnosis in the propensity score-matched main cohort (South Korea), replication cohort A (Japan), and replication cohort B (UK).

Cohort	Exposure	Incidence rate*	HR (95% CI)	
			Model 1 ^a	Model 2 ^b
Allergic diseases				
Main cohort	Non-infected	23.3	1.00 (reference)	1.00 (reference)
Main cohort	Patients with COVID-19	26.8	1.15 (1.08 to 1.23)	1.15 (1.08 to 1.23)
Replication cohort A	Non-infected	27.4	1.00 (reference)	1.00 (reference)
Replication cohort A	Patients with COVID-19	75.0	2.64 (2.61 to 2.68)	2.56 (2.52 to 2.59)
Replication cohort B	Non-infected	7.2	1.00 (reference)	1.00 (reference)
Replication cohort B	Patients with COVID-19	8.2	1.13 (1.05 to 1.21)	1.11 (1.04 to 1.19)
Asthma				
Main cohort	Non-infected	0.9	1.00 (reference)	1.00 (reference)
Main cohort	Patients with COVID-19	1.9	2.17 (1.67 to 2.83)	2.18 (1.67 to 2.84)
Replication cohort A	Non-infected	2.7	1.00 (reference)	1.00 (reference)
Replication cohort A	Patients with COVID-19	7.7	2.81 (2.68 to 2.94)	2.67 (2.55 to 2.80)
Replication cohort B	Non-infected	3.3	1.00 (reference)	1.00 (reference)
Replication cohort B	Patients with COVID-19	3.8	1.15 (1.04 to 1.28)	1.13 (1.01 to 1.25)
Allergic rhinitis				
Main cohort	Non-infected	16.9	1.00 (reference)	1.00 (reference)
Main cohort	Patients with COVID-19	20.1	1.19 (1.10 to 1.28)	1.19 (1.11 to 1.29)
Replication cohort A	Non-infected	20.4	1.00 (reference)	1.00 (reference)
Replication cohort A	Patients with COVID-19	62.4	2.97 (2.92 to 3.02)	2.88 (2.83 to 2.93)
Replication cohort B	Non-infected	1.1	1.00 (reference)	1.00 (reference)
Replication cohort B	Patients with COVID-19	1.4	1.22 (1.03 to 1.46)	1.21 (1.02 to 1.44)
Atopic dermatitis				

Main cohort	Non-infected	2.5	1.00 (reference)	1.00 (reference)
Main cohort	Patients with COVID-19	2.9	1.16 (0.95 to 1.41)	1.16 (0.95 to 1.41)
Replication cohort A	Non-infected	5.2	1.00 (reference)	1.00 (reference)
Replication cohort A	Patients with COVID-19	6.8	1.24 (1.19 to 1.29)	1.21 (1.16 to 1.26)
Replication cohort B	Non-infected	0.04	1.00 (reference)	1.00 (reference)
Replication cohort B	Patients with COVID-19	0.05	1.18 (0.47 to 2.99)	1.17 (0.46 to 2.98)
Food allergy				
Main cohort	Non-infected	3.4	1.00 (reference)	1.00 (reference)
Main cohort	Patients with COVID-19	2.6	0.76 (0.63 to 0.93)	0.76 (0.63 to 0.93)
Replication cohort A	Non-infected	1.3	1.00 (reference)	1.00 (reference)
Replication cohort A	Patients with COVID-19	2.6	2.05 (1.90 to 2.20)	1.82 (1.69 to 1.96)
Replication cohort B	Non-infected	3.1	1.00 (reference)	1.00 (reference)
Replication cohort B	Patients with COVID-19	3.4	1.07 (0.96 to 1.20)	1.06 (0.95 to 1.18)

BMI, body mass index; CI, confidence interval; HR, hazard ratio.

* Incidence rate is expressed as per 1,000 person-years.

The data in bold indicate significant differences ($P < 0.05$).

^a **Model 1:** Adjusted for age (20–39, 40–59, and ≥ 60 years) and sex.

^b **Model 2 (main cohort):** Adjusted for age (20–39, 40–59, and ≥ 60 years); sex; household income (low income, middle income, and high income); region of residence (urban and rural); Charlson comorbidity index (0, 1, and ≥ 2); BMI (underweight [$< 18.5 \text{ kg/m}^2$], normal [$18.5\text{--}23.0 \text{ kg/m}^2$], overweight [$23.0\text{--}25.0 \text{ kg/m}^2$], obese [$\geq 25.0 \text{ kg/m}^2$], and unknown); blood pressure (systolic blood pressure $< 140 \text{ mmHg}$ and diastolic blood pressure $< 90 \text{ mmHg}$, systolic blood pressure $\geq 140 \text{ mmHg}$ or diastolic blood pressure $\geq 90 \text{ mmHg}$, and unknown); fasting blood glucose (< 100 , $\geq 100 \text{ mg/dL}$, and unknown); serum total cholesterol (< 200 , $200\text{--}240$, $\geq 240 \text{ mg/dL}$, and unknown); glomerular filtration rate

[revised manuscript text omitted]

Table S26. The HR with 95% CI for the long-term sequelae risk of incident allergic diseases following moderate to severe COVID-19 diagnosis in the propensity score-matched main cohort (South Korea), replication cohort A (Japan), and replication cohort B (UK).

Cohort	Exposure	Incidence rate*	HR (95% CI)	
			Model 1 ^a	Model 2 ^b
Allergic diseases				
Main cohort	Non-infected	27.8	1.00 (reference)	1.00 (reference)
Main cohort	Patients with COVID-19	39.3	1.41 (1.24 to 1.61)	1.42 (1.24 to 1.61)
Replication cohort A	Non-infected	27.6	1.00 (reference)	1.00 (reference)
Replication cohort A	Patients with COVID-19	81.0	2.93 (2.68 to 3.20)	2.75 (2.49 to 3.03)
Replication cohort B	Non-infected	5.8	1.00 (reference)	1.00 (reference)
Replication cohort B	Patients with COVID-19	12.2	2.09 (1.04 to 4.20)	1.97 (0.95 to 4.06)
Asthma				
Main cohort	Non-infected	1.5	1.00 (reference)	1.00 (reference)
Main cohort	Patients with COVID-19	4.0	2.59 (1.66 to 4.03)	2.47 (1.58 to 3.85)
Replication cohort A	Non-infected	3.1	1.00 (reference)	1.00 (reference)
Replication cohort A	Patients with COVID-19	4.4	1.43 (1.03 to 1.99)	1.19 (0.83 to 1.71)
Replication cohort B	Non-infected	1.7	1.00 (reference)	1.00 (reference)
Replication cohort B	Patients with COVID-19	5.6	3.3 (1.06 to 10.25)	3.33 (1.05 to 10.60)
Allergic rhinitis				
Main cohort	Non-infected	19.4	1.00 (reference)	1.00 (reference)
Main cohort	Patients with COVID-19	27.7	1.42 (1.22 to 1.66)	1.43 (1.22 to 1.67)
Replication cohort A	Non-infected	21.0	1.00 (reference)	1.00 (reference)
Replication cohort A	Patients with COVID-19	67.2	3.19 (2.89 to 3.53)	3.09 (2.77 to 3.45)
Replication cohort B	Non-infected	1.2	1.00 (reference)	1.00 (reference)
Replication cohort B	Patients with COVID-19	0	NA	NA
Atopic dermatitis				

Main cohort	Non-infected	2.8	1.00 (reference)	1.00 (reference)
Main cohort	Patients with COVID-19	3.0	1.10 (0.70 to 1.73)	1.11 (0.70 to 1.74)
Replication cohort A	Non-infected	4.2	1.00 (reference)	1.00 (reference)
Replication cohort A	Patients with COVID-19	6.8	1.64 (1.25 to 2.15)	1.57 (1.16 to 2.11)
Replication cohort B	Non-infected	0	1.00 (reference)	1.00 (reference)
Replication cohort B	Patients with COVID-19	0	NA	NA
Food allergy				
Main cohort	Non-infected	4.6	1.00 (reference)	1.00 (reference)
Main cohort	Patients with COVID-19	5.4	1.18 (0.84 to 1.67)	1.18 (0.84 to 1.66)
Replication cohort A	Non-infected	1.5	1.00 (reference)	1.00 (reference)
Replication cohort A	Patients with COVID-19	5.4	3.77 (2.64 to 5.39)	2.53 (1.70 to 3.77)
Replication cohort B	Non-infected	2.9	1.00 (reference)	1.00 (reference)
Replication cohort B	Patients with COVID-19	7.5	2.47 (0.97 to 6.28)	1.99 (0.74 to 5.39)

BMI, body mass index; CI, confidence interval; HR, hazard ratio.

* Incidence rate is expressed as per 1,000 person-years.

The data in bold indicate significant differences ($P < 0.05$).

^a **Model 1:** Adjusted for age (20–39, 40–59, and ≥ 60 years) and sex.

^b **Model 2 (main cohort):** Adjusted for age (20–39, 40–59, and ≥ 60 years); sex; household income (low income, middle income, and high income); region of residence (urban and rural); Charlson comorbidity index (0, 1, and ≥ 2); BMI (underweight [$< 18.5 \text{ kg/m}^2$], normal [$18.5\text{--}23.0 \text{ kg/m}^2$], overweight [$23.0\text{--}25.0 \text{ kg/m}^2$], obese [$\geq 25.0 \text{ kg/m}^2$], and unknown); blood pressure (systolic blood pressure $< 140 \text{ mmHg}$ and diastolic blood pressure $< 90 \text{ mmHg}$, systolic blood pressure $\geq 140 \text{ mmHg}$ or diastolic blood pressure $\geq 90 \text{ mmHg}$, and unknown); fasting blood glucose (< 100 , $\geq 100 \text{ mg/dL}$, and unknown); serum total cholesterol (< 200 , $200\text{--}240$, $\geq 240 \text{ mg/dL}$, and unknown); glomerular filtration rate

[revised manuscript text omitted]

Comment 2.

Second, it's important to acknowledge that the datasets used in this study may exhibit substantial variation. For instance, the UK BioBank relies on voluntary participation, introducing potential selection bias that could impact the results. For readers who may not be familiar with these datasets, it would be highly beneficial if the authors could provide a clear exposition of their key characteristics and any fundamental distinctions between them. A pertinent question to consider is whether the other datasets also rely on voluntary participation. Providing such context would be invaluable.

Response:

Thank you for your insightful comments on the fundamental issues. As you mentioned, there are significant differences in the extent of voluntariness, with the K-COV-N and JMDC data relating to insurance claims (involuntary participation), whereas the UK Biobank data consists of self-reported information (voluntary participation). We acknowledge that there may be potential selection bias based on how participants are included in the analysis, and we found that this difference was not clearly explained in the manuscript. In this revision, we have incorporated these fundamental differences more explicitly in the methods, and we have also included a discussion of their limitations in the “Limitations and Strengths” section to enhance the completeness of the study. Additionally, recognizing that the descriptions of the cohorts were not sufficiently detailed, we have included more comprehensive information about the main cohort (K-COV-N) in the Method and replication cohorts (JMDC and UK Biobank) in the supplementary material. We hope this will serve as a useful resource for readers who may not have prior knowledge of each dataset, aiding in their understanding of our study.

Changes in text:

Methods, *Patient and public involvement*

In the case of the main cohort and replication cohort A, the outcome measures were determined independently, without any involvement from the participants. In contrast, for replication cohort B, the participants were directly involved in determining the outcome measures through a process of voluntary reporting. The study design and implementation were conducted without consultation. However, we plan to disseminate the results of this study to all study participants and wider relevant communities upon request.

Discussion, *Limitations and strengths*

Sixth, we used three cohorts with different reporting formats (self-report for the UK Biobank cohort and insurance claims for the K-COV-N and JMDC cohorts). However, the results were aligned with one another, which strengthens the robustness of the study.

Supplement material. Explanation of replication cohorts (Japan and the UK).

JMDC (replication cohort A)

Data source

Japan has health insurance provided by the universal insurance system.¹ The JMDC has contracts with over 60 insurance providers and includes health insurance claims records data of insured individuals who are primarily employees of relatively large companies in Japan.² JMDC has created a database, using data collected from medical institutions in Japan, consisting of patient-level data (unique identifier, family identifiers, relationship to the insured individual, age, sex) and claims for inpatient and outpatient treatment (disease class according to International Classification of Diseases [ICD]-10 code, prescribed drugs based on Anatomical Therapeutic Chemical class, and drug dosage form), diagnosis or therapeutic

procedure, institutional information (hospital character), and health checkup (i.e., body mass index, blood pressure, clinical laboratory test, medication status, and self-administered questionnaire such as smoking status, alcohol consumption, and physical activity).³

For more information, please see this webpage (<https://www.jmdc.co.jp/en/jmdc-claims-database/>).

UK Biobank (replication cohort B)

Data source

The UK Biobank, which is one of the datasets that rely on voluntary participation, is a globally recognized institution providing unique and extensive data for large-scale research through biomedical samples and health information.⁴ Launched in 2006 in the UK, this project has engaged numerous researchers who utilize participant health data and biological samples for diverse medical studies.⁵ With over 500,000 participants recruited nationwide, the cohort represents rich diversity in terms of age, race, gender, and geographical location, ensuring the applicability of research outcomes to diverse populations.⁶ Participants have provided a range of health-related data, including health surveys, physical measurements, blood pressure readings, and the provision of blood and urine samples. This invaluable data is employed in research across various fields, such as genomic analysis, cardiovascular and metabolic diseases,

and cancer research. Genomic data collected from participants contribute to studying genetic traits and variations, providing crucial information for understanding genetic contributions and exploring interactions between genes and environmental factors.⁷

For more information, please see this webpage (<https://www.ukbiobank.ac.uk/learn-more-about-uk-biobank>).

[REDACTED]

Supplement references

1. Reich MR, Ikegami N, Shibuya K, Takemi K. 50 years of pursuing a healthy society in Japan. *Lancet* 378, 1051-1053 (2011).
2. Setogawa N, Ohbe H, Isogai T, Matsui H, Yasunaga H. Characteristics and short-term outcomes of outpatient and inpatient cardiac catheterizations: A descriptive study using a

nationwide claim database in Japan. *J Cardiol* 82, 201-206 (2023).

3. Kaneko H, et al. Medication-Naïve Blood Pressure and Incident Cancers: Analysis of 2 Nationwide Population-Based Databases. *Am J Hypertens* 35, 731-739 (2022).

4. Keyes KM, Westreich D. UK Biobank, big data, and the consequences of non-representativeness. *Lancet (London, England)* 393, 1297 (2019).

5. Backman JD, et al. Exome sequencing and analysis of 454,787 UK Biobank participants. *Nature* 599, 628-634 (2021).

6. Bycroft C, et al. The UK Biobank resource with deep phenotyping and genomic data. *Nature* 562, 203-209 (2018).

7. Ganna A, Ingelsson E. 5 year mortality predictors in 498,103 UK Biobank participants: a prospective population-based study. *Lancet (London, England)* 386, 533-540 (2015).

Comment 3.

Additionally, the paper requires further elaboration on testing strategies. The suggestion that asymptomatic cases might not be identified hints at the possibility that testing was predominantly limited to individuals exhibiting symptoms. This aspect merits more detailed exploration to better comprehend its implications.

Response:

We truly appreciate your invaluable suggestions for our study. As you mentioned, the assessment and management of asymptomatic cases is a crucial element in estimating the level of disease occurrence. South Korea, Japan, and the UK followed up-to-date guidelines for asymptomatic patients with COVID-19 to minimize missed diagnoses and delays in diagnosis.^{1,2} People in close contact with patients who have already been diagnosed, even without any symptoms, are suggested to be monitored to rule out infection. The World Health Organization suggested prioritizing the use of Antigen-detecting rapid diagnostic tests in asymptomatic individuals with a high risk of infection.³ However, the variation in screening and testing exists within countries (e.g. Korea, pooled specimen test;⁴ UK, lateral flow device⁴). Although the three nations dedicate themselves to reducing misdiagnosis and delayed diagnosis, we admit that some asymptomatic cases might not be identified. Therefore, we added this in the “Limitation and Strengths” section to raise the transparency of the current study.

<Reference>

1. Hong KH, Kim GJ, Roh KH, Sung H, Lee J, Kim SY, Kim TS, Park JS, Huh HJ, Park Y, Kim JS, Kim HS, Seong MW, Ryoo NH, Song SH, Lee H, Kwon GC, Yoo CK; COVID-19 Task Force, the Korean Society for Laboratory Medicine and the Bureau of Infectious Disease Diagnosis Control, the Korea Disease Control and Prevention Agency. Update of Guidelines

for Laboratory Diagnosis of COVID-19 in Korea. *Ann Lab Med*. 2022 Jul 1;42(4):391-397. doi: 10.3343/alm.2022.42.4.391. PMID: 35177559; PMCID: PMC8859556.

2. Gao Z, Xu Y, Sun C, Wang X, Guo Y, Qiu S, Ma K. A systematic review of asymptomatic infections with COVID-19. *J Microbiol Immunol Infect*. 2021 Feb;54(1):12-16. doi: 10.1016/j.jmii.2020.05.001. Epub 2020 May 15. PMID: 32425996; PMCID: PMC7227597.

3. Organization WH. Antigen-detection in the diagnosis of SARS-CoV-2 infection: interim guidance, 6 October 2021: World Health Organization, 2021.

4. Arevalo-Rodriguez I, Seron P, Buitrago-García D, Ciapponi A, Muriel A, Zambrano-Achig P, Del Campo R, Galán-Montemayor JC, Simancas-Racines D, Perez-Molina JA, Khan KS, Zamora J. Recommendations for SARS-CoV-2/COVID-19 testing: a scoping review of current guidance. *BMJ Open*. 2021 Jan 6;11(1):e043004. doi: 10.1136/bmjopen-2020-043004. PMID: 33408209; PMCID: PMC7789202.

Changes in text:

Discussion, *Limitations and strengths*

Ninth, all asymptomatic cases may not be identified in the cohorts in spite of the dedication of governments to reducing misdiagnosis.

Comment 4.

Furthermore, the paper should address the extent of its population coverage and whether it encompasses a representative cross-section of individuals.

Response:

Thank you for your comment. We agree that understanding the representativeness of the data used in our analysis is crucial for interpreting the results. However, we acknowledge that our manuscript did not address this aspect adequately. Therefore, we have added more comprehensive information about the main cohort in the Method and replication cohorts in the supplementary material to reflect this point, ensuring a clearer understanding of the extent of population coverage and its representativeness.

Changes in text:

Methods, *Data source*

Both the K-COV-N and JMDC cohorts employ a universal health insurance system. The UKB, meanwhile, is a dataset comprised of voluntary participation, including biomedical samples and health information. Detailed explanations of the JMDC and UKB cohorts can be found in the supplemental method section.

Methods, *K-COV-N cohort (main)*

The K-COV-N cohort is a large-scale, nationwide, general population-based cohort in South Korea, covering 98% of the South Korean population.¹ The cohort was developed and provided by the National Health Insurance Service of South Korea (NHIS) and Korea Disease Control and Prevention Agency (KDCA) focused on individuals aged ≥ 20 years between January 1, 2018, and December 31, 2021.

Supplement material. Explanation of replication cohorts (Japan and the UK).

JMDC (replication cohort A)

Data source

Japan has health insurance provided by the universal insurance system.¹ The JMDC has contracts with over 60 insurance providers and includes health insurance claims records data of insured individuals who are primarily employees of relatively large companies in Japan.² JMDC has created a database, using data collected from medical institutions in Japan, consisting of patient-level data (unique identifier, family identifiers, relationship to the insured individual, age, sex) and claims for inpatient and outpatient treatment (disease class according to International Classification of Diseases [ICD]-10 code, prescribed drugs based on Anatomical Therapeutic Chemical class, and drug dosage form), diagnosis or therapeutic procedure, institutional information (hospital character), and health checkup (i.e., body mass index, blood pressure, clinical laboratory test, medication status, and self-administered questionnaire such as smoking status, alcohol consumption, and physical activity).²

For more information, please see this webpage (<https://www.jmdc.co.jp/en/jmdc-claims-database/>).

UK Biobank (Replication cohort B)

Data source

The UK Biobank, which is one of the datasets that rely on voluntary participation, is a globally recognized institution providing unique and extensive data for large-scale research through biomedical samples and health information.³ Launched in 2006 in the UK, this project has engaged numerous researchers who utilize participant health data and biological samples for diverse medical studies.⁴ With over 500,000 participants recruited nationwide, the cohort represents rich diversity in terms of age, race, gender, and geographical location, ensuring the applicability of research outcomes to diverse populations.⁵ Participants have provided a range of health-related data, including health surveys, physical measurements, blood pressure readings, and the provision of blood and urine samples. This invaluable data is employed in research across various fields, such as genomic analysis, cardiovascular and metabolic diseases, and cancer research. Genomic data collected from participants contribute to studying genetic traits and variations, providing crucial information for understanding genetic contributions and exploring interactions between genes and environmental factors.⁶

For more information, please see this webpage (<https://www.ukbiobank.ac.uk/learn-more-about-uk-biobank>).

[REDACTED]

Supplement references

1. Reich MR, Ikegami N, Shibuya K, Takemi K. 50 years of pursuing a healthy society in Japan. *Lancet* 378, 1051-1053 (2011).
2. Setogawa N, Ohbe H, Isogai T, Matsui H, Yasunaga H. Characteristics and short-term outcomes of outpatient and inpatient cardiac catheterizations: A descriptive study using a nationwide claim database in Japan. *J Cardiol* 82, 201-206 (2023).
3. Kaneko H, et al. Medication-Naïve Blood Pressure and Incident Cancers: Analysis of 2 Nationwide Population-Based Databases. *Am J Hypertens* 35, 731-739 (2022).
4. Keyes KM, Westreich D. UK Biobank, big data, and the consequences of non-representativeness. *Lancet (London, England)* 393, 1297 (2019).
5. Backman JD, et al. Exome sequencing and analysis of 454,787 UK Biobank participants. *Nature* 599, 628-634 (2021).
6. Bycroft C, et al. The UK Biobank resource with deep phenotyping and genomic data. *Nature* 562, 203-209 (2018).
7. Ganna A, Ingelsson E. 5 year mortality predictors in 498,103 UK Biobank participants: a prospective population-based study. *Lancet (London, England)* 386, 533-540 (2015).

Comment 5.

With regard to vaccination, it would be pertinent to examine the potential effects over time, including waning immunity.

Response:

Thank you for your valuable comments. Exploring how the time passed after vaccination affects allergic conditions can deepen our understanding by monitoring the shifts in immune responses over time. However, grouping various types of vaccines together for analysis might lead to missing out on the unique immunogenicity and waning patterns of each vaccine.¹ Therefore, it's important to pay attention to the distinct immunogenicity of different vaccines like Ad5 or Pfizer work. Since the type of vaccine can greatly impact allergic outcomes, we decided to focus only on mRNA vaccines.

We designed an analysis to investigate potential impacts such as a decline in immune protection following COVID-19 vaccination over time (Table S28). For this study, we used non-infected individuals as a control and classified patients with COVID-19 into four groups based on the timing of infection after vaccination and the number of vaccinations: (1) patients infected with COVID-19 within 30 days prior to vaccination once, (2) patients infected with COVID-19 after 30 days prior to vaccination once, (3) patients with COVID-19 within 30 days prior to vaccination ≥ 2 times, and (4) patients infected with COVID-19 after 30 days prior to vaccination ≥ 2 times. Then, we evaluated the long-term sequelae of allergic diseases following SARS-CoV-2 infection for each group.

As a result, patients infected with COVID-19 within 30 days prior to vaccination once showed a significant increase in long-term allergic outcomes while the other groups did not. Neutralizing antibodies are not directly elicited right after the initial immunization. As it takes 21 days to make sufficient immune response after the first dose of mRNA vaccine,² patients

infected with COVID-19 within 30 days after once vaccinated are at a high risk of allergy. After 30 days prior to the first immunization, a greater immune response with high antibody titer is shown and it ameliorates long-term sequelae of allergic diseases. The revival of immune response due to boosters helped to maintain the ameliorated effect against allergic outcomes following SARS-CoV-2 infection. Thank you for your time and valuable feedback.

<Reference>

1. Pérez-Alós L, Armenteros JJA, Madsen JR, Hansen CB, Jarlhelt I, Hamm SR, Heftdal LD, Pries-Heje MM, Møller DL, Fogh K, Hasselbalch RB, Rosbjerg A, Brunak S, Sørensen E, Larsen MAH, Ostrowski SR, Frikke-Schmidt R, Bayarri-Olmos R, Hilsted LM, Iversen KK, Bundgaard H, Nielsen SD, Garred P. Modeling of waning immunity after SARS-CoV-2 vaccination and influencing factors. *Nat Commun.* 2022 Mar 28;13(1):1614. doi: 10.1038/s41467-022-29225-4. PMID: 35347129; PMCID: PMC8960902.
2. Chaudhary N, Weissman D, Whitehead KA. mRNA vaccines for infectious diseases: principles, delivery and clinical translation. *Nat Rev Drug Discov.* 2021 Nov;20(11):817-838. doi: 10.1038/s41573-021-00283-5. Epub 2021 Aug 25. Erratum in: *Nat Rev Drug Discov.* 2021 Sep 21;: PMID: 34433919; PMCID: PMC8386155.

Changes in text:

Methods, *Sensitivity analysis*

In the same context, we conducted a time attenuation analysis to identify potential impacts, including the decrease in immunity over time (Table S28).

Table S28. The HR with 95% CI for the long-term sequelae risk of incident allergic disease by mRNA vaccine dose over time following COVID-19 diagnosis in the propensity score-matched main cohort (South Korea).

Factors	Group	Incidence rate *	HR (95% CI)	
			Model 1 ^a	Model 2 ^b
Allergic disease				
Number of SARS-CoV-2 vaccinations	Non-infected control	19.5	1.00 (reference)	1.00 (reference)
	Vaccination 1 time			
	Patients infected with COVID-19 within 30 days	31.8	1.70 (1.30 to 2.23)	1.69 (1.29 to 2.22)
	Patients infected with COVID-19 after 30 days	23.9	1.24 (0.71 to 2.16)	1.23 (0.71 to 2.14)
	Vaccination ≥2 times			
	Patients infected with COVID-19 within 30 days	20.6	1.01 (0.59 to 1.72)	0.97 (0.57 to 1.67)
	Patients infected with COVID-19 after 30 days	25.9	1.27 (0.87 to 1.86)	1.26 (0.86 to 1.84)
	Asthma			
Number of SARS-CoV-2 vaccinations	Non-infected control	0.7	1.00 (reference)	1.00 (reference)
	Vaccination 1 time			
	Patients infected with COVID-19 within 30 days	2.5	4.35 (1.51 to 12.53)	3.94 (1.35 to 11.51)
	Patients infected with COVID-19 after 30 days	1.8	3.17 (0.41 to 24.71)	3.20 (0.41 to 25.23)
	Vaccination ≥2 times			
	Patients infected with COVID-19 within 30 days	1.5	1.85 (0.24 to 14.34)	1.71 (0.22 to 13.37)
	Patients infected with COVID-19 after 30 days	0.9	1.36 (0.18 to 10.52)	1.39 (0.18 to 10.79)

Allergic rhinitis

Number of SARS-CoV-2 vaccinations	Non-infected control	14.6	1.00 (reference)	1.00 (reference)
Vaccination 1 time	Patients infected with COVID-19 within 30 days	24.2	1.72 (1.26 to 2.34)	1.72 (1.26 to 2.35)
	Patients infected with COVID-19 after 30 days	20.3	1.37 (0.75 to 2.51)	1.36 (0.75 to 2.50)
Vaccination ≥ 2 times	Patients infected with COVID-19 within 30 days	19.1	1.27 (0.73 to 2.22)	1.24 (0.71 to 2.17)
	Patients infected with COVID-19 after 30 days	21.4	1.40 (0.92 to 2.12)	1.37 (0.90 to 2.09)

Atopic dermatitis

Number of SARS-CoV-2 vaccinations	Non-infected control	2.2	1.00 (reference)	1.00 (reference)
Vaccination 1 time	Patients infected with COVID-19 within 30 days	2.0	1.01 (0.36 to 2.84)	1.01 (0.36 to 2.84)
	Patients infected with COVID-19 after 30 days	0.0	NA	NA
Vaccination ≥ 2 times	Patients infected with COVID-19 within 30 days	0.0	NA	NA
	Patients infected with COVID-19 after 30 days	1.8	0.81 (0.19 to 3.35)	0.80 (0.19 to 3.34)

Food allergy

Number of SARS-CoV-2 vaccinations	Non-infected control	2.4	1.00 (reference)	1.00 (reference)
Vaccination 1 time	Patients infected with COVID-19 within 30 days	3.5	1.52 (0.68 to 3.39)	1.40 (0.63 to 3.13)
	Patients infected with COVID-19 after 30 days	1.8	0.77 (0.11 to 5.60)	0.74 (0.10 to 5.37)

Vaccination ≥ 2 times	Patients infected with COVID-19 within 30 days	0.0	NA	NA
	Patients infected with COVID-19 after 30 days	1.8	0.74 (0.18 to 3.04)	0.73 (0.18 to 3.01)

BMI, body mass index; CI, confidence interval; HR, hazard ratio.

* Incidence rate is expressed as per 1,000 person-years.

The data in bold indicate significant differences ($P < 0.05$).

^a **Model 1:** Adjusted for age (20–39, 40–59, and ≥ 60 years) and sex.

^b **Model 2:** Adjusted for age (20–39, 40–59, and ≥ 60 years); sex; household income (low income, middle income, and high income); region of residence (urban and rural); Charlson comorbidity index (0, 1, and ≥ 2); BMI (underweight [$< 18.5 \text{ kg/m}^2$], normal [$18.5\text{--}23.0 \text{ kg/m}^2$], overweight [$23.0\text{--}25.0 \text{ kg/m}^2$], obese [$\geq 25.0 \text{ kg/m}^2$], and unknown); blood pressure (systolic blood pressure $< 140 \text{ mmHg}$ and diastolic blood pressure $< 90 \text{ mmHg}$, systolic blood pressure $\geq 140 \text{ mmHg}$ or diastolic blood pressure $\geq 90 \text{ mmHg}$, and unknown); fasting blood glucose (< 100 , $\geq 100 \text{ mg/dL}$, and unknown); serum total cholesterol (< 200 , $200\text{--}240$, $\geq 240 \text{ mg/dL}$, and unknown); glomerular filtration rate (< 60 , $60\text{--}90$, $\geq 90 \text{ mL/min/1.73 m}^2$, and unknown); smoking status (non-, ex-, current smoker, and unknown); alcoholic drinks (< 1 , $1\text{--}2$, $3\text{--}4$, ≥ 5 days per week, and unknown); aerobic physical activity (sufficient, insufficient, and unknown); previous history of cardiovascular disease, chronic kidney disease, and chronic obstructive pulmonary disease; history of medication use for diabetes mellitus, dyslipidemia, and hypertension; and missing indicators (BMI missing indicator [yes or no], blood pressure missing indicator [yes or no], fasting blood glucose missing indicator [yes or no], serum total cholesterol missing indicator [yes or no], glomerular filtration rate missing indicator [yes or no], smoking status missing indicator [yes or no], alcoholic drinks missing indicator [yes or no], and aerobic physical activity missing indicator [yes or no]).

REVIEWER COMMENTS

Reviewer #1 (Remarks to the Author):

The reviewer's questions and comments were addressed in a detailed and satisfactory manner. Thank you.

Reviewer #2 (Remarks to the Author):

The authors have carefully corrected the manuscript as suggested by the reviewers. They have re-run all the analyses and included a description of the data sources. The descriptive tables on missing data are now plausible.

However, the authors have not been able to explain why the results from the Japan cohort are so different from those from the Korea cohort and the UK Biobank. Without the Japanese cohort, only a high relative risk for asthma in the Korean cohort would remain. The authors excluded cases of dyspnoea. However, asthma may have been coded instead of the correct diagnosis of dyspnoea, as COVID-19 is a new disease. The authors also cited Zheng et al. 2018 to explain why a viral infection could cause asthma. The review did not look at the onset of asthma, but at the exacerbation of asthma. The attenuation effect shown in Table 3 also supports the idea that the SARS-CoV-2 virus exacerbates pre-existing allergic diseases rather than causing them. Even after excluding people with ICD-10 codes for allergic diseases in the baseline period.

Zheng, X.-y., Xu, Y.-j., Guan, W.-j. & Lin, L.-f. Regional, age and respiratory-secretion-specific prevalence of respiratory viruses associated with asthma exacerbation: a literature review. Archives of virology 163, 845-853 (2018).

Reviewer #3 (Remarks to the Author):

The authors have adequately addressed all the points raised and I am happy with the responses and the ensuing changes.

Response Letter

Table of Contents

1. Reviewer #1 2

2. Reviewer #2..... 3

3. Reviewer #3.....11

REVIEWER COMMENTS

Reviewer #1 (Remarks to the Author):

The reviewer's questions and comments were addressed in a detailed and satisfactory manner.

Thank you.

Response:

We sincerely appreciate your positive evaluation and am grateful for the time and effort you devoted to reviewing our manuscript. Your detailed questions and comments have significantly contributed to enhancing the quality of our work. It was our utmost priority to thoroughly address each point you raised, ensuring our manuscript's clarity, accuracy, and contribution to the field are maximized. Your insightful feedback has been invaluable in guiding our revisions and enhancing the overall quality of our work. We appreciate the time and effort you dedicated to reviewing our manuscript.

Thank you once again for your constructive feedback and support throughout the revision process. We are hopeful that our work will make a meaningful addition to the existing body of knowledge and look forward to the opportunity of contributing further to our field.

Reviewer #2 (Remarks to the Author):

The authors have carefully corrected the manuscript as suggested by the reviewers. They have re-run all the analyses and included a description of the data sources. The descriptive tables on missing data are now plausible.

However, the authors have not been able to explain why the results from the Japan cohort are so different from those from the Korea cohort and the UK Biobank. Without the Japanese cohort, only a high relative risk for asthma in the Korean cohort would remain. The authors excluded cases of dyspnoea. However, asthma may have been coded instead of the correct diagnosis of dyspnoea, as COVID-19 is a new disease. The authors also cited Zheng et al. 2018 to explain why a viral infection could cause asthma. The review did not look at the onset of asthma, but at the exacerbation of asthma. The attenuation effect shown in Table 3 also supports the idea that the SARS-CoV-2 virus exacerbates pre-existing allergic diseases rather than causing them. Even after excluding people with ICD-10 codes for allergic diseases in the baseline period.

Zheng, X.-y., Xu, Y.-j., Guan, W.-j. & Lin, L.-f. Regional, age and respiratory-secretion-specific prevalence of respiratory viruses associated with asthma exacerbation: a literature review. Archives of virology 163, 845-853 (2018).

Response:

Thank you for the constructive feedback on our manuscript. We acknowledge that our understanding and explanation regarding your main comment were insufficient in the previous revision. Therefore, in this revision, we aim to address the aspects we previously overlooked and thoroughly enhance the sections where explanations were lacking.

First, we acknowledge that despite conducting a negative control analysis, we were

fundamentally unable to explain the persistent differences observed in the Japan cohort compared to the other two cohorts. Regarding the cause of these differences, two possibilities were considered.

- 1) We figured that the difference may be due to the difference in the method of collecting the data and its structure. We utilized data from Korea cohort, which corresponds to the same claim-based data, to compare the data collection and construction methodologies with those used in Japan. The National Health Insurance Service (NHIS) of Korea cohort is a public medical insurance institution managed by the government, with 98% of the South Korean population enrolled. The NHIS operates a comprehensive health insurance system that includes data not only from hospitals but also from primary care facilities, thereby encompassing a wide range of medical cases, including mild cases diagnosed in primary care.¹ JMDC is a company that constructs and provides epidemiological data from more than 60 insurance providers. Similar to Korea, JMDC allows for the capture of all medical records even if a patient visits multiple medical institutions, making it likely that the data collected includes a wide range of cases (<https://www.jmdc.co.jp/en/jmdc-claims-database/>).

[REDACTED]

However, although the two claim-based datasets are considered to be collected in similar settings in terms of the timing of medical visits, differences in data construction and participant characteristics have been identified. For JMDC, the dataset is constructed primarily around claim data received from over 60 distinct insurance providers.² In addition, it mainly targets employed individuals and their families within Japan, indicating a potential limitation in demographic scope.³ On the other hand, as mentioned above, the Korean dataset is focused on the entire South Korean population. Moreover, there can be inherent differences in the coding of diseases across different countries. Therefore, despite being based on claim data, these structural differences inevitably imply the possibility of leading to variations in the outcomes.

- 2) It may be assumed that environmental and cultural factors unique to the population potentially influence the outcomes. Allergy is a disease where the recognition and diagnosis distinctly reflect cultural and ethnic contexts.⁴ This variation underscores the complexity of allergy as a global health issue, illustrating how cultural practices, dietary habits could significantly affect the prevalence and expression of allergic diseases across different populations.^{5,6} Furthermore, the 'hygiene hypothesis' suggests that the incidence of allergic disorders is linked to exposure to microbes, the size of family, and hygiene standards, positing a direct correlation between these factors and the immune system's development and functioning.⁷ Taking these insights into consideration, it becomes evident that the incidence and manifestation of allergic diseases could vary significantly from one country to another.

To enhance the completeness of our research, we will address this limitation by integrating these possibilities into the *Limitations and Strengths* section.

Second, we initially excluded individuals who had been diagnosed with asthma but were experiencing dyspnea in the analysis. It seems you were pointing out the possibility that due to the emergence of new diseases like COVID-19, individuals who were truly suffering from dyspnea could have been misclassified as having asthma. This means that those who should have been diagnosed with dyspnea might have been erroneously labeled with asthma, leading to potential confusion in disease diagnosis and classification. The complexities generated by COVID-19 could blur the distinction between asthma and dyspnea, potentially impacting the interpretation of our research findings. Although we used multinational cohorts to mitigate various bias including misclassification, we cannot entirely rule out the possibility of diagnostic errors. This represents a limitation of our study, and we added this consideration to the *Limitations and Strengths* section.

Third, as the reviewer mentioned, the reference we used to explain how viral infections could cause asthma is focused on the exacerbation, not the subsequent development. We improved the plausible mechanism under long-COVID effects on the onset of asthma using several previous studies. Since the early 1970s, there have been efforts to identify whether an infection with viruses, including respiratory syncytial virus⁸ and rhinovirus,⁹ is associated with the onset of asthma. To date, viral infections, in general, are known to stimulate morphological and immunological alternations. This induces pro-inflammatory cytokines from the airway epithelium, increasing allergic sensitization.¹⁰ As SARS-CoV-2 is a novel virus, the association between SARS-COV-2 infection and following asthma initiation has been less investigated. However, a ‘cytokine storm’, linked with the severe form of COVID-19, contributes to hyperinflammation that may be implicated in critical sequelae in respiratory tracts.¹¹ Nonetheless, we admit that the present study has not been able to exclude people with pre-

existing allergic diseases but undiagnosed in the baseline as we used the claims-based cohorts. Thus, we have decided to address this in the *Limitation and Strengths*. We believe that it could be a noteworthy area for further investigations.

<References >

1. Cheol Seong S, Kim YY, Khang YH, Heon Park J, Kang HJ, Lee H, Do CH, Song JS, Hyon Bang J, Ha S, Lee EJ, Ae Shin S. Data Resource Profile: The National Health Information Database of the National Health Insurance Service in South Korea. *Int J Epidemiol*. 2017 Jun 1;46(3):799-800. doi: 10.1093/ije/dyw253. PMID: 27794523; PMCID: PMC5837262.
2. Reich MR, Ikegami N, Shibuya K, Takemi K. 50 years of pursuing a healthy society in Japan. *Lancet*. 2011 Sep 17;378(9796):1051-3. doi: 10.1016/S0140-6736(11)60274-2IF: 168.9 Q1 . Epub 2011 Aug 30. PMID: 21885101.
3. Setogawa N, Ohbe H, Isogai T, Matsui H, Yasunaga H. Characteristics and short-term outcomes of outpatient and inpatient cardiac catheterizations: A descriptive study using a nationwide claim database in Japan. *J Cardiol*. 2023 Sep;82(3):201-206. doi: 10.1016/j.jjcc.2023.05.010. Epub 2023 May 28. PMID: 37247658.
4. Murrison LB, Brandt EB, Myers JB, Hershey GKK. Environmental exposures and mechanisms in allergy and asthma development. *J Clin Invest*. 2019 Apr 1;129(4):1504-1515. doi: 10.1172/JCI124612IF: 15.9 Q1 . Epub 2019 Feb 11. PMID: 30741719; PMCID: PMC6436881.
5. Jafri S, Janzen J, Kim R, Abrams EM, Gruber J, Protudjer JLP. Burden of Allergic Disease in Racial and Ethnic Structurally Oppressed Communities Within Canada and the United States:

- A Scoping Review. *J Allergy Clin Immunol Pract.* 2022 Nov;10(11):2995-3001. doi: 10.1016/j.jaip.2022.08.018IF: 9.4 Q1 . Epub 2022 Aug 19. PMID: 35995399.
6. Prescott SL. Early-life environmental determinants of allergic diseases and the wider pandemic of inflammatory noncommunicable diseases. *J Allergy Clin Immunol.* 2013 Jan;131(1):23-30. doi: 10.1016/j.jaci.2012.11.019IF: 14.2 Q1 . PMID: 23265694.
7. Okada H, Kuhn C, Feillet H, Bach JF. The 'hygiene hypothesis' for autoimmune and allergic diseases: an update. *Clin Exp Immunol.* 2010 Apr;160(1):1-9. doi: 10.1111/j.1365-2249.2010.04139.x. PMID: 20415844; PMCID: PMC2841828.
8. Pullan CR, Hey EN. Wheezing, asthma, and pulmonary dysfunction 10 years after infection with respiratory syncytial virus in infancy. *Br Med J (Clin Res Ed).* 1982 Jun 5;284(6330):1665-9. doi: 10.1136/bmj.284.6330.1665. PMID: 6805648; PMCID: PMC1498624.
9. Liu L, Pan Y, Zhu Y, Song Y, Su X, Yang L, Li M. Association between rhinovirus wheezing illness and the development of childhood asthma: a meta-analysis. *BMJ Open.* 2017 Apr 3;7(4):e013034. doi: 10.1136/bmjopen-2016-013034. PMID: 28373249; PMCID: PMC5387933.
10. Novak N, Cabanillas B. Viruses and asthma: the role of common respiratory viruses in asthma and its potential meaning for SARS-CoV-2. *Immunology.* 2020 Oct;161(2):83-93. doi: 10.1111/imm.13240. Epub 2020 Aug 17. PMID: 32687609; PMCID: PMC7405154.
11. Mehta P, McAuley DF, Brown M, Sanchez E, Tattersall RS, Manson JJ; HLH Across Speciality Collaboration, UK. COVID-19: consider cytokine storm syndromes and immunosuppression. *Lancet.* 2020 Mar 28;395(10229):1033-1034. doi: 10.1016/S0140-

6736(20)30628-0IF: 168.9 Q1 . Epub 2020 Mar 16. PMID: 32192578; PMCID: PMC7270045.

12. Tan C, Zheng X, Sun F, He J, Shi H, Chen M, Tu C, Huang Y, Wang Z, Liang Y, Wu J, Liu Y, Liu J, Huang J. Hypersensitivity may be involved in severe COVID-19. *Clin Exp Allergy*. 2022 Feb;52(2):324-333. doi: 10.1111/cea.14023. Epub 2021 Oct 9. PMID: 34570395; PMCID: PMC8652637.

Changes in text:

Discussion, *Plausible mechanisms*

It is well-established that viral infections, in general, stimulate morphological alternations including tissue remodeling, and trigger immune responses, which contributes to the initiation of allergic diseases.³¹ Moreover, regulatory T cells perturbation driven by long COVID induces uninhibited action of effector cells and enables latent SARS-CoV-2,³² which may lead to post-acute sequelae of allergy. Also, a ‘cytokine storm,’ which is linked to the severe form of COVID-19, contributes to hyperinflammation that may be implicated in critical sequelae in respiratory tracts.^{33,34}

Discussion, *Limitations and strengths*

First, allergy is a disease that the recognition and diagnosis distinctly reflect cultural and ethnic contexts. In addition, the ‘hygiene hypothesis’ suggests the incident allergic disorder is linked to exposure to microbes, size of family, and hygiene standards.³⁷ Although all diagnoses of allergic outcomes were based on the same ICD-10 codes, we observed consistently and remarkably higher incidence rates of allergic diseases in Japan than those of the others. **Second,**

the present study defined disease according to ICD-10 codes; thus, the findings should be interpreted with caution. The potential misclassification of dyspnea as asthma, particularly due to the diagnostic complexities introduced by COVID-19, represents a limitation. Also, although we excluded individuals who had been diagnosed with asthma to focus on the development of asthma, there may be some people with pre-existing asthma but undiagnosed in the baseline period. Therefore, the potential for disease misclassification necessitates a cautious interpretation of the data.

Eighth, we used three cohorts with different reporting formats (self-report for the UKB cohort and insurance claims for the K-COV-N and JMDC cohorts) and construction of dataset.

References

31. Novak, N. & Cabanillas, B. Viruses and asthma: the role of common respiratory viruses in asthma and its potential meaning for SARS-CoV-2. *Immunology* 161, 83-93 (2020). <https://doi.org:10.1111/imm.13240>
33. Mehta, P. et al. COVID-19: consider cytokine storm syndromes and immunosuppression. *Lancet* 395, 1033-1034 (2020). [https://doi.org:10.1016/s0140-6736\(20\)30628-0](https://doi.org:10.1016/s0140-6736(20)30628-0)
34. Tan, C. et al. Hypersensitivity may be involved in severe COVID-19. *Clin Exp Allergy* 52, 324-333 (2022). <https://doi.org:10.1111/cea.14023>

Reviewer #3 (Remarks to the Author):

The authors have adequately addressed all the points raised and I am happy with the responses and the ensuing changes.

Response:

We sincerely appreciate your thoughtful feedback and positive evaluation of our revisions. Your expertise and insightful comments have been instrumental in enhancing the quality and clarity of our manuscript. We are gratified to learn that the responses and changes made have met your expectations and addressed the concerns you raised. Moving forward, we are committed to further exploring the research avenues suggested during the review process and to contributing meaningful advancements to our field. We believe that the dialogue established through this review has not only improved our current work but will also guide our future research directions. Thank you once again for your valuable contributions and for the encouragement provided by your approval of our revisions.